# Number transcoding in bilinguals—A transversal developmental study

**Rémy Lachelin**[1]*, **Amandine van Rinsveld**[2], **Alexandre Poncin**[1], **Christine Schiltz**[1]

**1** Institute of Cognitive Science and Assessment, Department of Behavioural and Cognitive Sciences, University of Luxembourg, Esch-sur-Alzette, Luxembourg, **2** Graduate School of Education, Stanford University, Stanford, CA, United States of America

* remy.lachelin@uni.lu

**Data Availability Statement:** The data can be found at the following OSF URL: https://osf.io/2sjvu/.

## Abstract

Number transcoding is the cognitive task of converting between different numerical codes (*i.e.* visual "42", verbal "forty-two"). Visual symbolic to verbal transcoding and *vice versa* strongly relies on language proficiency. We evaluated transcoding of German-French bilinguals from Luxembourg in 5th, 8th, 11th graders and adults. In the Luxembourgish educational system, children acquire mathematics in German (LM1) until the 7th grade, and then the language of learning mathematic switches to French (LM2). French '70s '80s '90s are less transparent than '30s '40s '50s numbers, since they have a base-20 structure, which is not the case in German. Transcoding was evaluated with a reading aloud and a verbal-visual number matching task. Results of both tasks show a cognitive cost for transcoding numbers having a base-20 structure (*i.e.* '70s, '80s and '90s), such that response times were slower in all age groups. Furthermore, considering only base-10 numbers (*i.e.* '30s '40s '50s), it appeared that transcoding in LM2 (French) also entailed a cost. While participants across age groups tended to read numbers slower in LM2, this effect was limited to the youngest age group in the matching task. In addition, participants made more errors when reading LM2 numbers. In conclusion, we observed an age-independent language effect with numbers having a base-20 structure in French, reflecting their reduced transparency with respect to the decimal system. Moreover, we find an effect of language of math acquisition such that transcoding is less well mastered in LM2. This effect tended to persist until adulthood in the reading aloud task, while in the matching task performance both languages become similar in older adolescents and young adults. This study supports the link between numbers and language, especially highlighting the impact of language on reading numbers aloud from childhood to adulthood.

## Introduction

Exact numerical representations are supported by symbolic verbal (*e.g.* forty-two) and visual (*e.g.* 42) representations which are acquired through learning. However, in many languages the verbal number word's syntax differs from the visual one, *i.e.* the Arabic number system [1, 2]. This difference might have an influence on transcoding, *i.e.* the cognitive transformation from one code to another. Thus reading aloud for instance the number "42" implies the

**Funding:** The first author is supported by the Luxembourg National Research Fund (FNR) (PRIDE/15/10921377). The funders had no role in study design, data collection and analysis, decision to publish, or preparation of the manuscript.

**Competing interests:** The authors have declared that no competing interests exist.

transcoding from a visual ("42") to a verbal code ("forty-two"). Moreover, in the case of bi- and multilingualism, acquiring multiple languages means that during development multiple verbal codes are mapped with the visual code [3]. Since multilinguals are outnumbering monolinguals across the globe [4], the question of how numbers are processed and particularly transcoded using two (or more) verbal codes in a (developing) multilingual cognitive system is of crucial importance [5].

Several cognitive models have been proposed to describe transcoding. While some are taking also into account cognitive development, most of these models do not specifically account for transcoding differences between languages and multilingualism. These models are summarized below in two main categories: semantic and asemantic [6].

In semantic models, transcoding requires an obligatory access of the number's magnitude. For example McCloskey [7, 8] proposed a transcoding model in which the entry number—regardless from the input's format—accesses an abstract semantic representation. Power and Dal Martello [9] further proposed a model specifically for number dictation (*i.e.* verbal to Arabic format) which differs from the previous one in that semantic representations reflect the verbal word structure of the numbers, thus predicting differences between languages. Semantic models hence assume that transcoding difficulties depend on the quality and maturity of the semantic representations, therefore predicting worse performances in children, in particular for larger numbers.

Asemantic models propose that numbers can be transcoded without accessing magnitude. Deloche and Seron [10] proposed such a model for number naming (*i.e.* Arabic to verbal) where the first step involves parsing the input into primitives which are then submitted to a set of rules for the output system. Later on, Barrouillet and colleagues [6] proposed ADAPT (A Developmental Asemantic Procedural Model), explicitly including a developmental perspective on transcoding. In ADAPT, by repeating transcoding, number words become lexicalized with training, hence directly retrieved from long-term memory and bypassing the procedural processes. Therefore transcoding in children would depend on language-dependent characteristics. Yet, the language-dependent characteristics should diminish with increasing lexicalization. Another more general model considering number representations is the Triple Code Model (TCM). The TCM proposes a functional and topographical framework of how and where in the brain approximate, visual symbolic and verbal codes are processed. The TCM also implies asemantic language-dependent transcoding from or to a verbal code [11]. Language-specific training in transcoding would therefore increase the strength of the association between each code and their respective brain areas. More recently, Dotan and Friedmann [12] proposed a model for Arabic to verbal transcoding where numbers are first visually analysed for identity, order and decimal structure to build a language independent number word frame, which is passed to a second system which applies language-specific rules associated with their phonological and articulatory counter-parts. In sum, asemantic models predict that verbal transcoding depends on language characteristics. However those models do not explicitly model multilingualism, nor how transcoding develops by acquiring number words in a second language.

When investigating how the acquisition of several languages (and hence multiple number words) develops, the many forms and constellations of multilingualism must be taken into account; with bilingualism corresponding to the simplest instance. Today, multilingualism is often considered as a continuum that is shaped by numerous aspects such as relative language proficiency [13, 14], as well as age and duration of language learning [15]. Other factors influencing multilingual's profiles also depend on years of education (particularly literacy education), the amount of language input and output, and its privileged context (*e.g.* [16]). Thus, multilinguals often have a first language (L1) which is more dominant than later learned

languages (*i.e.* L2). On top of these factors, the structure of a language such as morphemes and syntax shape each language learning trajectory. These factors might therefore impact transcoding as well as arithmetic in general, since common processes are implicated in both [17–19]. See also [20] for an overview of language-related factors. In the following, we will briefly review two main factors that are critical with regards to our study: *Transparency of power* and *language of math acquisition*.

## Transparency of power

*Transparency of power* refers to the existence of different degrees of morpho-syntactic language transparencies in verbal numbers. In many Asian languages (*i.e.* Mandarin Chinese, Vietnamese) the morpho-syntactic structure of the verbal number system is highly consistent with the Arabic number system, and in general with the base-10 (*i.e.* $10^*x + y$, where x indexes the base and y the unit, see [21]). This linguistic characteristic, also termed *transparency of power*, facilitates learning to count [22–24] and solving arithmetic problems [25–27]. Another morpho-syntactic difference concerns the inversion between tens and ones as for example in Dutch and in German (*i.e.* $y + 10^*x$; in German 42 is "zwei-und-*vierzig*", literally "two-and-*forty*"). This linguistic characteristic, called *transparency of order* [28], explains some transcoding error patterns and reaction times [17, 18, 29–33], gives rise to specific pattern of compatibility effects [34–36] and complicates the solution of certain arithmetic problems [37–39].

   *Transparency of power* can also vary with a change of base, for example a base-20 system ($20^*x + y$), also called vigesimal. The use of the vigesimal system is found for example in French, Basque, or in Diola-Fogny (see [40]). French, to be precise, has a mixed system since it uses base-10 and base-20 systems. Up to the '60s in French, like in English, the counting system follows the base-10 rule. However, '80s to '90s follow a base-20 system, *e.g.* $87 = 4^*20 + 7$, "quatre-vingt-*sept*", literally "*four-twenty*-seven". Moreover, the teens (11 to 16) are lexical primitives, like in English and in stark contrast to the more transparent Asian languages. Note that the transparency contrast is additionally increased for 71 to 76 and 91 to 96, which are composed with a base-20 and teens, *e.g.* $96 = 4^*20 + 16$, "quatre-vingt-*seize*", literally "four-twenty-*sixteen*". Furthermore, the vigesimal system is subject to regional variances, for example in Belgian Wallonia '70s and '90s are not vigesimal (i.e. "*sept*ante" for "*seven*ty" and "*non*ante" for "*nine*ty", see [41]) and in some Swiss-French cantons the vigesimal system is entirely absent (*i.e.* including for 80, "*huit*ante", literally "*eight*y"). In a study comparing French-speaking 1st graders from Wallonia and France with a number dictation task, more mistakes were committed in the '70s and '90s by the latter [41]. In another study comparing English-speaking to French-speaking 5th graders, numbers above 60 were slower to transcode, revealing a cost for base-20 numbers [42]. Further indicating an interaction between the number structure and number processing, Basque-speaking adults solved additions faster when the operand was in the base-20 structure. For example, the solutions for 20 + 15 is facilitated over 25 + 10, since the result 35, is said "*hogeita*-hamabost", literally "*twenty*-fifteen" [43]. Strikingly, for Basque-speakers the distance effect also leads to different event-related potential brain responses for vigesimal compared to decimal Arabic digits [44]. While providing interesting insights into the neuro-cognitive mechanisms of vigesimal number processing, these studies might be limited by potential cultural and educational confounds associated with group comparisons [25] or curricular differences [45]. Hence, between-subject comparisons should be complemented by within-subject designs with multilingual participants. Furthermore, they do not provide information about the developmental trajectories of the acquisition of a language with a different number system since they focus on single age groups. They consequently need to be completed by designs comparing different age groups of multilinguals.

To sum up, the *transparency of power* refers to the morpho-syntactic structure of a language's number word system. Results with different populations and using various tasks consistently indicate an advantage for processing numbers in more *transparent* languages. While some studies started to explore the impact of vigesimal number structures, only a few studies focused on within-subject designs with bilinguals. And to the best of our knowledge, studies on the developmental trajectory in bilingual learners are still entirely missing.

## Language of mathematics acquisition

In a questionnaire asking multilinguals which language they preferentially use for mental calculations, the majority reported a preference for their first language L1, supposedly also their language of math acquisition (LM+) [46]. In contrast, solving arithmetic problems in the non-preferred language resulted in cognitive costs for vocal answers to arithmetic problems presented as Arabic digits [47, 48], auditory [49], or written number words [50]. While highlighting the impact of language dominance when doing math, these designs confound general L1 benefits with potential domain-specific benefits from the language in which mathematic was first learned [51].

Training experiments with bilinguals doing arithmetic in both languages indeed indicate a benefit for solving arithmetic problems in the language in which they were trained. Spelke and Tviskin [52] investigated Russian-English bilinguals, who were trained to solve arithmetic either in the dominant language, L1 (Russian) or their L2 (English). The results showed a cost for solving arithmetical problems when switching to the untrained language, independently if the testing was in the L1 or L2, indicating a Language Switching Cost (LSC). LSC in the context of math training was replicated in 9th and 11th graders attending German-French bilingual education curricula, who were trained to solve arithmetic in German or French [53]. Similar LSC was found in German-French [54] and German-English bilingual adults [55]. Hence, independently from language dominance, arithmetic and mathematical problem solving are facilitated when tested in the same language as they are learned, and they are accompanied by a cost in the untrained language. These findings underline the importance of how multilingual education school curricula are designed.

The LSC generalizes to more ecological learning settings, showing that mathematical problems are solved more accurately in LM+, even if it is not the dominant language (*i.e.* the LM + not being the same language as the L1). For example, Bernardo [56] investigated arithmetic among high school Filipino-English bilinguals who have Filipino as L1 but specifically learned mathematics in English (LM+). The results indicate a cost for arithmetical problems written in number words in Filipino (*i.e.* being the L1 but LM-) compared to English (*i.e.* being an L2 but LM+), which in turn showed comparable results than to problems presented as Arabic digits. The critical role of LM+ or LM1 is also confirmed by studies on highly proficient bilinguals. Note that here we make a distinction between the LM+, defined as the (only) language of learning mathematics, and LM1 defined as the first language of learning mathematics (which is followed by other languages used later in the learning process). For example in school curricula where the first language for learning mathematics is German (LM1) and math classes later switch to a second language (French, LM2), systematic costs have been found for LM2 arithmetic problem solving despite an equal number of years training the LM2 [57]. A recent meta-analysis found an advantage for solving arithmetic and naming numbers (but not for magnitude comparison tasks) in the L1 [58].

These behavioural findings are confirmed by recent neuroimaging studies on arithmetic and number comparison. Recording electroencephalogram during a true or false judgment of simple multiplications, Salillas and Wicha [59] studied fluent Spanish-English bilingual adults

who had learned arithmetic in either English or Spanish, the LM+ respectively. The problems presented in the LM+ were solved faster and the corresponding event-related potentials showed a larger N400 response than for problems in LM-, independently of the language representing LM+ and LM-. Assessing two-digit number comparison in Spanish-Basque bilinguals [44], could even show that the numerical distance effect was modulated by the language of math acquisition. Functional magnetic resonance imaging studies revealed that the LM1 recruited more temporal regions, supposedly related to direct semantic retrieval, than the LM2 for simple additions. In turn, the LM2 recruited a network of regions indicating the need for more generic cognitive resources [60]. On the contrary, Cerda et al., [61] recently investigated Spanish-English bilingual children's performance in a multiplication verification task and observed similar ERP responses in both of their languages. Although the language of math acquisition largely impacts bilingual's arithmetic skills, under certain conditions, such as early learning stages, the bilingual brain might consequently reveal some flexibility.

These results tend to support the above-mentioned TCM stating that precise numbers are encoded in a language-dependent format [62]. They fit less with the classical view from bilingualism research stipulating that representations are independent of languages (*e.g.* [63, 64]) and that both bilingual's languages are active, even in situations when only one language is needed (*e.g.* [65, 66]). However, they agree with recent reports from the bilingualism literature that academic knowledge acquired in one language is retrieved more efficiently in the learning language compared to another language [54]. Further research on numerical cognition is consequently needed to fully understand this complex bilingual situation. Studying how language of math acquisition influences number transcoding is especially interesting since the cognitive mechanisms of transcoding are closely related to word retrieval. Yet, such studies are still missing, both in adults and in children.

## Present study

The present study aimed to better understand the interaction between language and numbers by investigating number transcoding of German-French bilinguals from four different age groups. We targeted 5th, 8th and 11th graders, as well as adults to assess performances before and after the switch in the language of mathematical learning implemented in 7th grade in the Luxembourgish education system.

In the Luxembourgish multilingual school system, pre-schools (3 to 5 y.o.) are in Luxembourgish, which is linguistically close to German. The teaching language in primary school (1st to 6th grade, 6 to 12 y.o.) is German, except for French lessons. From 7th grade to 10th grade, the majority of subjects are taught in German, except for mathematics and French lessons, which are taught in French. Then from 11th grade until the end of obligatory school, all topics are thought in French. This multilingual education system aims to render students highly proficient in both German and French. In sum, critically for number transcoding, the language for teaching and learning mathematics switches from German to French at 7th grade [67]. Hence students' first language of math acquisition (LM1) is German, while their second language of math learning (LM2) is French. Therefore allowing within-subject designs from a sample with the same educational background. Furthermore, German-French bilingualism is interesting concerning number word structures since both languages differ in *transparency of order* and their *transparency of power*. Previous studies on this specific population have reported language effects in magnitude comparison tasks, showing comparable compatibility effects to monolingual German [68]. While arithmetic problems were solved faster in German than in French [48], at least for complex additions [57]. Since arithmetic correlates with transcoding [17], we expected similar findings with transcoding tasks. The present study

investigates the question of the role language proficiency has on number transcoding. Using a cross-sectional design allowed us to study whether and how (a) number word *transparency of power* and (b) *language of math acquisition* influence two-digit number transcoding at different stages of bilingualism.

We explored the effect of *transparency of power* by comparing transcoding of French numbers above 70 ('70s '80s '90s) and below 60 ('30s '40s '50s), following respectively a base-20 and a base-10 structure. We expected that transcoding performances in both bilinguals' languages would improve with age. We hypothesized that independently of bilinguals' age, language would influence number processing such that non vigesimal French numbers (following a base-10 structure) would be transcoded better than vigesimal numbers over 70 (following a base-20 structure), revealing an effect of *transparency of power*.

For the impact of *language of math acquisition* we compared the performances in both of the bilinguals' languages in the four age groups. Transcoding requires to access and retrieve lexical information on numbers stored in long-term memory. Therefore, we predicted that transcoding would be better in German (LM1) than French (LM2). To capture the different facets of this retrieval process [69], we deployed two complementary transcoding tasks. In the *reading aloud task* participants had to name Arabic digits, while the *verbal-visual matching task* required the matching of spoken number words with the corresponding Arabic digit. Both tasks are assumed to tap into distinct retrieval mechanisms, *i.e.* free recall and recognition respectively. The two tasks might thus reveal a somewhat different result pattern such as more marked linguistic influences in the reading aloud free recall task than on the verbal-visual recognition task, as already observed in Van Rinsveld et al. [68].

## Method

### Participants

From the initial full sample of 125, we first excluded participants who reported having French as their mother tongue, as these participants might have acquired French number words outside the context of formal schooling. This led to the exclusion of 25 participants. One additional adult was removed because of an otherwise missing measure in one of the crossed factors. Therefore we excluded a total of 26 participants, leading to a final sample of N = 99. This final sample consisted of four age groups: 5th graders n = 26, age = 10.69 *(0.55)*, 8th graders n = 28, age = 13.46 *(0.56)*, 11th graders n = 25, age = 16.48 *(0.59)*, adults n = 20, age = 23.58 *(5.12)*. The language profiles varied among the participants, 66 reported Luxembourgish as mother tongue, 5 Portuguese, 4 German, 4 English, 4 Italian, 3 Polish, and the 13 remaining all reported a different mother tongue.

Despite the different language backgrounds, all participants were currently living in Luxembourg, where Luxembourgish, German and French are official languages that can be encountered in daily life. Importantly the language support for the formal acquisition of mathematics through school curricula is first in German (LM1) from 1st to 6th grade and switches to French (LM2) in 7th grade. Therefore young pupils start the school curriculum with different degrees of proficiency in French and German to progressively become proficient bilingual adults throughout their education. The four age groups of German-French bilinguals were therefore sampled at different ages of LM2 (French) acquisition: 5th graders being before the language switch in mathematics, while the three older groups have gradually increasing experience and familiarity with practicing mathematics in French.

All pupils were enrolled and tested in schools, while the adults were enrolled and tested at the university. The Review Panel of the University of Luxembourg revised and approved the

study. Informed written consent was obtained from all the children's parents and adult participants. Monetary compensation after study completion was given to both pupils and adults.

## Material and stimuli

Headsets connected to the Iolab USB Button Box were used to both (a) record the voice onsets of the reading aloud task and (b) play the auditory stimuli for the verbal-visual matching task. The experiments were run with Psyscope XB7 [70] on an Apple 13' MacBook. Reaction times were measured with the Iolab USB Button Box from the end of the auditory recording until button press. Stimuli for each task were 28 different two-digit numbers ranging from 31 to 98. These were further subdivided into two sets of 14 distinct stimuli each, one for the German and the other for the French blocks. The experiment was part of a larger study, which lasted 45 minutes, at the end of which the participants were compensated with a gift voucher. Ties (*e.g.* 44) and tens (*e.g.* 30) were excluded from the sets. The list of all the stimuli used can be found in Table 1 in S3 File. The order in which the sets were presented was randomized. In the reading aloud task, the Arabic digits were presented visually on a computer screen. They appeared in the centre of a white screen in black (Arial, font size 90) and remained visible until participant responses. For the verbal-visual matching task, German and French recordings of two female native German and French speakers were presented as auditory stimuli (as in [42]). In the verbal-visual matching task, there were four possible visual Arabic numbers: one target and three distractors, which positions varied randomly. The three distractor-stimuli consisted of: *unit distractors* in which one unit was randomly added or subtracted to the heard number (*e.g.* for 42 distractors were 43 or 41). *Ten distractors* in which a ten unit was randomly added or subtracted to the heard number (*e.g.* for 42 the distractor was 32 or 52). *Unit and ten distractors* in which 11 were randomly added or subtracted from the target number.

## Procedure

All participants were tested individually in a quiet room, children in the schools, and adults at the University of Luxembourg. The participants passed both the reading aloud and verbal-visual matching tasks first in German or French in a counter-balanced order. Both tasks started with a five-stimuli training followed by 14 test stimuli. Participants were instructed to respond as accurately and fast as possible.

In the reading aloud task after the participant named the visually presented number, the answers were written down by the experimenter, who started the following trial by button press. Reaction times were measured with voice key from stimuli screen presentation until first vocal onset. In the verbal-visual matching task, the auditory stimuli were first presented

**Table 1. Results of the reading aloud task's RT's linear mixed model.**

|  | num *df* | den *df* | *F* | *p-value* |
|---|---|---|---|---|
| **Age** | 3 | 92.38 | 16.43 | <0.001 |
| **Language** | 1 | 97.42 | 131.18 | <0.001 |
| **Number Size** | 1 | 22.10 | 39.02 | <0.001 |
| **Age x Language** | 3 | 89.32 | 16.35 | <0.001 |
| **Age x Number Size** | 3 | 1837.97 | 3.24 | 0.02 |
| **Language x Number Size** | 1 | 21.95 | 52.82 | <0.001 |
| **Age x Language x Number Size** | 3 | 1838.02 | 2.92 | 0.03 |

Note: degrees of freedom (df) calculated with Satterthwaite approximation; num: numerator, den: denominator.

via the headsets. Then, four numbers were visually presented on the screen. Reaction time recording started when the auditory stimulus presentation was completed and ended when one of the response buttons was pressed. Stimuli were presented with an inter-trial interval of 500 ms. At the end of the experiment the participants were compensated with a gift voucher.

### Design and statistical analyses

The transcoding of two-digit numbers was compared in a transversal developmental design by Linear Mixed Models (LMM) on the dependent variables reaction time and correct responses. To control for age related variabilities analogous analyses were conducted on *z-score* transformed RT, separately for each group. Furthermore, to control for word length (see [71]), we replicated the same analyses taking into account only numbers where the corresponding number words were constituted of four syllables (see Table 1 in S3 File for which stimuli were included in these analyses). Data were analysed with R [72] using the *afex* package [73], while graphs were drawn with *ggplot2* [74].

## Results

For the present analyses, we grouped the stimuli in two categories according to their size and number word structure in French: Small numbers ('30s, '40s and '50s) and Large numbers ('70s, '80s and '90s) in a factor referred to as "Number Size". That is, '60s numbers were excluded in order to create two equally sized groups clearly differing with respect to their French number word structure (*i.e.* decimal vs. vigesimal). However, before removing the '60s we applied common exclusion criteria to the data of both tasks. First, invalid trials of the reading aloud task including voice key recording errors, misspellings, inversions and confusions were removed, 4.29% of the initial sample. Second, to exclude aberrant reaction times (RT) we filtered longer than 5 seconds responses leading to the exclusion of 1.09% trials in the reading aloud task and 1.52% in verbal-visual matching task. Third, to exclude outlier responses, faster or slower than 3 standard deviations within each individual mean were removed, additionally, thus excluding 1.29% and 1.52% of the trials from the initial responses. Then we removed the '60s numbers (accounting for additional 13.71% and 14.07% respectively for both tasks). For reaction time analyses all incorrect responses were removed, accounting for 2.63% of the reading task and 5.99% of the matching task, corresponding to the total error rate. As suggested by an anonymous reviewer, we additionally removed the trials following an error which might be affected by post-error slowing in particular given the high accuracy rate [75], thus additionally accounting for 1.95% and 5.63% of the trials respectively.

### Analyses

Both tasks were analysed with Linear Mixed Models (LMM) in R using the mixed function from the *afex* package's *mixed* function [73], which relies on *lme4* [76]. Follow-up analyses were computed with *emmeans* [77]. When our initially designed full model failed to converge, we reduced the model complexity by gradually simplifying the random effect structure until convergence was reached.

The analyses aimed at testing for all age groups the two main hypotheses: (1) the effect of *transparency of power* in French and (2) the effect of *language of math acquisition*. First, the effect of *transparency of power* in French, corresponding to the cost for vigesimal numbers in the base-20 system was tested by comparing small numbers (*i.e.* '30s, '40s, and '50s) to large numbers (*i.e.* '70s, '80s, and '90s) in French only (see A in Fig 1). For this hypothesis, we predicted that both tasks would lead to worse performances, henceforth a cost, for large compared to small numbers. Developmentally, the cost might either be re-absorbed with increasing

**Fig 1. Venn diagram of the design rationale.** Design rationale of the comparisons on the different hypotheses of number word transparency (A) and age of math acquisition (B).

training using LM2 number words or remain stable across age groups. Additional control for the potential confounder of a number size effect was implemented by applying the same comparison within German (A.1 in Fig 1). Second, the hypothesis concerning the *language of math acquisition* was tested by comparing only small numbers (*i.e.* '30s, '40s, and '50s) in French and German (B in Fig 1). We predicted that both tasks would lead to a cost in French

compared to German. Developmentally, this cost could remain stable throughout adolescence and adulthood, or gradually diminish with age, potentially even resorbing in adults.

## Reading aloud task

**Reaction times.** The maximal linear mixed model was defined with main effects and interactions between the fixed between-group factor *Age* (levels: 5th, 8th, 11th grade and adults) and two fixed within-group factors: *Language* (German, French), and *Number Size* (Large, Small), all levels being defined as categorical. As random factors, we modelled individual differences (*i.e. Subject*) and item-related variability (*i.e. Item*). The maximum model was defined taking into account individual differences by using random slopes and intercepts *per Subject* for the interaction between *Language* and *Number Size*. Moreover, item-related variability was modelled using random intercepts and slopes *per Item* for the interaction between *Language* and *Age*. This led to the model with the following R syntax form (A0):

$$RT \sim 1 + Age * Language * Number\ Size + (1 + Language * Number\ Size|Subject) + (1 + Language * Age|Item). \tag{A0}$$

However, since the maximal model led to a singular fit (due to high correlations between the random parts of Number Size per Subject and Age per Item) we reduced the complexity of the model by removing those problematic random terms [78]. Therefore, the final model takes the following R syntax form (A):

$$RT \sim 1 + Age * Language * Number\ Size + (1 + Language|Subject) + (1 + Language|Item). \tag{A}$$

P-values and degrees of freedom were obtained with the Satterthwaite approximation method by comparing the full model against the model without the effect [73]. Follow-ups were calculated by comparing estimated marginal means (EMM) obtained with the emmeans package [77], and p-values were adjusted with the Bonferroni method. All contrasts were set to sum.

The model for the reading aloud task RTs resulted in three significant main effects and three two-way interactions (see Table 1). The main effect of Age was decomposed with follow-up analyses showing that the 5th graders (978*(555)* ms) were slower than the older groups, which had similar RTs (8th 788*(282)* ms, 11th 777*(246)* ms and adults 822*(320)* ms; standard deviations in parenthesis). Furthermore, the main effect of Language indicated overall slower naming in French than in German, and the main effect of Number Size indicated slower naming for bigger than smaller numbers. Significant two-way interactions were found between Age and Language, Age and Number Size, as well as Language and Number Size. Finally the three-way interaction was also significant, see Table 1 and Fig 2.

Follow-up analyses were performed on the model's estimated marginal means (EMMs) of the interaction's terms [see 79] with Satterthwaite estimation of degrees of freedom. Then EMM were compared with two-tailed pairwise tests with Bonferroni adjusted p-values. This confirmed the first hypothesis on *number word transparency* (*cf.* A in Fig 1), since in French the vigesimal '70s, '80s and '90s numbers were named slower than '30s, '40s and '50s numbers by all age groups: 282 ms slower for 5th graders, 219 ms for 8th, 188 ms or 11th and 236 ms for adults (all $t > 5.13$, $p < .001$). While as control (*cf.* A.1 in Fig 1), there was no significant difference in German (max difference 9 ms) between naming large or small numbers (all $t < 0.30$, $p = 1.0.$).

Secondly, the hypothesis on *language of math acquisition* was confirmed by the two youngest age groups (*cf.* B in Fig 1). Naming in French (LM2) compared to German (LM1) was

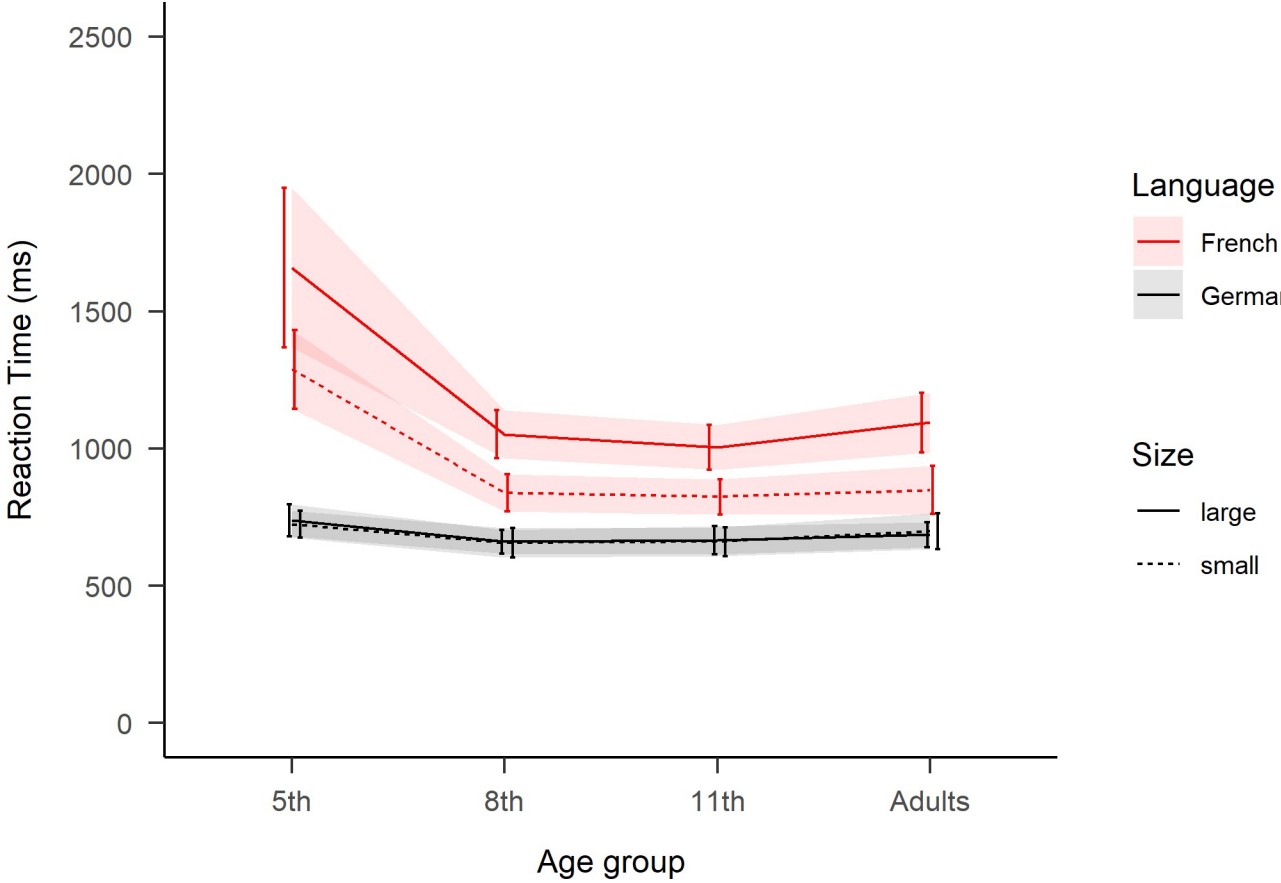

**Fig 2. Reaction times of the reading aloud task.** For each age group as a function of "Language" and "Number Size". Large numbers correspond to '70s, '80s and '90s, while small numbers correspond to '30s, '40s and '50s. Ribbons represent standard error.

significantly slower by 510 ms for 5th graders ($t(126.17) = 9.09$, $p<.001$) and by 177 ms for 8th graders ($t(117.4) = 2.83$, $p<.05$). Although positive, the difference was not significant for the two older age groups: 161 ms for 11th grades ($t(117.9) = 2.42$, $p = .10$) and 158 ms for adults ($t(117.17) = 1.94$, $p = .32$). To further establish the robustness of these results, we replicated the analyses using the same model (A), but on RTs transformed into z-scores per age-group (see Fig 3) and on a sub-sampled dataset including only four-syllable long number words (see S2 File).

The z-score transformation aimed to reduce the variability of RT among age groups and confirmed the main effects and the interaction between Language and Number Size, while the triple interaction was not significant anymore (see Table 1 in S2 File). Using the dataset with number words of 4 syllables (see Table 3 in S2 File) to control for the possible word length effect confounder ([see 71]) also replicated the main effects and the two-way interactions, but again the three-way interactions between Age, Language and Number Size was not significant anymore.

**Correct responses.**   Correct responses (CR) were analysed using a binomial approach and *p-values* estimated by the likelihood ratio test. Since applying the same model as for RT (see (A)) did not converge, we had to drop the random per-*Item* factor and the fixed factor *Age*. In sum the final model had the following syntax (B):

$$CR \sim Language * Size + (1|Sujet) \tag{B}$$

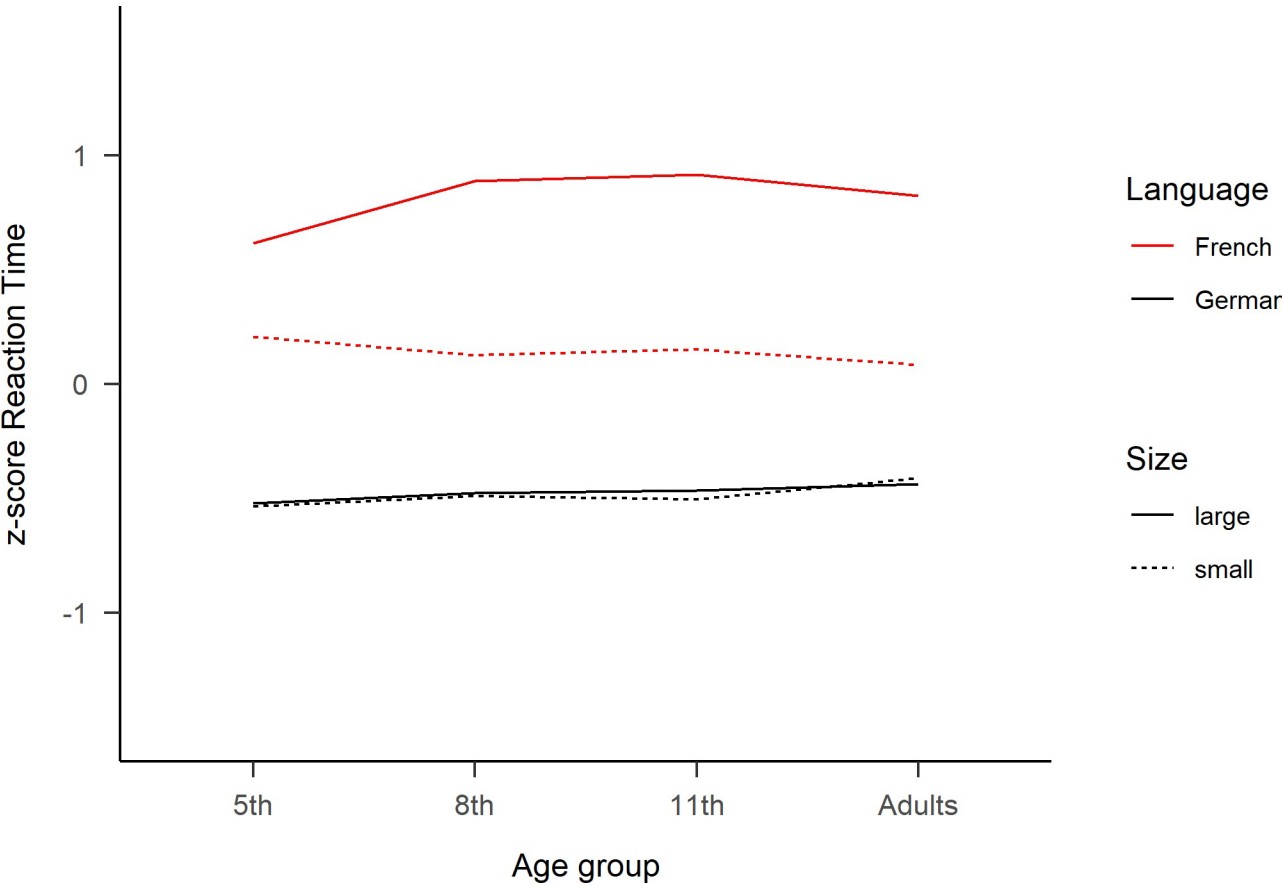

**Fig 3. Z-score reaction times of the reading aloud task.** Standardized reaction times for each age group as a function of "Language" and "Number Size". Large numbers correspond to '70s, '80s and '90s, while small numbers correspond to '30s, '40s and '50s.

The failure of convergence with the more complex model might be due to ceiling effects, as the task was very simple, particularly for older age groups, see Fig 4 and Table 2 in S1 File.

The binomial model (see (B) and Table 2) indicated a significant main effect of Language and in contradiction to the RT analyses, the main effect of Number Size was not significant. However, critically, like for RT, the interaction between Language and Number Size persisted with CR.

Follow-up analyses confirmed with CR the pattern observed with RT: in French, 4.8% more errors were made with '70s, '80s and '90s than '30s, '40s and '50s numbers ($z = -3.37$, $p<.01$),

**Table 3. Results of the verbal-visual matching task's RT's linear mixed model.**

|  | num *df* | den *df* | *F* | *p-value* |
|---|---|---|---|---|
| **Age** | 3 | 92.13 | 49.90 | < .001 |
| **Language** | 1 | 78.27 | 76.63 | < .001 |
| **Number Size** | 1 | 21.32 | 29.00 | < .001 |
| **Age x Language** | 3 | 83.25 | 14.64 | < .001 |
| **Age x Number Size** | 3 | 1731.94 | 15.06 | < .001 |
| **Language x Number Size** | 1 | 22.31 | 78.61 | < .001 |
| **Age x Language x Number Size** | 3 | 1730.32 | 11.74 | < .001 |

Note: degrees of freedom (df) calculated with Satterthwaite approximation, num = numerator, den = denominator

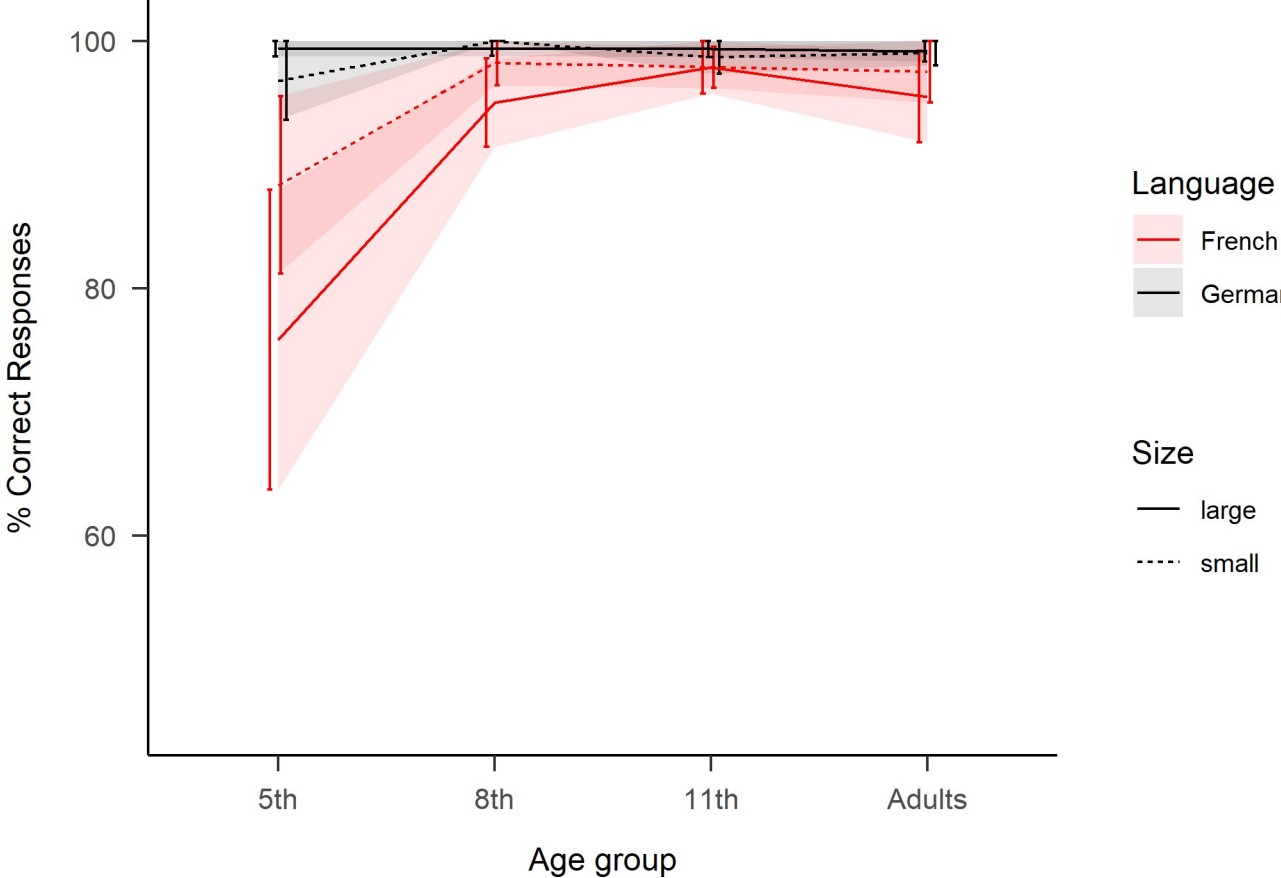

**Fig 4. Correct Responses in the reading aloud task.** Percent correct for each age group as a function of "Language" and "Number Size". Large numbers correspond to '70s, '80s and '90s, while small numbers correspond to '30s, '40s and '50s. Ribbons represent standard error.

indicating a cost for vigesimal number transcoding. The same contrast in German did not lead to any significant differences ($z = 0.63$, *n.s.*). Secondly, comparing '30s, '40s and '50s numbers in both languages revealed 2.6% more errors were made in French than in German ($z = -2.89$, $p<.05$), pointing towards a disadvantage for transcoding numbers in French (LM2).

In summary, the analyses on RT and CR confirmed both hypotheses which predicted effects of (1) *transparency of power* penalizing vigesimal number words and (2) *language of math acquisition* benefitting LM1 (*i.e.* German). The effect of *transparency of power* was robust across age groups since it could be replicated on z-score transformed data and by limiting the analyses to four-syllable long number words. The effect of *language of math acquisition* was found only in the two youngest age groups of the initial analyses, but persisted across age groups in the two additional analyses.

**Table 2. Results of the reading aloud task's CR's linear mixed model.**

|  | *df* | $\chi^2$ | *p-value* |
|---|---|---|---|
| **Language** | 1 | 47.31 | <0.001 |
| **Number Size** | 1 | 0.62 | 0.43 |
| **Language x Number Size** | 1 | 3.90 | 0.05 |

## Verbal-visual matching task

**Reaction times.** For coherence of interpretation and comparison with the previous verbal-visual matching task, the same model (A) and parameters for p-values, degrees of freedom and follow-up's were applied here on RT. That is the factors Language, Age and Number Size were modelled as fixed effects. Additionally the model included random factors to consider item-related variability as a function of language and random intercepts to account for individual differences per subject.

As a result, all main effects, two-way and three-way interaction were significant (all $F > 11.74$, $p < .001$, see Table 3). Overall a similar pattern than for the reading aloud task was found: the main effect of Age was driven by the slow responses measured in 5th graders (as shown by the follow-up analyses), the main effect of Language indicated slower responses in French than in German, and the effect of Number Size revealed slower responses for large (*i.e.* '70s, '80s and '90s) than small numbers (*i.e.* '30s, '40s and '50s) All two-way interactions were significant (see Table 3) and we found a three-way interaction between Age, Language and Number Size.

Follow-up analyses were applied comparing the estimated marginal means with paired comparisons on Satterthwaite corrected degrees of freedom and Bonferroni adjusted p-values. Follow-up analyses confirm the hypothesis relating to *transparency of power*, given a cost in French for '70s, '80s and '90s compared to '30s, '40s and '50s numbers for all age groups: 5th graders with a 682 ms cost ($t(97.97) = 10.84$, $p<.001$), 8th graders 339 ms cost ($t(53.31) = 5.27$, $p<.001$), 11th graders a 202 ms cost ($t(31.66) = 3.35$, $p<.01$), and adults with a 218 ms cost ($t(77.60) = 3.30$, $p<.01$). In contrast, the same comparison in German did not result in any significant differences (*all $t<1.05$, n.s.*, max difference = 46 ms). In sum, the cost for vigesimal numbers observed in the reading aloud task is replicated with the verbal-visual matching task for all age groups, see Fig 5.

However, in comparison to the reading aloud task, the hypothesis on *language of math acquisition* tested on small numbers (*i.e.* '30s, '40s and '50s; see B in Fig 1) was less supported, despite all differences in all four age-groups being positive. Indeed, performance advantages for German (LM1) were observed only in the youngest age group: while 5th graders were 682 ms slower in French than in German ($t(121.9) = 5.55$, $p<.001$), the same comparison between older age groups was not significant, 340 ms for 8th graders ($t(106.54) = 1.25$, $p = 1.0$), 202 ms for 11th graders ($t(107.1) = 1.34$, $p = 1.0$) and 218 ms for adults ($t(104.8) = 0.82$, $p = 1.0$). As for the previous reading aloud task, we compared with z-scores transformed and on a subsampled dataset to test the robustness of these results, see Fig 6.

The linear model applied on z-scores (see Fig 6) replicated the results with raw data described above, except, the three-way interaction with age, suggesting that the age differences observed with raw RTs, might be caused by differences between age-group variability (see Table 3 in S2 File). The analyses on the subset with four syllable-long number words replicated the effects mainly for the younger age groups, as the number size effect revealed that a cost associated with vigesimal numbers was present for 5th and 8th graders. Moreover, the effect of language in favour of the first language of math acquisition (LM1, *i.e.* German) was significant only in 5th graders (See Table 3 in S2 File).

**Correct responses.** For the matching task, the correct responses were analysed with the same method and model as for the reading aloud task, namely a binomial approach and likelihood ratio test, see Formula (B). Thus for the verbal-visual matching task, we modelled again the main effects and interactions between Language and Number Size and added random intercept per subject to consider individual differences.

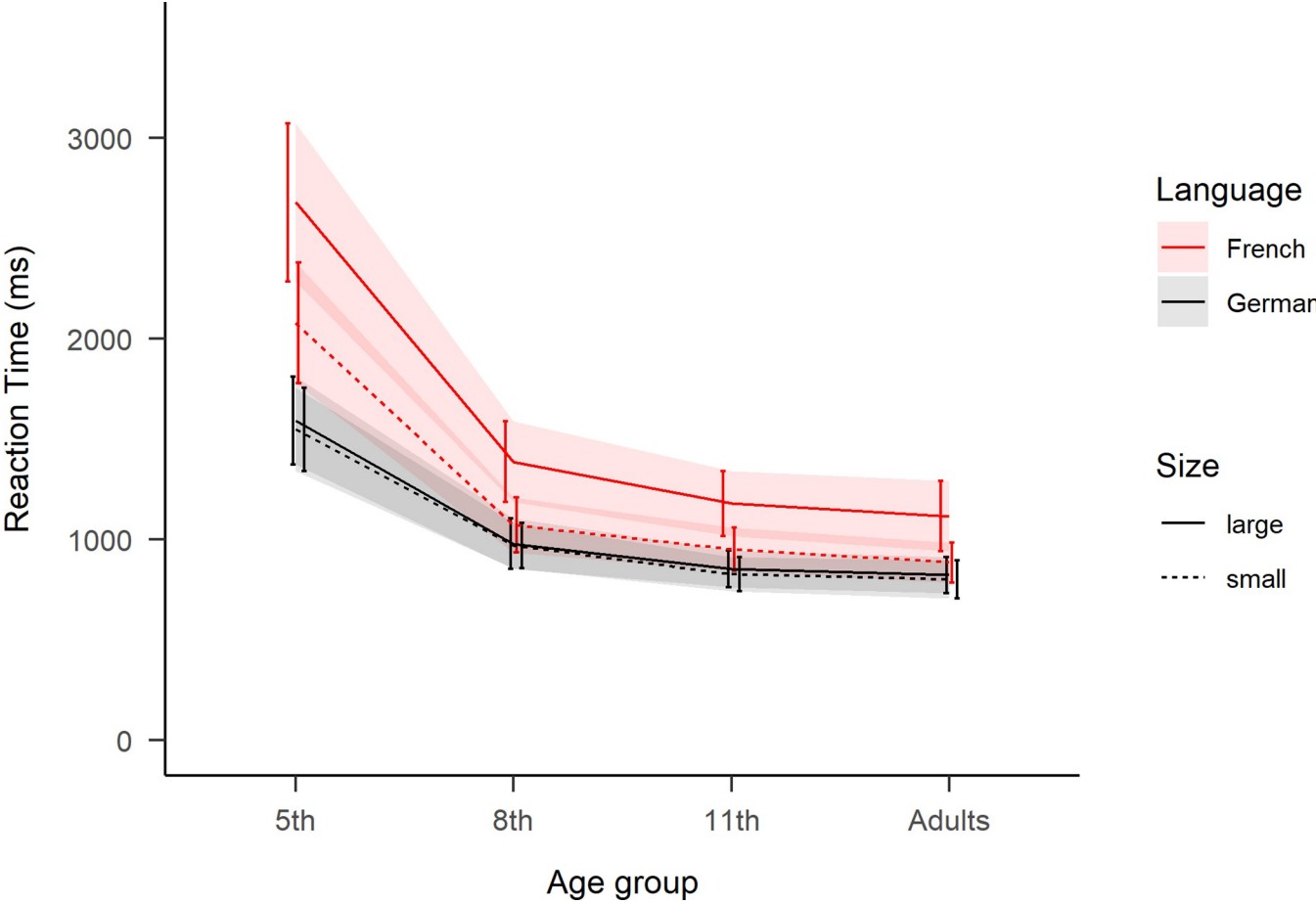

**Fig 5. Reaction times in the verbal-visual matching task.** For each age group as a function of "Language" and "Number Size". Large numbers correspond to '70s, '80s and '90s, while small numbers correspond to '30s, '40s and '50s. Ribbons represent standard error.

Similarly to the reading aloud task, the correct response rates showed a main effect of Language indicating in average 5.45% more errors were done in French than in German, see Table 4. The main effect of Number Size was marginally significant, potentially indicating more errors for large numbers. The two-way interaction between Language and Number Size was also significant again, see Fig 7.

Follow-up analyses indicate that 8.5% more errors were made when matching the French vigesimal '70s, '80s and '90s numbers than '30s, '40s and '50s numbers ($z = -4.75$, $p < .001$), as would be expected from the effect of transparency of power. No such difference was found in German ($z = 1.37$, $p = 1.0$). Secondly, comparing small numbers in both languages revealed no differences between French (LM2) and German (LM1) ($z = -0.46$, $p = 1.0$), thus again failing to reveal effects of language of math acquisition in the verbal-visual matching task.

In summary, for the verbal-visual matching task the (1) *transparency of power* hypothesis could be confirmed, as transcoding performances were overall lower for French '70s, '80s and '90s numbers having a vigesimal structure, stably across age groups. Concerning the hypothesis on (2) *language of math acquisition*, the advantage for transcoding in German (LM1) was only stable across age groups, when considering standardized RTs (see Fig 6).

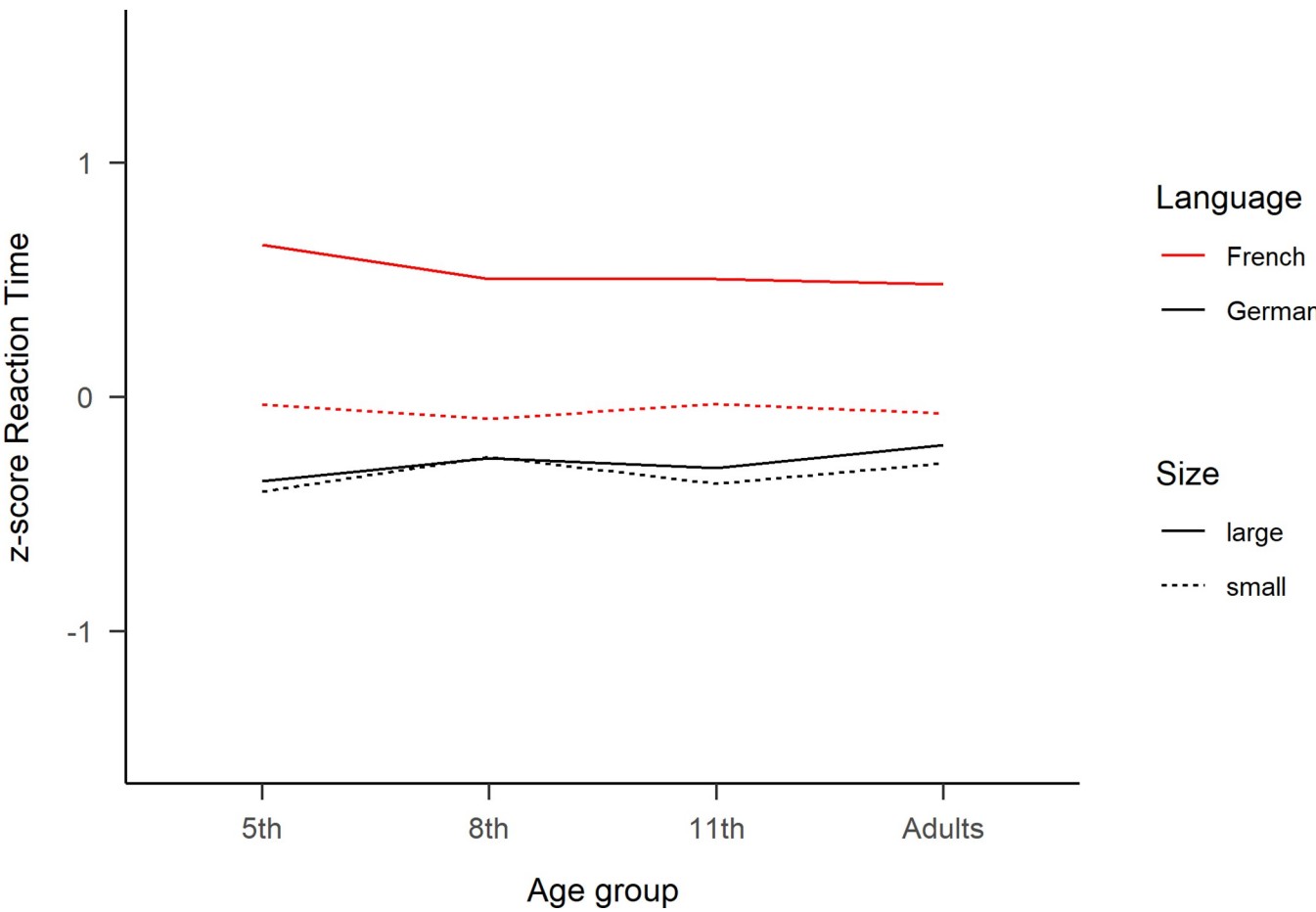

**Fig 6. Z-score reaction times of the verbal-visual matching task.** Standardized reaction times for each age group as a function of "Language" and "number Size". Large numbers correspond to '70s, '80s and '90s, while small numbers correspond to '30s, '40s and '50s.

## Discussion

Transcoding speed and accuracy of two-digit numbers were measured in bilingual participants during a reading numbers aloud and verbal-visual matching task, with a transversal developmental design. Participants were German-French bilinguals from four age groups consisting of 5th, 8th, 11th grades and adults. Since, the language of learning mathematics switches from German (LM1) to French (LM2) in 7th grade in the Luxembourg education system, participants from older age groups were increasingly trained with numbers and math in their LM2. For the reading aloud task, numbers were transcoded from visual to verbal formats. In the verbal-visual matching task numbers were transcoded from verbal to visual formats, with an additional target selection among three distractors. The main strength of the present study was the within-subject design which permits to examine language effects, while limiting external

**Table 4. Results of the verbal-visual matching task's CR's linear mixed model.**

|  | *df* | $\chi^2$ | *p-value* |
|---|---|---|---|
| **Language** | 1 | 22.31 | <0.001 |
| **Number Size** | 1 | 3.20 | 0.074 |
| **Language x Number Size** | 1 | 16.23 | <0.001 |

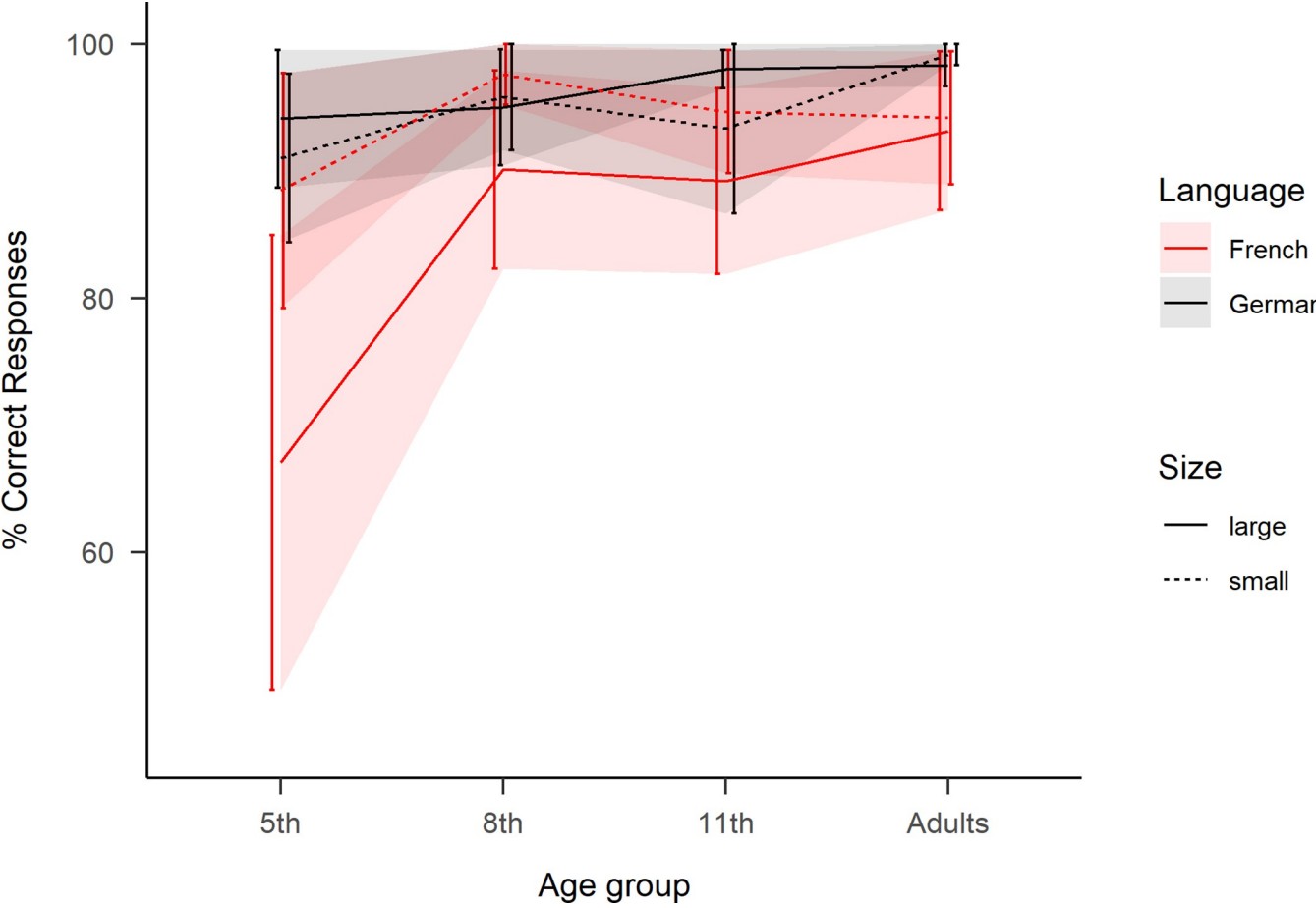

**Fig 7. Correct Responses in the reading aloud task.** Percent correct for each age group as a function of "Language" and "Number Size". Large numbers correspond to '70s, '80s and '90s, while small numbers correspond to '30s, '40s and '50s. Ribbons represent standard error.

influences such as inter-individual variability or differences in culture or education, typically present in cross-linguistic studies [45]. The use of the same task across all age groups allowed direct comparison of transcoding across different stages of bilingualism.

As expected, performance improved with age and education, such that participants from older age groups were faster and more accurate in transcoding two-digit numbers than younger one. Critically, independently of age and task, within French, there was a response time cost for transcoding '70s, '80s and '90s compared to '30s, '40s and '50s numbers. These results are in line with the concept of *transparency of power* since '70s, '80s and '90s number words in French follow a less transparent and hence more costly base-20 system than numbers under 60, which are characterized by a more transparent base-10 structure. This pattern was confirmed by analysing correct responses. However, age could not be included in the linear mixed model for accuracy, probably due to ceiling effects in the older age groups. In addition, and equally important, the results indicated a relative slow-down for LM2 (French) compared to LM1 (German) for numbers under 60 across ages in the reading aloud task. Correct responses were also affected by LM2 in the reading task, but for the same reasons as mentioned above we could not investigate age differences in accuracy. Since these effects were not consistently observed in the verbal-visual matching task, the hypothesis relating to *language of math acquisition* was mostly confirmed by the reading aloud task.

In sum, the results reveal two different linguistic effects on number transcoding: a cost for number words with weaker *transparency of power* (*i.e.* French vigesimal) and a cost for the later *language of mathematical learning* (*i.e.* the language which was not the language of math acquisition). In the following, we will discuss these effects sequentially.

## Transparency of power

In French, '70s, '80s and '90s numbers differ in their *transparency of power* [28] from '30s, '40s, '50s numbers, with the former being in base-20. Comparing these numbers across both tasks, we found a cost for French '70s, '80s and '90s compared to '30s, '40s and '50s numbers across age groups and task. We interpret this as an effect of *transparency of power* that is independent of participant's age and bilingual proficiency level. An advantage for larger compared to smaller numbers could also be explained by a number size effect [80]. However, this interpretation can be excluded, because the same comparison in German did not reveal any differences which, under the untested assumption that 5th graders do not have mature magnitude representations, would speak against the prediction of semantic models. The effect was not explained by word length either (see [71]), since it was replicated with all age groups except adults when analysing only number words having the same length of four syllables. Comparable results were observed in terms of correct responses, although the different age groups could not be compared due to insufficient model fit. Since the number of correct responses displayed ceiling effects for older participants in both tasks and languages, correct responses might lack sensitivity and is therefore not an ideal dependent variable for measuring language differences in the present study [29].

The cost for less transparent number words structures is in line with previous results. Investigating participants from the same German-French bilingual population than the present study, Van Rinsveld and colleagues reported that arithmetical problems with numbers over 70 in French led to slower, less accurate solutions than in German both in visual and auditory modes [57]. Furthermore, our results replicate and extend previous transcoding studies, finding more errors in France-French (less transparent) than Wallonia-French (more transparent) speaking seven years old children [41]. Costs for base-20 numbers were also found for number reading and matching tasks when comparing 5th graders speaking French or English (*i.e.* base-10 numbers) [42]. The present study extends these findings by comparing different age groups of increasingly proficient German-French bilinguals performing the same transcoding tasks. Notably, the cost was found for both reading aloud and number matching tasks that rely on transcoding from visual to verbal processes and *vice-versa*. The results from the z-scores standardization suggest that the cost explained by a difference in *transparency of power* was generally and stably persisted from 5th grade up to adulthood. In other words the cost maintains across age groups, tasks and the associated degree of language proficiency. In the following, we give two possible accounts for the cost for transcoding French base-20 numbers.

From the perspective of cognitive models presented in the introduction, the cost for *transparency of power* in French would fit with Power and Dal Martello's semantic model if rules reflecting the vigesimal form are added, since more rules would mean slower production [9]. Taking into account development, the asemantic ADAPT model (A Developmental Asemantic and Procedural model for Transcoding [6]), would partly fit with our results. ADAPT was proposed for number dictation (*i.e.* verbal to visual Arabic transcoding), hence similar processes could have taken place in the verbal to visual matching and reading aloud tasks (*i.e.* visual Arabic to verbal) of the present study design. In ADAPT, French '70s, '80s and '90s require more rules and therefore also more processing time. However, the model further proposes a lexicalization through repetitions over development, *i.e.* leading to faster transcoding of '70s, '80s and

'90s. Yet, when standardizing age groups variability through z-score transformations, we noted that numbers above 70 were similarly slower to transcode in all age groups, arguing against a lexicalization with training. Also, in answer to an interesting comment in the review process, we conducted additional analyses on reaction times (see S4 File), by adding all decades as levels for the Number Size factor (hence 8 levels from '30s to '90s). Those additional analyses confirm that for both tasks in French, reaction times become slower from the '70s decade onwards. The non-lexicalization of '70s, '80s and '90s vigesimal numbers in French might interact with their late formal acquisition (*i.e.* LM2 acquired at 7th grade, around 11–12 years old). Finally, the cost for '70s, '80s and '90s numbers also fits with Dotan and Freidman's recent transcoding model which suggests that reading '70s and '90s French number words requires additional irregular rules. When reading 75 for example, a number word frame would be structured by a decade and a teen instead of ten, then the ten frame filling would be changed from "7" to "6" (the filled frame would be: [6:tens] [15:teens] to give "*soixante-quinze*", literally "*six*ty-*fifteen*") [12].

Speculatively, it could be argued that for French morpho-syntactically complex number words that are not lexicalized, multiple additional morphemes need to be retrieved (see *i.e.* [81]). This could be the origin of the cognitive slow-down as it requires to retrieve more lexical morphemes (*i.e.* four for 97: "quatre", "vingt", "dix", "sept"). Eventually, these morphemes, which are made of other numbers might interfere among each other and slow down RTs (*i.e.* proactive interference [82, 83]).

Please note that all three models would also generally predict slower responses for German number words above 12 due to the additional inversion rule in German. Such effects of *transparency of order* have been found in previous studies [17, 29, 33], but it is possible these were masked in the present study by the effect of language of math acquisition described in the following section.

To sum up, the effect of *transparency of power* was confirmed by costs for French '70s, '80s and '90s vigesimal numbers, independently from age or task. The origin of this cost could be explained by the non-automatized, hence not lexicalized, transcoding process for these non-transparent numbers in French. However, since French is the second language of mathematical acquisition (LM2) of the current sample, it remains an empirical question whether this interpretation can be generalized to early learners of French.

### Language of mathematical acquisition: LM1 vs. LM2

To assess whether and how *language of math acquisition* impacted transcoding in the different age groups, we compared performances in German (LM1) and French (LM2). To avoid the potentially confounding effect of *transparency of power* described above, only '30s, '40s and '50s in both languages were compared. We can also exclude that differences *in transparency of order* between German and French explain LM2 costs since this would have meant the opposite effect, that is slower responses in German (LM1, having an inverted number word structure, *e.g.* [17, 31–33]), rather than in French (LM2, having a non-inverted number word structure). In line with our expectations, we observed costs during the reading aloud task in LM2 (French) compared to the LM1 (German); these costs were observed in the two youngest age groups in the analyses on RT, but they appeared at all ages when considering standardized response speed and four-syllable words. In the verbal-visual matching task, costs were consistently visible only in the analyses of standardized response times.

Our findings are in line with studies reporting qualitatively different arithmetic performance with LM+ compared to LM- in bilingual Filipino-English and Spanish-Basque participants, even if LM- corresponds to participants' mother language [56, 59]. Finally, they also

match and extend the finding that solving addition problems was slower in LM2 than LM1 in participants coming from the same education system than in the present study [57].

Interestingly, while the *language of math acquisition* impacted the number reading task, this was considerably less the case in the verbal-visual matching task. Lexically, the different pattern of results for the LM2 could be explained by different memory retrieval mechanisms [69] since the number reading task can be considered as a form of free recall while the matching task is more similar to a familiarity judgment. During free recall all possible number words can interfere with the retrieval of the correct number word, entailing a kind of lexical competition among the different verbal codes causing a cost for the less dominant language (LM2). In contrast, visual familiarity of the target number might underlie participants' responses during matching, weakening the role of language code(s) activation during this task. Similar task differences were indeed reported by Vander Beken and Brysbaert [69] in a study investigating the recall of text in university students' first and second languages. An additional explanation might rely on the nature of the stimuli. In the matching task, the number word input is already language-specific and therefore might be less susceptible to between-language lexical competition. While for the number reading task the identical visual Arabic digits might lead to a lexical competition between the LM1 and LM2.

To interpret the LM2 cost different theoretical perspectives can be taken. A possible interpretation can be derived from the ADAPT model of number transcoding [6]. LM1 (German) could be lexicalized (*i.e.* directly retrieved from long-term memory), while the LM2 (French), could rely on slower procedural rules, even for numbers under 60. In line with this view, weaker fMRI temporal lobe activation was observed when solving simple additions in LM2, proposedly reflecting less verbal retrieval than for LM1 additions [60]. Furthermore, since in ADAPT algorithmic rules are enacted by the short-term memory, it could potentially impact its capacity by using more resources [84, 85] and in turn explain parts of the LM2 costs observed in the same bilingual population for exact arithmetic [57]. It is worthwhile noticing that this disadvantage for storing and accessing LM2 numbers in verbal short-term memory might also impact the results obtained in neuro-developmental diagnostic tests. Indeed, using LM2 numbers for tests such as the number span test of the Wechsler intelligence scale or for different tasks in dyscalculia batteries might hamper performance and lead to an underestimation of children's cognitive abilities.

The LM2 cost is also explainable from a psycho-linguistic perspective: here we present a syntactic and a lexical interpretation. Syntactically, the LM2 cost might result from an overgeneralization of the LM1 syntactic *inversion* rule (see for example [29]): transcoding number words in French would require inhibition of the inversion rule (over)learned from German. A lexical explanations can be found in more general bilingual models such as the bilingual interactive activation model (BIA+), predicting a response competition between both languages, with faster activation for the more frequently used language [86]. In this framework, the slower lexical retrieval of LM2 would also have detrimental impacts on short-term memory [71, 87] and hence arithmetic [88].

In summary, the performance was overall better for LM1 (German) compared to LM2 (French) '30s, '40s, '50s numbers, confirming the importance of *language of math acquisition* for two-digit number transcoding in bilinguals. These effects were stronger in the number reading task than in the verbal-visual matching task and could be interpreted from bilingual lexical or syntactic perspectives.

A limitation of the present study is that it did not allow to disentangle the role played by the *language of math acquisition* from the familiarity with number words in the two languages. Since the language used for early learning (LM1) is probably also the one used more frequently, both factors might be confounded. Yet, both processes are supposed to rely on

different neuronal substrates [15]. It is, however, noteworthy that math education extended for one year longer in LM2 (7 years of secondary education) than in LM1 (6 years of primary education), which might help to balance the frequencies in both languages.

Concerning the effect of *transparency of power* which we observed with '70s, '80s and '90s French numbers, we cannot exclude that part of the effect comes from the mixed nature of French number words. Since French contains both base-10 and base-20 numbers, it remains indeed to be determined whether similar effects would be observed with a language containing exclusively base-20 numbers (see for example [89]). Likewise, further interpretation of the LM2 costs would benefit from a study assessing the effects of LM1 on transcoding with a language having simpler number word structures (*e.g.* English or Asian languages). Finally, it is still debated whether transcoding requires access to semantics or not (see [6]). Further studies should investigate LM2 costs during number processing tasks systematically activating number semantics (*e.g.* in masked priming designs as in [90]).

Taken together the present results confirm and extend the interactions between language and number processing observed in more complex tasks such as arithmetic problem solving, corroborating the role of language in numerical cognition.

## Conclusion

The *transparency of power* consistently affected transcoding in bilinguals from the four age groups, ranging from grade 5 to adulthood. Base-20 number words in French were transcoded slower than base-10 words and this effect could not be explained by a semantic number size effect or the length of number words. Furthermore, *language of math acquisition* affected the speed with which Arabic numbers were named, such that transcoding in LM2 entailed a cost across age groups. This allows us to conclude that linguistic factors influence basic numerical tasks such as transcoding until adulthood.

## Supporting information

**S1 File. Mean reaction times and correct responses of both tasks across the factors age group, language and size.**
(DOCX)

**S2 File. Linear mixed models on z-score transformed data and on four syllables long sub-sampled dataset of both tasks.**
(DOCX)

**S3 File. Stimuli and number of syllables used in the experiment.** Number of syllables for each number word in German and in French as used to control for word length.
(DOCX)

**S4 File. Supplementary analyses.** Analyses carried on each decades. Linear mixed models for both tasks' reaction times, but replacing the Number Size factor's level with decades from '30s until '90s (instead of small and large).
(DOCX)

## Acknowledgments

The funders had no role in study design, data collection and analysis, decision to publish, or preparation of the manuscript. We are grateful to the staff in schools for opening their doors to our team. We also thank the subjects of all age groups for their participation.

## Author Contributions

**Conceptualization:** Amandine van Rinsveld, Christine Schiltz.

**Data curation:** Amandine van Rinsveld, Alexandre Poncin.

**Formal analysis:** Rémy Lachelin.

**Writing – original draft:** Rémy Lachelin, Christine Schiltz.

**Writing – review & editing:** Christine Schiltz.

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
