## [Decision Letter · Decision Letter 0]

2 Aug 2021

PONE-D-21-18244

Number Transcoding in Bilinguals - A Transversal Developmental Study.

PLOS ONE

Dear Dr. Lachelin,

Thank you for submitting your manuscript to PLOS ONE. I have sent it to two expert reviewers and have now received their comments back. As you can see at the bottom of this email, both reviewers think that the topic is interesting and noted the unique sample investigated. I concur with these points. However, the reviewers also raise several major comments. Notably, reviewer #1 is concerned with the theoretical framework of the study, which they do not find appropriate. The reviewer also has a number of suggestions for improvement. Reviewer #2 has major concerns regarding the analytic strategy and therefore suggests a reworking of the analyses. The reviewers have been very thorough and I will not reiterate here their specific points. Because I still see merit in your manuscript, I invite you to submit a revised version of the manuscript that addresses all the points raised during the review process. Note, however, that the revisions suggested by the experts are substantial and that I will send the manuscript back to the original reviewer. Please know that submission of a revision does not guarantee that your manuscript will be accepted.

We look forward to receiving your revised manuscript.

Kind regards,

Jérôme Prado

Academic Editor

PLOS ONE

Journal Requirements:

The first author is supported by the Luxembourg National Research Fund (FNR) (PRIDE/15/10921377)

5. Please ensure that you refer to Figure 2 and 3 in your text as, if accepted, production will need this reference to link the reader to the figure.

Reviewers' comments:

Reviewer #1: The manuscript describes two number transcoding tasks – reading aloud and matching – done in two languages (German, French), in different age groups, in bilingual individuals. The study found two main effects: (1) An advantage of German (L1) over French (L2); and (2) Difficulty in French numbers above 60 relative to smaller numbers, presumably due to their vigesimal structure (e.g. 74 = soixante-quatorze, sixty-fourteen). These results are in good agreement with several previous investigations of verbal numbers and with common number processing models.

The study design is good and the analyses are generally appropriate. I would therefore like to see this study published in PLOS ONE. My main concern is that in my opinion, the manuscript itself should be considerably improved. It seems to me that some serious fundamental rewriting is needed here, after which I will gladly recommend the article for publication.

Major comments

1. The theoretical framing is poor, unfocused and sometimes incorrect.

I do not refer to the important/impact of the study, because this is not a publication criterion in PLOS ONE. However, irrespectively of the question impact, the manuscript should still interpret the result in terms of relevant theory.

- There is no reference to relevant cognitive models of number processing, although there are quite a few of those (McCloskey; Dehaene & Cohen; Butterworth & Cipolotti; Dotan & Friedmann; in a way, also Huber, Nuerk, & Willmes; and more).

- The topic of this article is number transcoding, but for some reasons the authors chose to dedicate much of the theoretical discussions to other issues, which are only vaguely related – ANS, arithmetic, and language. Perhaps the two most extreme examples for this is that (1) the first page of the manuscript (!!) is simply irrelevant – except the last few lines, it talks mostly about ANS and brain regions, two topics that are only vaguely related to the article’s main point. (2) The first sentence in the abstract is simply incorrect. It says that the study examined the “analogic, verbal, and visual symbolic codes”, but the “analogic code” (ANS) was not really examined here.

- Similarly, the selection of cited literature seems a bit odd throughout most of the manuscript (but in the Discussion it is good). There is a lot of published work on transcoding, but the article ignores much of this literature and cites literature from other domains, which is less relevant or hardly relevant. There is no citation at all of many relevant works by researchers who did a lot of theoretical work on transcoding, e.g., Michael McCloskey, Carlo Semenza, Margaret Delzaer, Marie-Pascal Noël, Xavier Seron, Laurent Cohen and more – including works that specifically addressed the base-10 vs base-20 issue. Even when talking about arithmetic, there are additional studies that specifically examined the role of the verbal structure of numbers and could be cited (e.g., Campbell & Clark, 1992; Gonzalez & Kolers, 1982; Noël & Seron, 1997).

On few occasions, the manuscript cited general language literature as a reference to number-specific phenomena, e.g. references [49] [52]. I would prefer seeing citation of studies more specific to numbers.

2. The article’s main theoretical question requires the comparison of base-10 versus base-20 verbal numbers. To operationalize this, the authors took the French numbers above 60 as reflecting the base-20 system. This does not seem justified. In French, the syntactic structure of verbal numbers in the 60’s and 80’s is congruent with both systems (b10 and b20). Only numbers in the 70’s and 90’s are clear examples for base-20 syntax.

I would guess that if you focus only on the 70’s and 90’s, your results will actually improve. Also, perhaps you may even be able to show that the 80’s are as easy as the smaller numbers – this could be a nice control for an artefact of number size (as opposed to syntactic structure, the effect you’re actually interested in).

The manuscript seems to assume that numbers in the 80’s are cognitively represented as “20 times 4 plus X”, e.g., 83=20x4+3. Presumably, the reason is that the French word for 80 is “quatre-vingt”, “four-twenty” (although this is not explained explicitly). Although this argument has been made in past studies, it is not convincing at all – “quatre-vingt” is very likely to be one lexicalized value, even if its etymology is 20x4. If there is empirical evidence showing that 80 is represented in a decomposed manner in the relevant population, please cite it. If not, please remove this claim, or at least present it as an unsupported hypothesis.

Note that even if you view 80 as decomposed 20x4, this still does not mean that the numbers 81-89 are in base-20, because the issue of 80 vs. 20x4 concerns only the tens word, not the way it is composed with the ones word.

3. The theoretical interpretation in the discussion is not convincing and should be improved.

(a) The theoretical interpretation in lines 560-575 seems incorrect in several ways.

- Lines 560-562: this is not in good agreement with commonly-accepted cognitive models of number production (e.g. McCloskey et al 1986). Phonologically, both “quatre-ving” and “dix-sept” are probably single lexical entries so these are two words, not four. I am not sure this was tested in French, but certainly you cannot claim the opposite as if it was a fact.

- Even if we assume that French 97 requires activating 4 lexical entries, it is still not clear why the cost should be attributed specifically to interference (and not, for example, to the effort arising from the need to retrieve more words).

- Lines 566-568, reference [58]: the example is completely irrelevant, and in this context, even a bit misleading. Of course, some cognitive interference effects exist. But it doesn’t mean that this is the case for numbers too. Especially since numerous studies show fundamental dissociations between the production mechanisms of number words and non-number words.

(b) Lines 576-598, 623-629: it seems strange to interpret your findings according to the ADAPT model, because ADAPT is a model for number writing (verbal-to-arabic transcoding), and the task here is reading (arabic-to-verbal transcoding). You can use models of number reading instead.

I don’t think this should disturb you because you don’t really rely on any particularity of the ADAPT model, you merely rely on the well-accepted, widely-supported distinction between lexical processing and syntactic processing in numbers. What your data shows is probably a phenomenon that was shown in many previous studies: numbers with complicated syntactic structure are harder to process than syntactically-simple numbers, and syntactic ability develops with age. I think it would be more appropriate to frame your claim in these more-general terms, rather than present it as if the results support a particular cognitive model – I don’t think they do. If I were you, I would just drop the interpretation in lines 560-575, stick with the second interpretation, and present the results as evidence for a syntactic-complexity effect (a conceptual replication of several previous studies), and for the development of syntactic abilities with age. But needless to say, this is of course totally your decision.

Either way, I think that the manuscript will benefit from using the terminology “number syntax”. Although sometimes vague, this is standard terminology (certainly more standard than “transparency of power”) and it seems appropriate for your research question.

(c) Lines 613 seem like a vague mixture of several phenomena/issues. I found it more confusing than clarifying.

- Lines 613-614: your findings do not seem to be about language switching, but about familiarity with language. While I agree that language switching is probably relevant, I find this analogy more blurring than clarifying. At least discuss the difference between the two phenomena.

- 619-620: the encoding complex model makes very specific assumptions about the types of information coded in language-specific way (in particular arithmetic facts). The language effect you have found doesn’t seem to support these assumptions – it merely shows that language, and familiarity with language, matters. Frankly, this effect is a bit trivial, and pretty much agrees with any number-processing model I am aware of.

(d) Lines 630-638: your data doesn’t seem to support working-memory effects in any way. Certainly, difficulty in lexical retrieval does not necessarily imply a WM issue.

4. In the results section, some of the statistical analyses are described dryly, without any meaning/interpretation. Please add their theoretical implication. E.g. in p.15, the precise pattern of interactions with age seem to strengthen your theoretical claim about a progression of learning – why not say it explicitly right away? Especially in the summary paragraphs (lines 406-410, 495-503) I would expect that the text doesn’t just talk about factors and effects but also say something about the cognitive implications.

Even when the results are interpreted, this is sometimes done separately from the result. E.g. the analyses of both tasks start by saying that all double interactions were significant, but these interactions are explained only later in the text. Please fix: unite each result with its interpretation.

The comparison of the two tasks doesn’t seem to have any theoretical relevance and should probably be deleted (or at least pushed down to somewhere less central in the text). As far as I understand, there is no theoretical reason to assume that the performance in the two tasks would be similar, and we don’t learn any insights from this comparison.

5. Some of the terminology used in the manuscript is imprecise, non-standard or incorrect.

- “Language of math acquisition” is a confusing and imprecise term. You did not manipulate the language of acquisition (=learning), you manipulated the language in which the experiment was done. But I certainly agree that the reason for this effect is likely to be LM1 vs. LM2.

- The first task is neither “a production task” nor “number naming”, it’s reading aloud (it includes both visual input and verbal production). The second task would be better described not as a recognition task, but as a matching task.

- “Transparency of power” is not a standard term in the number transcoding literature, so please explain it before using it for the first time. Perhaps avoid using it in the abstract.

- When saying the performance is better in condition X, say better than what (e.g. p. 17 line 407)

- The inversion in German is not between tens and units, it is between tens and ones (“units” = digits, “ones” = words).

- “Verbal” and “visual symbolic” codes – the verbal code is symbolic too.

- Lines 579-581: there is a confusion here between two meanings of the term “semantic”. In the ADAPT model (and in other models, e.g. McCloskey’s), “semantic” often refers to a central abstract representation of the number. The number size effect refers to semantics as the quantity associated with the number via the ANS. These are two completely different things.

- “Analogic code” is a correct/acceptable term for ANS, but it is not standard. “Approximate” would be a more standard terminology

6. I would recommend using linear mixed model instead of ANOVA. It is more reliable, and also might improve your statistical results.

7. The quality of writing should be improved.

- The manuscript should undergo an English language review.

- The text is too long, please write more concisely. Just to give a few examples:

(1) The issue of French>60 examines transparency of power is explained too many time.

(2) The first paragraph in the results section can be cut into a short sentence. But there are many more.

(3) The text describing the analyses with (p. 16) and without (p.15) z-scored are redundant and tiring. You can write the text once, and say you got essentially the same results with and without z-scores (with results of both ANOVAs in a table). Similarly, fig. 6 is redundant and can be removed. Same issue for recognition task.

But I emphasize that these are just examples, please write the whole text more concisely.

-

Minor comments

1. The statistical analyses are described in detail, which is excellent, but because there are quite a few of them, I would recommend moving them to the results section. I completely lost track of the analyses while reading, and it was very hard to relate specific results to specific analyses.

2. Fig. 3 – the Y axis label says RT instead of accuracy

3. Fig. 4 – instead of o60 and u60, perhaps write something clearer, e.g. >60, <60

4. Lines 344/355 – I think there is an unintended paragraph break here.

5. There are quite a lot of ANOVA results. Consider putting them all in table/s instead of in the text – it will cut the text shorter and could make it easier to find the relevant stats.

6. Lines 344-351: to show that the German and French trends are different with respect to >60 vs <60, here would be the correct place to talk about the Size x Language interaction.

7. Line 390: how many 4-syllable numbers are there? Also, as far as I know, the relevant measure for word length is typically assumed to be the number of phonemes, not the number of syllables (Nickels & Howard, 2004, Cognitive Neuropsychology).

8. The particularity of the German language (tens-ones inversion) should be mentioned earlier on in the text.

9. All figures: measurement units “millisec” should be written only on axis label, not in the figure caption.

10. The introduction and discussion mention a transparency-of-power analysis for numbers above 70 but I think I didn’t see it in the results section, did I miss it?

11. Line 556: as far as I can see this is incorrect - the base-20 cost was not independent of age, it decreased with age (even if it existed to some extent in all age groups).

12. The reference to ADAPT in line 577 is incorrect, [60] instead of [64]. Please double check all references.

13. Lines 180-182: this claim does not seem correct – at least, this is not the common assumption for how number transcoding works. Certainly, you cannot base this claim on reference [52] not from the number domain.

Reviewer #2: In this study, the French and German transcoding abilities of children in Luxembourg, who are taught in German in primary school and in French in secondary school, were measured in different age groups. The Luxembourgish multilingual background makes this an interesting group to study. However, the study has several shortcomings. First, it is difficult to draw any definite conclusions without a comparison group, because language characteristics, age and educational experience are all confounded with each other.

The difficulties of French are elaborated upon in much detail, but the German system is barely discussed and it even looks like a section on this has been removed (with the appropriate references still present in the reference list), giving the impression that German is much easier than French while in reality the picture is more nuanced than that.

But most problematic, the analyses are messy, both in the terminology that was used (‘groups of contrasts’), in the analysis plan and presentation of the results, and in fact I’d argue that ideally these data require a mixed effects model, not an ANOVA. This means that a reanalysis is necessary and at present it isn’t possible to review the results in detail. Perhaps with a thorough reanalysis that also includes monolingual control groups it would be possible to draw conclusions but for now I have to recommend to reject the paper.

I explain all this (and some more comments) in more detail below.

line 44-46: ‘Since multilinguals are outnumbering monolinguals across the globe [12],the question of how numbers are processed and transcoded in different languages is of crucial importance.’ Why? Undeniably there are a lot of multilinguals but that fact doesn’t necessarily make this particular research question crucially important. Please either remove or explain the relevance of the research question for all those multilinguals. This is a bit related to the discussion, see also later. If this is fundamental study that is perfectly fine, but don't pretend it also has immediate applications or benefits.

line 75-78: This section would benefit from a clearer explanation of the different counting systems in German and French for those unfamiliar with those languages. Now the base-10 vs 20 is highlighted, but first I’d argue that the base 20 system in French is only visible for the 80s and 90s and not below (or else 62 would be 3x20+2 which is not the case). I’d say that the true base 20 starts at 80, as the 60’s follow the regular base 10 system and the 70’s are neither (as they are 60+10, 60+11 etc). And then there are the irregular number words for 11-16.

Line 77 ‘only eighty decades are vigesimal’ probably supposed to mean ‘only the decade 80’?

Note by the way that also languages like English and French are not as transparent as Chinese in the base 10 system. If they were, a number like 60 in French would be ‘six dix’ (six ten) instead of ‘soixante’.

And please also elaborate on differences between the French and German counting system. The explanation of the base 10 vs 20 system is very elaborate but the decade unit inversion in German is barely mentioned, so short that I doubt whether a reader who is unfamiliar with languages that have this inversion even understands what it is. And what where the results from references [18] and [51]? And I wanted to recommend some more papers on this phenomenon but I already found a few in the reference list without referencing to them in the text [65-67]. Was this part deliberately taken away to give the simplified impression that French numbers are always more difficult than German numbers? Decade unit inversion makes transcoding more difficult. This could become especially visible in the

The ADAPT model of number processing is mentioned in the discussion but perhaps this could be already done in the introduction, with an integrated approach discussing the difficulties of all languages.

p. 4 line 82. ‘Analogously’ but what follows is not analogously, it is in contrast.

The present study – Please, for a complete picture, describe what language children in Luxembourg speak before going to school. Luxembourgish, German, French? Many children speak Luxembourgish, that is highly similar to German, right? This may especially be relevant if parents stimulate number activities. I see that it follows in the methods section- a lot of different languages and French was excluded. Please elaborate on these choices and potential effects of this choice. In fact, the children with French as native language may form a very interesting group (although a bit small in size if there is age variation) to disentangle L1 from ML1 influences.

p.8 I understand how numbers below 60 are a purer indication of an effect of language of math acquisition (although confounded with decade-unit inversion) but this doesn’t stop at 60, sure numbers above 60 are also affected by this, not just by an effect of transparency of power.

70 makes much more sense than 60 though, because in French only 70 and above cannot be transcribed by merely writing the numbers that you hear, the numbers in the 60s can. It is stated that the analyses were also performed with the cut-off at 70 but were there any relevant differences?

Furthermore, the study makes an attempt to identify effects related to the language structure and effects related to language of acquisition but the two are confounded while a control group is lacking. Any effect in favor of German can be due to L1/math language or to language structure. It would be helpful to compare the results to monolingual French and German control groups (and as mentioned above, the native French speaking children in Luxembourg), not just to mention this in the discussion but actually add those groups.

Design and statistical analyses: it’s not explained what was done with separate stimuli. How were aggregate scores determined? Why not run a mixed effects model instead of ANOVA? The afex package that was used can handle that as well and you can estimate the effects more precisely because it can also handle random slopes across participants (and items). Not taking that into account can lead to alpha inflation. Similarly for accuracy, which is bounded by 0% and 100%, ANOVA typically gives the wrong results, often spurious significant results but p-values can also be too low. Especially when values are close to the lower or upper boundary (which is the case with these very high accuracy rates close to ceiling) because ANOVA requires unbounded data. You need a model for proportional data such as a binomial mixed effects model in that case. See for instance Jaeger, T. F. (2008). Categorical data analysis: Away from ANOVAs (transformation or not) and towards logit mixed models. Journal of memory and language, 59(4), 434-446.

But also for the RT data a huge benefit of a mixed effects model is that you can enter all stimuli separately and you don’t need to calculate averages per condition, thus making maximal use of the available information.

Do note though that mixed effects models aren’t easy, and should be carried out appropriately with sufficient knowledge, or else the results may be off as well.

Analysis plan: explain the dependent variable.

Why speak of ‘groups of contrasts’ instead of just ‘contrast’? Moreover, I’m a bit confused about the terminology here because a contrast is applied to compare the levels of a single variable in a pre-specified way. For example, if you have 4 groups, comparing group A and B to group C and D, and then comparing group C to group D, that would be a planned contrast. In the context of the present paper, the term contrast seems to apply to just the main effect of a single variable (with only two levels). There is no need to specify all this. A three- or four-way ANOVA automatically gives the effect of each variable, and given that for most variables there are only two levels, this effect automatically tests if these two levels differ. No contrast necessary.

Only age has four groups so here a contrast could theoretically be applied, but this is not specified.

You do need to explain in the analysis plan which interactions you are going to test and how you are planning to proceed if the interaction is significant.

It is a common procedure if one interaction is significant, to decompose this interaction by going back one level, so if an interaction of A*B is significant you test the simple effect of B for each level of A. Or if A*B*C is significant, test interaction A*B at all levels of C.

p.13 why include an analysis with so many interaction terms if you ignore the results of those interactions anyway? You only look at one main effect. Note that many of the interactions are actually significant, meaning you can’t interpret a main effect, but you have to proceed with caution. Significant higher order interactions mean that the pattern is more complicated than any of the lower order interactions (or main effects) can capture, so their results can’t be interpreted isolated from the rest.

It would make more sense to decompose this significant four-way interaction by a series of follow up analyses: first three-way interactions, then two-way, as mentioned above. Along the way this automatically captures all the intended other effects (of language, number size, type of task and age).

Then some inconsistencies like in line 332 ‘Finally, the triple interaction between age group, language and number size was significant for RT and CR’.

But the discussion says (line 556): ‘The results reveal that the cost entailed by processing base-20 numbers is independent of age and the associated degree of language.’

But it’s not independent of age, it becomes weaker over time, that’s what the three-way interaction showed. The cost did not disappear entirely but it did wane.

Line 670 ‘Our results have practical implications’, this looks forced. What follows is very general ‘linguistic factors have to be taken into account when instructing and evaluating children’. That may be true but it doesn’t follow from this study, it may be more relevant then to look at international comparisons: does Luxembourg drop in rankings after children switch to French? Then I’d see the issue, but in this study, how bad is it if someone is a few milliseconds slower in writing down a certain number word? And if it is bad, how can a teacher take this into account when instructing and evaluating children and aren’t they doing that already?

Please specify the practical implications of this particular study (or argue that this is a fundamental study and as such there are no practical implications, which is perfectly acceptable by the way. Not every study has practical implications).

Minor points

p. 3 line 47-48: ‘The simplest declination corresponding to bilingualism’. What is declination supposed to mean here? Is it form perhaps?

p. 4 line 80 unusual use of ‘peers’, peers are children from the same environment, commonly classmates in a school context, not children from different countries.

P. 6 line 121 what is LM-? The language in which mathematics was not learned? Please define.

Correct terminology is two-way or three-way interaction, not double or triple.

Line 659: ‘It is, however, noteworthy that math education extended for one year longer in LM2 (7 years of primary education) than in LM1 (6 years of secondary education), which might help to

balance the frequencies in both languages.’

This only makes sense if the words primary and secondary were reversed. But it may help to explain the Luxembourgish situation in a bit more detail in the introduction.

---

## [Author Response · Author response to Decision Letter 0]

27 Oct 2021

Reviewer #1

Dear Editor, Prof. Dr Prado, 

dear Reviewer,

We would like to thank the editor and the two anonymous reviewers for their very valuable comments and their constructive suggestions. As synthesized by the editor, two major points were raised by the reviewers: the first point(s) concerned the theoretical framework in general, and the second the statistical analysis applied to the dataset. In the revised version of the manuscript we have addressed these concerns in depth and believe the manuscript’s quality has greatly improved as a consequence. In the following, we describe how we implemented the different points raised by both reviewers into the revised manuscript. Below you can find our answers (A) to the comments (C) to each of the two reviewer’s points. Please note that adjustments in the manuscript are indicated in this colour code.

C1: The theoretical framing is poor, unfocused and sometimes incorrect.

C1.1: I do not refer to the important/impact of the study, because this is not a publication criterion in PLOS ONE. However, irrespectively of the question impact, the manuscript should still interpret the result in terms of relevant theory. There is no reference to relevant cognitive models of number processing, although there are quite a few of those (McCloskey; Dehaene & Cohen; Butterworth & Cipolotti; Dotan & Friedmann; in a way, also Huber, Nuerk, & Willmes; and more). 

A1.1: We agree with the reviewer’s suggestion that the manuscript was not sufficiently considering the different cognitive models that are relevant for understanding number processing and transcoding in particular. To complete the theoretical framework with the above mentioned transcoding models, we added three paragraphs in the introduction describing cognitive models of transcoding (lines 51-77):

Several cognitive models have been proposed to describe transcoding, with some also taking into account cognitive development or the acquisition of a second language. These models are summarized below in two main categories: semantic and asemantic (Barrouillet et al., 2004).

In semantic models, transcoding requires an obligatory access of the number’s magnitude. For example McCloskey (McCloskey, 1992; McCloskey et al., 1985) proposed a transcoding model in which the entry number - regardless from the input’s format - accesses an abstract semantic representation from where it is then handled into a production system. Power and Dal Martello (Power & Dal Martello, 1990) further proposed a model specifically for number dictation (i.e. verbal to Arabic format) which differs from the previous one in that semantic representations reflect the verbal word structure of the numbers. In sum for these models, transcoding activates a common semantic representation, independently of language or age.

In contrast, asemantic models propose that numbers can be transcoded without accessing magnitude. Deloche and Seron (Deloche & Seron, 1982) proposed such a model for number naming (i.e. Arabic to verbal) where the first step involves parsing the input into primitives which are then submitted to a set of rules for the output system. Later on, Barrouillet and colleagues (Barrouillet et al., 2004) proposed ADAPT (A Developmental Asemantic Procedural Model), which includes the developmental perspective that frequent number words become lexicalized and are thus directly retrieved from long term memory, hence bypassing the procedural processes. Another more general model considering number representations is the Triple Code Model (TCM). The TCM proposes a functional and topographical framework of how and where in the brain approximate, visual symbolic and verbal codes are processed. Approximate processing of magnitudes (i.e. not exact quantities) are taken over by the horizontal segment of the intraparietal sulcus. Visual symbols, depending on visuospatial abilities, are processed by the parietal and fusiform gyrus (number form area). Finally, verbal codes, depending on linguistic processes, involve the left angular gyrus of the temporal-parietal gyrus (Dehaene et al., 2003). Therefore, the TCM implies that transcoding from or to a verbal code does not require semantic magnitude activations and is language-dependent (Dehaene, 1992). More recently, Dotan and Friedmann (Dotan & Friedmann, 2018) proposed a model for Arabic to verbal transcoding where numbers are first visually analysed for identity, order and decimal structure and then passed to a second system which applies language-specific rules which are associated with their phonological and articulatory counter-parts. In sum, these asemantic models sustain that semantics are not necessarily activated for transcoding, some including developmental perspectives and others language differences.

In addition we added these paragraph in the discussion, in order to link the cognitive models presented in the introduction with the results that were found (lines 497 to 510): 

From the perspective of cognitive models presented in the introduction, the cost for transparency of power in French would fit with Power and Dal Martello’s semantic model if rules reflecting the vigesimal form are added, since more rules would mean slower production (Power & Dal Martello, 1990). Taking into account development, the asemantic ADAPT model (A Developmental Asemantic and Procedural model for Transcoding (Barrouillet et al., 2004)), would partly fit with our results. ADAPT was proposed for number dictation (i.e. verbal to visual Arabic transcoding), hence similar processes could have taken place in the verbal to visual matching and reading aloud tasks (i.e. visual Arabic to verbal) of the present study design. In ADAPT, French `70s, `80s and `90s require more rules and therefore also more processing time. However, the model further proposes a lexicalization through repetitions over development, i.e. leading to faster transcoding of `70s, `80s and `90s. Yet, when standardizing age groups variability through z-score transformations, we noted that numbers above 70 were similarly slower to transcode in all age groups, arguing against a lexicalization with training. The non-lexicalization of `70s, `80s and `90s vigesimal numbers in French might interact with their late formal acquisition (i.e. LM2 acquired at 7th grade, around 11-12 years old). Finally, the cost for `70s, `80s and `90s numbers could be accounted in Dotan and Freidman’s recent transcoding model (Dotan & Friedmann, 2018) at the phonological level of binding between digits and number words. 

Despite being a model about number dictation, we also added the ADAPT model. First, because it was recommended by the second reviewer. Second, it is the only transcoding model at our knowledge that includes a developmental perspective, which is one of the approaches of our study. Third, a recently published article with a similar verbal visual matching task than our study (Steiner et al., 2021) (p.15) included the ADAPT model and we adapted this article’s reasoning that is generally speaking a model for verbal to Arabic digit’s transcoding, thus implying common processes are probably required in our tasks too.

C1.2: The topic of this article is number transcoding, but for some reasons the authors chose to dedicate much of the theoretical discussions to other issues, which are only vaguely related – ANS, arithmetic, and language. Perhaps the two most extreme examples for this is that (1) the first page of the manuscript (!!) is simply irrelevant – except the last few lines, it talks mostly about ANS and brain regions, two topics that are only vaguely related to the article’s main point. (2) The first sentence in the abstract is simply incorrect. It says that the study examined the “analogic, verbal, and visual symbolic codes”, but the “analogic code” (ANS) was not really examined here.

A1.2: We agree with the reviewer’s critique about the length of the first page. Our original aim was to refer to the triple code model of Dehaene, underlying the possible multiplicity of verbal codes. We see now that it is over-emphasized and the approximate code is irrelevant, especially for the introduction. Hence, we have reduced the first page’s content to a paragraph, removing the passages concerning approximate representations (lines 42-48). 

Exact numerical representations are supported by symbolic verbal (e.g. forty-two) and visual (e.g. 42) representations which are acquired through learning. However, in many languages the verbal number word’s syntax differs from the visual one, i.e. the Arabic number system (Chrisomalis, 2010; Ifrah & Bellos, 2000). This difference might have an influence on transcoding, i.e. the cognitive transformation from one code to another. Thus reading aloud for instance the number “42” implies the transcoding from a visual (“42”) to a verbal code (“forty-two”). Moreover, in the case of bi- and multilingualism, acquiring multiple languages means that during development multiple verbal codes are mapped with the visual code (Salillas & Martínez, 2018). 

Accordingly, the first sentence of the abstract was also adjusted (lines 21 to 23)

Number transcoding is the cognitive task of converting between different numerical codes (i.e. visual “42”, verbal “forty-two”). Visual symbolic to verbal transcoding and vice versa strongly relies on language proficiency.

C1.3: Similarly, the selection of cited literature seems a bit odd throughout most of the manuscript (but in the Discussion it is good). There is a lot of published work on transcoding, but the article ignores much of this literature and cites literature from other domains, which is less relevant or hardly relevant. There is no citation at all of many relevant works by researchers who did a lot of theoretical work on transcoding, e.g., Michael McCloskey, Carlo Semenza, Margaret Delzaer, Marie-Pascal Noël, Xavier Seron, Laurent Cohen and more – including works that specifically addressed the base-10 vs base-20 issue. Even when talking about arithmetic, there are additional studies that specifically examined the role of the verbal structure of numbers and could be cited (e.g., Campbell & Clark, 1992; Gonzalez & Kolers, 1982; Noël & Seron, 1997).

A1.3: We thank the reviewer for her/his suggestion concerning the literature. We added several additional references such as McCloskey [7] and Seron [10]. However, we decided not to add all the proposed references, since the aim of the present study was not to compare results between developmental and neuropsychological literature.

C1.4: On few occasions, the manuscript cited general language literature as a reference to number-specific phenomena, e.g. references [49] [52]. I would prefer seeing citation of studies more specific to numbers. 

A1.4: We agree with the reviewer’s suggestion to remain focused on literature concerning numerical cognition. We deleted the references from outside the field of numerical cognition, since these are not strictly necessary (lines 176 to 177):

However, they agree with recent reports from the bilingualism literature that academic knowledge acquired in one language is not (or difficultly) transferable to the other language (Volmer et al., 2018).,60,61]. 

C2: The article’s main theoretical question requires the comparison of base-10 versus base-20 verbal numbers. To operationalize this, the authors took the French numbers above 60 as reflecting the base-20 system. This does not seem justified. In French, the syntactic structure of verbal numbers in the 60’s and 80’s is congruent with both systems (b10 and b20). Only numbers in the 70’s and 90’s are clear examples for base-20 syntax. I would guess that if you focus only on the 70’s and 90’s, your results will actually improve. Also, perhaps you may even be able to show that the 80’s are as easy as the smaller numbers – this could be a nice control for an artefact of number size (as opposed to syntactic structure, the effect you’re actually interested in).

The manuscript seems to assume that numbers in the 80’s are cognitively represented as “20 times 4 plus X”, e.g., 83=20x4+3. Presumably, the reason is that the French word for 80 is “quatre-vingt”, “four-twenty” (although this is not explained explicitly). Although this argument has been made in past studies, it is not convincing at all – “quatre-vingt” is very likely to be one lexicalized value, even if its etymology is 20x4. If there is empirical evidence showing that 80 is represented in a decomposed manner in the relevant population, please cite it. If not, please remove this claim, or at least present it as an unsupported hypothesis. 

Note that even if you view 80 as decomposed 20x4, this still does not mean that the numbers 81-89 are in base-20, because the issue of 80 vs. 20x4 concerns only the tens word, not the way it is composed with the ones word.

A2: We thank the reviewer for this very valuable suggestion. We agree that strictly speaking ‘80s (quatre-vingt 4*20) and ‘90s (quatre-vingt-dix 4*20+10) are clear representations of a base-20 system. On a conceptual level, ‘70s (soixante-dix 60+10) are not in base-10 neither, hence we used the term “vigesimal” to refer to these particular number word structure that distinguishes French from German. The question if base-20 are transcoded from algorithmic or procedural mechanisms (i.e. 20 times 4 plus X”, e.g., 83=20x4+3) remains theoretical. Certain transcoding models (i.e. Power Dal Marelo, Seron, Barrouillet) propose it might be the case, at least for very young children. This would explain the higher rate of transcoding errors in (Seron & Fayol, 1994), and the prolonged reaction times in (Van Rinsveld & Schiltz, 2016). 

Accordingly, we decided to adjust our analyses to make a slight change regarding the comparisons between numbers over and under 60. Instead, we dropped the ‘60s (as it is in base-10) and compared ‘30s, ‘40s, ‘50s vs ‘70s, ‘80s, ‘90s. This approach is supported by additional analyses (see annex), indicating that ‘60s were not significantly different than ‘50s, and hence the cognitive cost related to processing vigesimal numbers in French is operating from ‘70s (“soixante-dix”) to ‘90s (“quatre-vingt-dix”) . Hence, also figures and tables were updated. 

Since the question if ‘80s are as fast to transcode as smaller numbers is an empirical one, we tested this statistically. In the annex, we compared the different decades to test the hypothesis of (a) are ‘60s faster than ‘70s (see previous paragraph) and (b) are ‘80s (“quatre-vingt”) faster than ‘70s and ‘90s. This allowed contrasting each decade with respectively previous and following one, to spot a “statistical jump” in terms of RT. As a result, we noticed that ‘60s are indeed not significantly different from ‘50s. Moreover, ‘80s were also not significantly different from ‘70s and ‘90s in both tasks. Note that for the reading aloud tasks in 5th and 11th graders, ‘80s and ‘90s were almost significantly different (z = 1.96, p = .05), probably due to number words lengths (i.e. 87:“quatrevingtsept” vs 97: “quatrevingdixtsept”). Hence, the majority of these comparisons seem to refute the hypothesis that “quatre-vingt” (80) are lexicalized.

C3: The theoretical interpretation in the discussion is not convincing and should be improved. 

C.3 (a) The theoretical interpretation in lines 560-575 seems incorrect in several ways.

- Lines 560-562: this is not in good agreement with commonly-accepted cognitive models of number production (e.g. McCloskey et al 1986). Phonologically, both “quatre-ving” and “dix-sept” are probably single lexical entries so these are two words, not four. I am not sure this was tested in French, but certainly you cannot claim the opposite as if it was a fact.

- Even if we assume that French 97 requires activating 4 lexical entries, it is still not clear why the cost should be attributed specifically to interference (and not, for example, to the effort arising from the need to retrieve more words).

- Lines 566-568, reference [58]: the example is completely irrelevant, and in this context, even a bit misleading. Of course, some cognitive interference effects exist. But it doesn’t mean that this is the case for numbers too. Especially since numerous studies show fundamental dissociations between the production mechanisms of number words and non-number words.

A.3 (a) We thank the reviewer for this suggestion and agree that it is probably too speculative to state that the two words building “quate-vingt” require multiple lexical activation’s generating lexical interference. We have therefore depleted this interpretation from the discussion. We have restated the interpretations by referring more explicitly to cognitive models of transcoding in these rewritten paragraphs. We also designated more explicitly some possible links across the numerical cognition and psycholinguistic domains (lines 497-519)

From the perspective of cognitive models presented in the introduction, the cost for transparency of power in French would fit with Power and Dal Martello’s semantic model if rules reflecting the vigesimal form are added, since more rules would mean slower production (Power & Dal Martello, 1990). Taking into account development, the asemantic ADAPT model (A Developmental Asemantic and Procedural model for Transcoding (Barrouillet et al., 2004)), would partly fit with our results. ADAPT was proposed for number dictation (i.e. verbal to visual Arabic transcoding), hence similar processes could have taken place in the verbal to visual matching and reading aloud tasks (i.e. visual Arabic to verbal) of the present study design. In ADAPT, French `70s, `80s and `90s require more rules and therefore also more processing time. However, the model further proposes a lexicalization through repetitions over development, i.e. leading to faster transcoding of `70s, `80s and `90s. Yet, when standardizing age groups variability through z-score transformations, we noted that numbers above 70 were similarly slower to transcode in all age groups, arguing against a lexicalization with training. The non-lexicalization of `70s, `80s and `90s vigesimal numbers in French might interact with their late formal acquisition (i.e. LM2 acquired at 7th grade, around 11-12 years old). Finally, the cost for `70s, `80s and `90s numbers could be accounted in Dotan and Freidman’s recent transcoding model (Dotan & Friedmann, 2018) at the phonological level of binding between digits and number words. 

Speculatively, it could be argued that for French morpho-syntactically complex number words that are not lexicalized, multiple additional morphemes need to be retrieved (see i.e. (Meeuwissen et al., 2003)). This could be the origin of the cognitive slow-down as it requires to retrieve more lexical morphemes (i.e. four for 97: “quatre”, “vingt”, “dix”, “sept”). Eventually these morphemes, which are made of other numbers might interfere among each other, in a similar fashion as for multiplications (i.e. proactive interference (Campbell, 1995; De Visscher & Noël, 2014)). 

Please note that all three models would also generally predict slower responses for German number words above 12 due to the additional inversion rule in German. Such effects of transparency of order have been found in previous studies (Poncin et al., 2019; Steiner et al., 2021; Zuber et al., 2009), but it is possible these were masked in the present study by the effect of language of math acquisition described in the following section.

An also at lines (565-571):

The LM2 cost is also explainable from a psycho-linguistic perspective: here we present a syntactic and a lexical interpretation. Syntactically, the LM2 cost might result from an over-generalization of the LM1 syntactic inversion rule (see for example (Zuber et al., 2009)): transcoding number words in French would require inhibition of the inversion rule (over)learned from German. A lexical explanations can be found in more general bilingual models such as the bilingual interactive activation model (BIA+), predicting a response competition between both languages, with faster activation for the more frequently used language (Dijkstra & van Heuven, 2002). In this framework, the slower lexical retrieval of LM2 would also have detrimental impacts on short term memory (Ellis & Hennelly, 1980; Gathercole & Baddeley, 1993) and hence arithmetic (Friso-van den Bos et al., 2013).

C.3 (b): “Lines 576-598, 623-629: it seems strange to interpret your findings according to the ADAPT model, because ADAPT is a model for number writing (verbal-to-arabic transcoding), and the task here is reading (arabic-to-verbal transcoding). You can use models of number reading instead.

I don’t think this should disturb you because you don’t really rely on any particularity of the ADAPT model, you merely rely on the well-accepted, widely-supported distinction between lexical processing and syntactic processing in numbers. What your data shows is probably a phenomenon that was shown in many previous studies: numbers with complicated syntactic structure are harder to process than syntactically-simple numbers, and syntactic ability develops with age. I think it would be more appropriate to frame your claim in these more-general terms, rather than present it as if the results support a particular cognitive model – I don’t think they do. If I were you, I would just drop the interpretation in lines 560-575, stick with the second interpretation, and present the results as evidence for a syntactic-complexity effect (a conceptual replication of several previous studies), and for the development of syntactic abilities with age. But needless to say, this is of course totally your decision.

Either way, I think that the manuscript will benefit from using the terminology “number syntax”. Although sometimes vague, this is standard terminology (certainly more standard than “transparency of power”) and it seems appropriate for your research question.”

A.3 (b): We thank the reviewer for the suggestion. Given the well-funded request to provide a more thorough theoretical embedding of our work (e.g. C1), we decided to keep the ADAPT model as an interpretation of the results, since the model proposes a verbal to visual transcoding (that is involved in our tasks). For this approach we also aligned with recent studies on transcoding adopting a similar verbal-visual matching paradigm (Steiner et al., 2021) .

C.3 (c) Lines 613 seem like a vague mixture of several phenomena/issues. I found it more confusing than clarifying.

- Lines 613-614: your findings do not seem to be about language switching, but about familiarity with language. While I agree that language switching is probably relevant, I find this analogy more blurring than clarifying. At least discuss the difference between the two phenomena

A.3 (c): We agree with the reviewer’s suggestion. Hence, we have adapted the following paragraph (lines 535 to 543)

Our findings are in line with studies reporting qualitatively different arithmetic performance with LM+ compared to LM- in bilingual Filipino-English and Spanish-Basque participants, even if LM- corresponds to participants’ mother language (Bernardo, 2001; Salillas & Wicha, 2012). Finally they also match and extend the finding that solving addition problems was slower in LM2 than LM1 in participants coming from the same education system than in the present study (Van Rinsveld et al., 2015). 

Interestingly, while the language of math acquisition impacted the number reading task, this was considerably less the case in the verbal-visual matching task. Lexically, the different pattern of results for the LM2 could be explained by different memory retrieval mechanisms (Vander Beken & Brysbaert, 2018) since the number reading task can be considered as a form of free recall while the matching task is more similar to a familiarity judgment.

And lines 576-577:

A limitation of the present study is that it did not allow to disentangle the role played by the language of math acquisition from the familiarity with number words in the two languages.

- 619-620: the encoding complex model makes very specific assumptions about the types of information coded in language-specific way (in particular arithmetic facts). The language effect you have found doesn’t seem to support these assumptions – it merely shows that language, and familiarity with language, matters. Frankly, this effect is a bit trivial, and pretty much agrees with any number-processing model I am aware of.

A.3 (c): We agree with the reviewer’s suggestion and have therefore deleted the following lines: (lines 517)

These costs support the Encoding Complex Model suggesting that numbers might be encoded in a language and format-specific manner [70]

C.3 (d) Lines 630-638: your data doesn’t seem to support working-memory effects in any way. Certainly, difficulty in lexical retrieval does not necessarily imply a WM issue.

A.3 (d): We thank the reviewer for this suggestion. While the present study did not indeed address the role of working memory, the relation between transcoding and short-term memory is sustained by several studies (Camos, 2008; Imbo et al., 2014; Pixner et al., 2011; Steiner et al., 2021; Zuber et al., 2009). We therefore included a brief mention to working memory in the discussion, as a potential link between transcoding and arithmetics.

C.4 In the results section, some of the statistical analyses are described dryly, without any meaning/interpretation. Please add their theoretical implication. E.g. in p.15, the precise pattern of interactions with age seem to strengthen your theoretical claim about a progression of learning – why not say it explicitly right away? Especially in the summary paragraphs (lines 406-410, 495-503) I would expect that the text doesn’t just talk about factors and effects but also say something about the cognitive implications.

Even when the results are interpreted, this is sometimes done separately from the result. E.g. the analyses of both tasks start by saying that all double interactions were significant, but these interactions are explained only later in the text. Please fix: unite each result with its interpretation.

The comparison of the two tasks doesn’t seem to have any theoretical relevance and should probably be deleted (or at least pushed down to somewhere less central in the text). As far as I understand, there is no theoretical reason to assume that the performance in the two tasks would be similar, and we don’t learn any insights from this comparison.

A4: We agree with the reviewer’s suggestion and because of the change of statistical analyses (LMM instead of ANOVA) we have rewritten the result section taking into account these suggestions. First, we have taken into account the unification between interpretation within the results (see Results, p.12 to 19).

Second, we have dropped the statistical comparison between the tasks. By removing the following lines (line 180) 

While transcoding involves the search within a full set of lexical information in the production task, the recognition task only requires an already cued search amongst a limited number of response alternatives, which is largely based on familiarity processing [52].

C5: Some of the terminology used in the manuscript is imprecise, non-standard or incorrect.

- “Language of math acquisition” is a confusing and imprecise term. You did not manipulate the language of acquisition (=learning), you manipulated the language in which the experiment was done. But I certainly agree that the reason for this effect is likely to be LM1 vs. LM2

- The first task is neither “a production task” nor “number naming”, it’s reading aloud (it includes both visual input and verbal production). The second task would be better described not as a recognition task, but as a matching task.

 -“Transparency of power” is not a standard term in the number transcoding literature, so please explain it before using it for the first time. Perhaps avoid using it in the abstract.

C5 We agree thank the reviewer for her/his suggestion. Despite the fact that language of mathematical acquisition was not experimentally manipulated, we kept this wording, as it is in line with previous work on arithmetic problem solving in the same multilingual educational context (Van Rinsveld et al., 2015, 2017). Throughout the paper, we have adapted the terminology for the two tasks according to the reviewers’ suggestions (i.e. from production task to reading aloud task and from recognition to verbal-visual number matching task. In general throughout the paper we also tried to reduce repetitions and shorten paragraphs (for brevity, these changes are not listed below, but can be found in the revised paper). The term “transparency of power” we also clarified, by explaining it in the abstract, and referencing its previous use in a taxonomy for the different effects of language on transcoding (Bahnmueller et al., 2018; Dowker & Nuerk, 2016) (see also line 91).

- When saying the performance is better in condition X, say better than what (e.g. p. 17 line 407)

- The inversion in German is not between tens and units, it is between tens and ones (“units” = digits, “ones” = words). 

Thank you for this suggestion, we have accordingly modified the following sentence (lines 95-97).

Another morpho-syntactic difference concerns the inversion between tens and ones as for example in German and Dutch (i.e. y + 10*x). 

- Lines 579-581: there is a confusion here between two meanings of the term “semantic”. In the ADAPT model (and in other models, e.g. McCloskey’s), “semantic” often refers to a central abstract representation of the number. The number size effect refers to semantics as the quantity associated with the number via the ANS. These are two completely different things.

Thank you for noting this, it was not our intention to link both terms for semantics, due to possible confusion between the meanings of semantics, we deleted those lines.

- “Analogic code” is a correct/acceptable term for ANS, but it is not standard. “Approximate” would be a more standard terminology

We substituted this terminology throughout the paper, adopting the more standard one (lines 66-69)

The TCM proposes a functional and topographical framework of how and where in the brain approximate, visual symbolic and verbal codes are processed. Approximate processing of magnitudes (i.e. not exact quantities) are taken over by the horizontal segment of the intraparietal sulcus. Visual symbols, depending on visuospatial abilities, are processed by the parietal and fusiform gyrus (number form area).

C6: I would recommend using linear mixed model instead of ANOVA. It is more reliable, and also might improve your statistical results.

A6: We have re-written anew the result section, since we changed the analyses from ANOVA to linear mixed models. However, since the experiment was not designed for LMM, we first had to test different models because to assure convergence of the models. The model that could be fitted for both tasks RT’s includes per-item random slopes and intercepts for each language, but only random intercepts per subjects. As suggested (“70 makes much more sense than 60 though”) we adjusted the analyses to compare `30s, `40s and `50s vs `70s, `80s and `90s numbers, instead of over and under 60. (see Results section, pages 12 to 19).

C7: The quality of writing should be improved.

- The manuscript should undergo an English language review.

- The text is too long, please write more concisely. Just to give a few examples:

(1) The issue of French>60 examines transparency of power is explained too many time.

(2) The first paragraph in the results section can be cut into a short sentence. But there are many more.

(3) The text describing the analyses with (p. 16) and without (p.15) z-scored are redundant and tiring. You can write the text once, and say you got essentially the same results with and without z-scores (with results of both ANOVAs in a table). Similarly, fig. 6 is redundant and can be removed. Same issue for recognition task.

But I emphasize that these are just examples, please write the whole text more concisely

A8: Throughout the paper we have thoroughly revised the English language and formulations, aiming to remove redundancies and formulate some paragraphs more concisely. For example we have shortened the te first paragraph in the Transparency of power section (lines 92-95):

In many Asian languages (i.e. Mandarin Chinese, Vietnamese) the morpho-syntactic structure of the verbal number system is highly consistent with the Arabic number system, and in general with the base-10 (i.e. 10*x + y , see (Miura et al., 1988)). This linguistic characteristic, also termed transparency of power, facilitates learning to count (Lê & Noël, 2020; Miller et al., 1995; Miller & Stigler, 1987) and solving arithmetic problems (Geary et al., 1996; McClung & Arya, 2018; Rodic et al., 2015).

C8 Minor comments

We thank the reviewer for the minor comments, we shortly comment them below :

C8.1. The statistical analyses are described in detail, which is excellent, but because there are quite a few of them, I would recommend moving them to the results section. I completely lost track of the analyses while reading, and it was very hard to relate specific results to specific analyses.

A 8.1: The whole result section has been rewritten anew.

C8.2. Fig. 3 – the Y axis label says RT instead of accuracy

A8.2: Thank you for pointing out this inadvertence error, which has been corrected.

C8.3. Fig. 4 – instead of o60 and u60, perhaps write something clearer, e.g. >60, <60

A8.3: we have changed the dataset subdivision (see above), hence now the comparison is between small (‘30s, ‘40s and ’50s) vs big (‘70s, ‘80s and 90’s) decades

C8.4. Lines 344/355 – I think there is an unintended paragraph break here.

A8.4. Thank you for pointing this out.

C8.5. There are quite a lot of ANOVA results. Consider putting them all in table/s instead of in the text – it will cut the text shorter and could make it easier to find the relevant stats.

A8.6. We have added tables with the results (see Table 1 to 4)

C8.7. Lines 344-351: to show that the German and French trends are different with respect to >60 vs <60, here would be the correct place to talk about the Size x Language interaction.

A8.7. The result section is completely rewritten anew.

C8.8. Line 390: how many 4-syllable numbers are there? Also, as far as I know, the relevant measure for word length is typically assumed to be the number of phonemes, not the number of syllables (Nickels & Howard, 2004, Cognitive Neuropsychology).

A8.8. We have followed the same procedure as in (Van Rinsveld & Schiltz, 2016)

C8.9. The particularity of the German language (tens-ones inversion) should be mentioned earlier on in the text.

For the German number system, we added a description of transparency of power in the introduction (lines 95 to 99).

Another morpho-syntactic difference concerns the inversion between tens and ones as for example in German and Dutch (i.e. y + 10*x). This linguistic characteristic, called transparency of order (Bahnmueller et al., 2018), explains some transcoding error patterns and reaction times (Clayton et al., 2020; Imbo et al., 2014; Moeller et al., 2015; Pixner et al., 2011; Poncin et al., 2019; Steiner et al., 2021; Zuber et al., 2009), gives rise to specific pattern of compatibility effects (Bahnmueller et al., 2015; H. Nuerk et al., 2005; H.-C. Nuerk et al., 2004) and complicates the solution of certain arithmetic problems (Göbel et al., 2014; Lonnemann & Yan, 2015; Xenidou-Dervou et al., 2015).

A8.9. We added a few lines in the introduction

C8.10. All figures: measurement units “millisec” should be written only on axis label, not in the figure caption.

A8.10 This has been modified accordingly

C8.11. The introduction and discussion mention a transparency-of-power analysis for numbers above 70 but I think I didn’t see it in the results section, did I miss it?

A8.11. In the present revised manuscript version we only use the cut-off from 70 onwards

C8.12. Line 556: as far as I can see this is incorrect - the base-20 cost was not independent of age, it decreased with age (even if it existed to some extent in all age groups).

A8.12. We interpreted that the cost in French for `70s `80s `90s numbers was independently of age based on the z-scores. The reasoning being that the z-scores standardized the variability present in each age groups and since we do not find an interaction between Age and Size (see Supplementary Materials), it might mean that the results from the raw RT are biased due to bigger variance. This has now been clarified in lines 491 to 495.

Notably, the cost was found for both reading aloud and number matching tasks that is transcoding from visual to verbal and vice-versa. The results from the z-scores standardization suggest that the cost explained by a difference in transparency of power, generally and stably persisted from 5th grade up to adulthood. In other words the cost maintains across age groups, tasks and the associated degree of language proficiency. 

C8.13. The reference to ADAPT in line 577 is incorrect, [60] instead of [64]. Please double check all references.

A8.13 We thank the reviewer for noticing this. We have corrected the reference and checked that all references.

C8.14. Lines 180-182: this claim does not seem correct – at least, this is not the common assumption for how number transcoding works. Certainly, you cannot base this claim on reference [52] not from the number domain.

A8.14. We thank the reviewer for noticing this. We have removed this interpretation.

 

Reviewer #2

Dear Editor, Prof. Dr Prado, 

dear Reviewer,

We would like to thank the editor and the two anonymous reviewers for their very valuable comments and their constructive suggestions. As synthesized by the editor, two major points were raised by the reviewers: the first point(s) concerned the theoretical framework in general, and the second the statistical analysis applied to the dataset. In the revised version of the manuscript we have addressed these concerns in depth and believe the manuscript’s quality has greatly improved as a consequence. In the following, we describe how we implemented the different points raised by both reviewers into the revised manuscript. Below you can find our answers (A) to the comments (C) to each of the two reviewer’s points. Please note that adjustments in the manuscript are indicated in this colour code.

C1: line 44-46: Since multilinguals are outnumbering monolinguals across the globe [12], the question of how numbers are processed and transcoded in different languages is of crucial importance.’ Why? Undeniably there are a lot of multilinguals but that fact doesn’t necessarily make this particular research question crucially important. Please either remove or explain the relevance of the research question for all those multilinguals. This is a bit related to the discussion, see also later. If this is fundamental study that is perfectly fine, but don't pretend it also has immediate applications or benefits.

A1: We agree with the reviewer’s suggestion concerning the applicability of the study’s outcome. In general, and this is also related to a later comment of the reviewer (see A13), we tried to reorient the formulations into a fundamental rather than applied relevance of our study. For example, as suggested by the first reviewer, by adding new paragraphs concerning transcoding models in both introduction (lines 51-77) and discussion (lines 477-491). Furthermore, we reformulated also why we think it is relevant to investigate transcoding in bilinguals (lines 48-50): 

Given the importance of language in representing numbers comparing these additional transcoding routes is of crucial importance. Since multilinguals are outnumbering monolinguals across the globe (Grosjean, 2010), the question of how numbers are processed and particularly transcoded using two (or more) verbal codes in a (developing) multilingual cognitive system is of crucial importance (Wicha et al., 2018). 

C2.1: line 75-78: This section would benefit from a clearer explanation of the different counting systems in German and French for those unfamiliar with those languages. Now the base-10 vs 20 is highlighted, but first I’d argue that the base 20 system in French is only visible for the 80s and 90s and not below (or else 62 would be 3x20+2 which is not the case). I’d say that the true base 20 starts at 80, as the 60’s follow the regular base 10 system and the 70’s are neither (as they are 60+10, 60+11 etc). And then there are the irregular number words for 11-16.

Line 77 ‘only eighty decades are vigesimal’ probably supposed to mean ‘only the decade 80’? Note by the way that also languages like English and French are not as transparent as Chinese in the base 10 system. If they were, a number like 60 in French would be ‘six dix’ (six ten) instead of ‘soixante’.

And please also elaborate on differences between the French and German counting system. The explanation of the base 10 vs 20 system is very elaborate but the decade unit inversion in German is barely mentioned, so short that I doubt whether a reader who is unfamiliar with languages that have this inversion even understands what it is. And what where the results from references [18] and [51]? 

And I wanted to recommend some more papers on this phenomenon but I already found a few in the reference list without referencing to them in the text [65-67]. Was this part deliberately taken away to give the simplified impression that French numbers are always more difficult than German numbers? Decade unit inversion makes transcoding more difficult. This could become especially visible in the

A2.1: We fully agree with the reviewer’s suggestion to improve the clarification of the French and German counting systems and their differences. Indeed, we had included a paragraph concerning the inversion in German in a draft version that we finally removed for reasons of conciseness in our last version (leaving by mistake the related references). While the effect of inversion in German is not directly investigated in our study (motivating our initial removal of the passage), we nevertheless agree with the reviewer that it should be mentioned for the sake of completeness. The following paragraph has been added to explain the inversion in German and related difficulties (lines 95-99).

Another morpho-syntactic difference concerns the inversion between tens and ones as for example in German and Dutch (i.e. y + 10*x). This linguistic characteristic, called transparency of order (Bahnmueller et al., 2018), explains some transcoding error patterns and reaction times (Clayton et al., 2020; Imbo et al., 2014; Moeller et al., 2015; Pixner et al., 2011; Poncin et al., 2019; Steiner et al., 2021; Zuber et al., 2009), gives rise to specific pattern of compatibility effects (Bahnmueller et al., 2015; H. Nuerk et al., 2005; H.-C. Nuerk et al., 2004) and complicates the solution of certain arithmetic problems (Göbel et al., 2014; Lonnemann & Yan, 2015; Xenidou-Dervou et al., 2015).

Concerning the base-20 in French, we agree that the base-20 system is better represented in the ‘80s (quatre-vingt 4*20) and ‘90s (quatre-vingt-dix 4*20+10) than in the ‘60s. While the ‘70s (soixante-dix 60+10) are nor clearly in base-20, nor in base-10. However, from a conceptual level since ‘70s use teen numbers that are only used in another base-20 decade (‘90s) we believe they are more similar in their structure to ‘80s and ‘90s than ‘60s. In particular, when put in comparison with German. Hence we used the term “Vigesimal” to refer to this particular number word structure that distinguishes French from German. We now state this more clearly this in the below paragraph. We also thank the reviewer for correctly pointing out the lexical primitives that are present in French vigesimal numbers, we added one sentence specifying this characteristic (lines 100-108): 

Transparency of power can also vary with a change of base, for example a base-20 system (20*x + y), also called vigesimal. The use of the vigesimal system is found for example in French, Basque, or in Diola-Fogny (see (Haspelmath et al., 2005)). French, to be precise, has a mixed system since it uses base-10 and base-20 systems. Up to the `60s in French, like in English, the counting system follows the base-10 rule. However, `80s to `90s follow a base-20 system, e.g. 87 = 4*20 + 7, “four-twenty-seven”. `70s are a particular case, as they are composed with `60s and teens numbers, e.g. 77 = 60 + 17, “sixty-seventeen”. Moreover, the teens (11 to 16) are lexical primitives, like in English and in stark contrast to the more transparent Asian languages. This system is subject to regional variances, for example in Belgian Wallonia only `80s are vigesimal (80, “quatre-vingt”, literally “four-twenty”) and in some Swiss-French cantons the vigesimal system is entirely absent (80, “huitante”, literally “eighty”). 

C3: The ADAPT model of number processing is mentioned in the discussion but perhaps this could be already done in the introduction, with an integrated approach discussing the difficulties of all languages.

A3: We thank the reviewer for this suggestion. We agree about the importance of the ADAPT model and have added it in the introduction, integrated into a novel paragraph on different transcoding models (lines 62-65). 

Later on, Barrouillet and colleagues (Barrouillet et al., 2004) proposed ADAPT (A Developmental Asemantic Procedural Model), which includes the developmental perspective that frequent number words become lexicalized and are thus directly retrieved from long term memory, hence bypassing the procedural processes. 

C4: p. 4 line 82. ‘Analogously’ but what follows is not analogously, it is in contrast.

A4: We thank the reviewer for noting this ambiguity in this sentence’s formulation. We reformulated it follows (lines 112-113)

Further indicating an interaction between the number structure and number processing, Basque-speaking adults solved additions faster when the operand was in the base-20 structure.

C5: The present study – Please, for a complete picture, describe what language children in Luxembourg speak before going to school. 

Luxembourgish, German, French? Many children speak Luxembourgish, that is highly similar to German, right? This may especially be relevant if parents stimulate number activities. I see that it follows in the methods section- a lot of different languages and French was excluded. Please elaborate on these choices and potential effects of this choice. In fact, the children with French as native language may form a very interesting group (although a bit small in size if there is age variation) to disentangle L1 from ML1 influences. 

A5: Concerning the language that children speak at home, this is probably mainly Luxembourgish, which is linguistically close to German as correctly noted by the reviewer. We fully agree on the plausible role of parental activity on numerical learning in the present samples. In the present study participant’s mother tongues (which are likely also their home-language) profiles were very diverse, with nonetheless a majority of Luxembourgish (64 out of 99), limiting the possible influence of different parental number language activity on transcoding. Here we excluded participants speaking French at home, since our study compared transcoding in German (i.e. LM1) and French (i.e. LM2). We reformulated the corresponding sentence for greater clarity (l278-280). We find the reviewer’s suggestion to analyze French mother-tongue participants very interesting (that is the participants who were excluded from the final sample). Unfortunately, the French sample was too small to consider each age group to obtain robust results (total 25, but 6 5th graders, 4 8th graders, 7 11th graders and 8 adults). (lines 216-218)

From the initial full sample of 125, we first excluded participants who reported having French as their mother tongue, as these participants might have acquired French number words outside the context of formal schooling.

C6. p.8 I understand how numbers below 60 are a purer indication of an effect of language of math acquisition (although confounded with decade-unit inversion) but this doesn’t stop at 60, sure numbers above 60 are also affected by this, not just by an effect of transparency of power.

A6. We agree with the reviewer that the effect of language of math acquisition also concerns numbers above 60. Probably, the effect of language transparency and the effect of language of math acquisition are cumulative, such that vigesimal number words in French are slower because they are less transparent and they constitute the second language of learning mathematics. However, the effect of transparency of power is based on the difference within French, hence the effect of language of math acquisition should be factored out when comparing transparent (< 60) and non-transparent (>70) numbers (see also Fig.1 in the paper).

C7: 70 makes much more sense than 60 though, because in French only 70 and above cannot be transcribed by merely writing the numbers that you hear, the numbers in the 60s can. It is stated that the analyses were also performed with the cut-off at 70 but were there any relevant differences?

A7: We agree with the reviewer’s suggestion that ‘70s is a better cut-offs than ‘60s. We used ‘60s as cut-off in the first version because we while hearing ‘60s, the participant cannot predict if it would have been the ‘60s or the ‘70s decade (i.e. starting with the same number words soixante) (Avanzi & Thibault, 2019). However, we agree that using a cut-off of 70 allows to better contrast numbers having distinct transparencies of power. In accordance with the reviewer’s suggestion, we directly therefore re-analyzed the data comparing ‘30s, ‘40s, ‘50s to ‘70s, ‘80s and ‘90s (see results section, and changes throughout the manuscript). The analyses with the cut-off at ‘70s led to similar patterns of results. Furthermore this approach is in line with the observation that transcoding of French ‘60s is not different in terms of RT than transcoding of ‘50s (while ‘70s are, see annex for additional fine-grained analyses where we compared each decade with each other)

C8: Furthermore, the study make an attempt to identify effects related to the language structure and effects related to language of acquisition but the two are confounded while a control group is lacking. Any effect in favor of German can be due to L1/math language or to language structure. It would be helpful to compare the results to monolingual French and German control groups (and as mentioned above, the native French speaking children in Luxembourg), not just to mention this in the discussion but actually add those groups.

A8: Regarding the suggestion to add a monolingual group as a control, we argue that this additional group would be difficult to compare with a bilingual group. Indeed, we think that including a monolingual group might also not be an ideal control because it would entail the comparison of different school systems and national curricula (Krinzinger et al., 2011). This difference might be additionally accentuated due to school systems that are designed for monolingual or bilingual teaching approaches. Monolingual German and French groups would also display differences in their transcoding patterns, especially in the younger age groups (see (Poncin et al., 2019; Steiner et al., 2021). Moreover, the individual cognitive level, the speed of lexical retrieval of monolinguals is faster than for bilinguals (even for the L1), which would cause different baselines (Gollan et al., 2005; Ivanova & Costa, 2008; Sullivan et al., 2018). This would add several additional layers of complexity to the study design and interpretation, which would go beyond the aims and scope of the present investigation.

C9: Design and statistical analyses: it’s not explained what was done with separate stimuli. 

How were aggregate scores determined? 

Why not run a mixed effects model instead of ANOVA? The afex package that was used can handle that as well and you can estimate the effects more precisely because it can also handle random slopes across participants (and items). Not taking that into account can lead to alpha inflation. 

Similarly for accuracy, which is bounded by 0% and 100%, ANOVA typically gives the wrong results, often spurious significant results but p-values can also be too low. Especially when values are close to the lower or upper boundary (which is the case with these very high accuracy rates close to ceiling) because ANOVA requires unbounded data. You need a model for proportional data such as a binomial mixed effects model in that case. See for instance Jaeger, T. F. (2008). Categorical data analysis: Away from ANOVAs (transformation or not) and towards logit mixed models. Journal of memory and language, 59(4), 434-446.

But also for the RT data a huge benefit of a mixed effects model is that you can enter all stimuli separately and you don’t need to calculate averages per condition, thus making maximal use of the available information.

Do note though that mixed effects models aren’t easy, and should be carried out appropriately with sufficient knowledge, or else the results may be off as well.

A9: We agree with the reviewer’s suggestion that mixed linear model allow considering all available information optimally. Therefore, we followed this suggestion and completely re-analyzed our data using linear mixed models instead of ANOVA. We also applied binomial models for correct responses. Due to difficulties with convergence, we nevertheless had to remove the age group factor for the latter analyses. As suggested in C7 we adjusted the analyses to compare `30s, `40s and `50s vs `70s, `80s and `90s, instead of over and under 60. This led to similar patterns of results than with the ANOVA. Hence, we have rewritten the whole result section anew (see results section pages 12 to 19). Since the experiment was not designed for LMM, we first had to test different models because of issues with convergences. The model that could be fitted for both tasks' RTs includes per-item random slopes and intercepts for each language, but only random intercepts per subject. 

C10: Analysis plan: explain the dependent variable.

A10: We agree with the reviewer’s suggestion and added the dependent variables (lines 267-268)

The transcoding of two-digit numbers was compared in a transversal developmental design by Linear Mixed Models (LMM) on the dependent variables reaction time and correct responses.

C11: Why speak of ‘groups of contrasts’ instead of just ‘contrast’? Moreover, I’m a bit confused about the terminology here because a contrast is applied to compare the levels of a single variable in a pre-specified way. For example, if you have 4 groups, comparing group A and B to group C and D, and then comparing group C to group D, that would be a planned contrast. In the context of the present paper, the term contrast seems to apply to just the main effect of a single variable (with only two levels). There is no need to specify all this. A three- or four-way ANOVA automatically gives the effect of each variable, and given that for most variables there are only two levels, this effect automatically tests if these two levels differ. No contrast necessary.

Only age has four groups so here a contrast could theoretically be applied, but this is not specified.

You do need to explain in the analysis plan which interactions you are going to test and how you are planning to proceed if the interaction is significant.

It is a common procedure if one interaction is significant, to decompose this interaction by going back one level, so if an interaction of A*B is significant you test the simple effect of B for each level of A. Or if A*B*C is significant, test interaction A*B at all levels of C.

p.13 why include an analysis with so many interaction terms if you ignore the results of those interactions anyway? You only look at one main effect. Note that many of the interactions are actually significant, meaning you can’t interpret a main effect, but you have to proceed with caution. Significant higher order interactions mean that the pattern is more complicated than any of the lower order interactions (or main effects) can capture, so their results can’t be interpreted isolated from the rest.

It would make more sense to decompose this significant four-way interaction by a series of follow up analyses: first three-way interactions, then two-way, as mentioned above. Along the way this automatically captures all the intended other effects (of language, number size, type of task and age).

A11: We thank the reviewer these detailed suggestions regarding the statistical analyses. Since we have completely re-analyzed the dataset with linear mixed models, we have directly implemented the suggestions in the newly written result section (pages 12 to 19), were we compared the different interactions. 

C12: Then some inconsistencies like in line 332 ‘Finally, the triple interaction between age group, language and number size was significant for RT and CR’.

But the discussion says (line 556): ‘The results reveal that the cost entailed by processing base-20 numbers is independent of age and the associated degree of language.’

But it’s not independent of age, it becomes weaker over time, that’s what the three-way interaction showed. The cost did not disappear entirely but it did wane.

A12: We thank the reviewer for noticing these inconsistencies between results and discussion. We based the interpretation that the transparency of power is independent of age mainly on the z-score analyses, since the difference among age groups might be due to higher inter-individual variability, which might be considered as “noise in the signal” (i.e. the high standard deviation in Grade 5). These variabilities might be due to developmental differences (i.e. in processing speed) rather than cognitive processing differences. While after z-score transformation, the interaction with age was not present. We added a sentence in the discussion to clarify this point (lines 505-507).

Yet, when standardizing age groups variability through z-score transformations, we noted that numbers above 70 were similarly slower to transcode in all age groups, arguing against a lexicalization with training.

C13: Line 670 ‘Our results have practical implications’, this looks forced. What follows is very general ‘linguistic factors have to be taken into account when instructing and evaluating children’. That may be true but it doesn’t follow from this study, it may be more relevant then to look at international comparisons: does Luxembourg drop in rankings after children switch to French? Then I’d see the issue, but in this study, how bad is it if someone is a few milliseconds slower in writing down a certain number word? And if it is bad, how can a teacher take this into account when instructing and evaluating children and aren’t they doing that already? Please specify the practical implications of this particular study (or argue that this is a fundamental study and as such there are no practical implications, which is perfectly acceptable by the way. Not every study has practical implications).

C13: We agree with the reviewer’s suggestion. As already mentioned in A1, we reoriented the interpretation to a more fundamental than applied research direction. Practically, we removed the following paragraph we had written having in mind the educational situation in Luxembourg, where the costs have also been found for arithmetic (Van Rinsveld et al., 2015).

Our results have practical implications, especially when aiming to provide equal opportunities for all students. Since linguistic factors can impact mathematics, these have to be taken into account when instructing and evaluating children. Considering that mathematical learning is cumulative, meaning that solid bases of previous knowledge are necessary to understand more elaborate concepts in mathematics, the impact of language on numerical and mathematical learning needs to be considered as early as possible [76]. Taken together the present results confirm and extend the interactions between language and number processing observed in more complex tasks such as arithmetic problem solving, corroborating the role of language in numerical cognition.

Minor points

C13.1: p. 3 line 47-48: ‘The simplest declination corresponding to bilingualism’. What is declination supposed to mean here? Is it form perhaps?

A13.1: We thank the reviewer for this suggestion. We changed the following lines accordingly: (line 78-80) 

When investigating how the acquisition of several languages (and hence multiple number words) develops, the many forms and constellations of multilingualism must be taken into account; with bilingualism corresponding to the simplest instance. 

C13.2: p. 4 line 80 unusual use of ‘peers’, peers are children from the same environment, commonly classmates in a school context, not children from different countries. 

A13.2: We thank the reviewer for this suggestion. We changed the following lines accordingly: (line 108-109) 

In a study comparing French-speaking 1st graders from Wallonia and France with a number dictation task, more mistakes were committed in the `70s and `90s by the latter (Seron & Fayol, 1994). 

C13.3: P. 6 line 121 what is LM-? The language in which mathematics was not learned? Please define.

A13.3: We thank the reviewer for noticing that this term definition was not complete. Hence we tried to be more clear about the definition of LM- (lines 146 to 151) 

The LSC generalizes to more ecological learning settings, showing that mathematical problems are solved more accurately in LM+, even if it is not the dominant language (i.e. the LM+ not being the same language as the L1). For example, Bernardo (Bernardo, 2001) investigated arithmetic among high school Filipino-English bilinguals who have Filipino as L1 but specifically learned mathematics in English (LM+). The results indicate a cost for arithmetical problems written in number words in Filipino (i.e. being the L1 but LM-) compared to English (i.e. being an L2 but LM+), which in turn showed comparable results than for problems presented as Arabic digits. 

C13.4: Correct terminology is two-way or three-way interaction, not double or triple.

A13.4: We thank the reviewer for noticing this terminological imprecision. We have substituted the terminology in the present paper (i.e. lines 312 and 331).

C13.5: Line 659: ‘It is, however, noteworthy that math education extended for one year longer in LM2 (7 years of primary education) than in LM1 (6 years of secondary education), which might help to balance the frequencies in both languages.’ This only makes sense if the words primary and secondary were reversed. But it may help to explain the Luxembourgish situation in a bit more detail in the introduction.

A13.5: We thank the reviewer for noticing this inversion. We have corrected it in the following lines (lines 578 to 580)

It is, however, noteworthy that math education extended for one year longer in LM2 (7 years of primarysecondary education) than in LM1 (6 years of secondaryprimary education), which might help to balance the frequencies in both languages.

 

Annex:

In the following additional analyses with linear mixed models, we compared all decades. That is, in the following, the factor Number Size, has 8 levels: ‘30s, ‘40s, ‘50s, ‘60s, ‘70s, ‘80s, ‘90s. Otherwise, the analysis remains exactly the same as in the main manuscript. 

Reading aloud task:

 num Df den Df F Pr(>F)

Age 3 94.51 17.15 < 0.001

Language 1 21.21 682.32 <0.001

Number Size 6 21.40 6.35 <0.001

Age x Language 3 2320.04 111.10 <0.001

Age x Number Size 18 2316.98 2.07 0.005

Language x Number Size 6 21.02 12.54 < 0.001

Age x Language x Number Size 18 2316.46 1.09 0.355

Note: Number Size has 8 levels: ‘30s, ‘40s, ‘50s, ‘60s, ‘70s, ‘80s, ‘90s.

The model converged and the interaction between Language and Number Size was significant (F (6,21.02) = 12.54, p < .001).

“Stepwise” contrasts:

The following custom contrasts compare each decade with the previous and following one. For example ‘30 vs ‘40, ‘40 vs ‘50, etc. 

French

contrasts estimate SE z.ratio p.value

30 - 40 46.62 50 0.93 0.35

40 - 50 -52.09 50 -1.04 0.30

50 - 60 -75.68 49.8 -1.52 0.13

60 - 70 -194.7 51.1 -3.81 0.0001

70 - 80 -3.28 51.9 -0.06 0.95

80 - 90 99.97 50.9 1.96 0.05

German

contrast estimate SE z.ratio p.value

30 - 40 35.01 44.3 0.79 0.43

40 - 50 51.23 44.4 1.15 0.25

50 - 60 -18.19 44.5 -0.41 0.68

60 - 70 -38.15 44.5 -0.86 0.39

70 - 80 38.73 44.3 0.87 0.38

80 - 90 -60.04 44.2 -1.36 0.17

In French, ‘30s, ‘40s, ‘50s, ‘60s numbers have comparable RTs. However, ‘60s have significant faster RT than ‘70s, which are not different from ‘80s. Interestingly, ‘90s are (p = .05) almost slower than ‘80s. In German there are no differences among the decades (all p > .17).

Figure 1 Mean reaction time for each decade at each age groups. Ribbons represent one standard error.

Verbal-visual matching

 num Df den Df F Pr(>F)

Age 3 94.85 55.64 <0.001

Language 1 20.95 282.15 <0.001

Number Size 6 20.87 5.51 0.001

Age x Language 3 2317.05 67.15 <0.001

Age x Number Size 18 2315.26 2.75 <0.001

Language x Number Size 6 20.74 9.10 <0.001

Age x Language x Number Size 18 2314.64 1.59 0.054

The model converged and the interaction between Language and Number Size was significant (F (6,20.74) = 9.10, p < .001).

Figure 2: Mean reaction time for each decade at each age groups. Ribbons represent one standard error.

Stepwise” contrasts:

The following custom contrasts compare each decade with the previous and following one. For example ‘30 vs ‘40, ‘40 vs ‘50, etc. 

French

contras estimate SE z.ratio p.value

30 - 40 -154.04 84.6 -1.82 0.07

40 - 50 98.16 84.8 1.16 0.24

50 - 60 -127.58 85.6 -1.49 0.13

60 - 70 -222.20 86.6 -2.56 0.01

70 - 80 -105.11 87.4 -1.20 0.23

80 - 90 43.99 88.5 0.50 0.62

German:

contras estimate SE z.ratio p.value

30 - 40 -16.31 77.8 -0.21 0.83

40 - 50 14.37 77.7 0.18 0.85

50 - 60 -43.83 77.9 -0.56 0.57

60 - 70 61.03 77.9 0.78 0.43

70 - 80 -99.86 77.7 -1.28 0.20

80 - 90 0.391 77.8 0.005 0.99

In French, ‘30s, ‘40s, ‘50s, ‘60s numbers have comparable RTs. However, ‘60s have significant faster RT than ‘70s, which are not different from ‘80s. In German there are no differences among the decades (all p > .20).

 

References:

Avanzi, M., & Thibault, A. (2019). Documenter le système vigésimal et sa contrepartie décimale (70-80-90) en français et dans les dialectes galloromans. Langages, N°215(3), 89. https://doi.org/10.3917/lang.215.0089

Bahnmueller, J., Moeller, K., Mann, A., & Nuerk, H.-C. (2015). On the limits of language influences on numerical cognition – no inversion effects in three-digit number magnitude processing in adults. Frontiers in Psychology, 6. https://doi.org/10.3389/fpsyg.2015.01216

Bahnmueller, J., Nuerk, H.-C., & Moeller, K. (2018). A Taxonomy Proposal for Types of Interactions of Language and Place-Value Processing in Multi-Digit Numbers. Frontiers in Psychology, 9. https://doi.org/10.3389/fpsyg.2018.01024

Barrouillet, P., Camos, V., Perruchet, P., & Seron, X. (2004). ADAPT: A Developmental, Asemantic, and Procedural Model for Transcoding From Verbal to Arabic Numerals. Psychological Review, 111(2), 368–394. https://doi.org/10.1037/0033-295X.111.2.368

Bernardo, A. B. I. (2001). Asymmetric activation of number codes in bilinguals: Further evidence for the encoding complex model of number processing. Memory & Cognition, 29(7), 968–976. https://doi.org/10.3758/BF03195759

Camos, V. (2008). Low working memory capacity impedes both efficiency and learning of number transcoding in children. Journal of Experimental Child Psychology, 99(1), 37–57. https://doi.org/10.1016/j.jecp.2007.06.006

Campbell, J. (1995). Mechanisms of Simple Addition and Multiplication: A Modified Network-interference Theory and Simulation. Mathematical Cognition, 1, 121–164.

Chrisomalis, S. (2010). Numerical Notation: A Comparative History. Cambridge University Press.

Clayton, F. J., Copper, C., Steiner, A. F., Banfi, C., Finke, S., Landerl, K., & Göbel, S. M. (2020). Two-digit number writing and arithmetic in Year 1 children: Does number word inversion matter? Cognitive Development, 56, 100967. https://doi.org/10.1016/j.cogdev.2020.100967

De Visscher, A., & Noël, M.-P. (2014). The detrimental effect of interference in multiplication facts storing: Typical development and individual differences. Journal of Experimental Psychology: General, 143(6), 2380–2400. https://doi.org/10.1037/xge0000029

Dehaene, S. (1992). Varieties of numerical abilities. Cognition, 44(1), 1–42. https://doi.org/10.1016/0010-0277(92)90049-N

Dehaene, S., Piazza, M., Pinel, P., & Cohen, L. (2003). Three parietal circuits for number processing. Cognitive Neuropsychology, 20(3–6), 487–506. https://doi.org/10.1080/02643290244000239

Deloche, G., & Seron, X. (1982). From three to 3: A differential analysis of skills in transcoding quantities between patients with Broca’s and Wernicke’s aphasia. Brain, 105(4), 719–733. https://doi.org/10.1093/brain/105.4.719

Dijkstra, T., & van Heuven, W. J. B. (2002). The architecture of the bilingual word recognition system: From identification to decision. Bilingualism: Language and Cognition, 5(3), 175–197. https://doi.org/10.1017/S1366728902003012

Dotan, D., & Friedmann, N. (2018). A cognitive model for multidigit number reading: Inferences from individuals with selective impairments. Cortex, 101, 249–281. https://doi.org/10.1016/j.cortex.2017.10.025

Dowker, A., & Nuerk, H.-C. (2016). Editorial: Linguistic Influences on Mathematics. Frontiers in Psychology, 7. https://doi.org/10.3389/fpsyg.2016.01035

Ellis, N. C., & Hennelly, R. A. (1980). A bilingual word-length effect: Implications for intelligence testing and the relative ease of mental calculation in Welsh and English. British Journal of Psychology, 71(1), 43–51. https://doi.org/10.1111/j.2044-8295.1980.tb02728.x

Friso-van den Bos, I., van der Ven, S. H. G., Kroesbergen, E. H., & van Luit, J. E. H. (2013). Working memory and mathematics in primary school children: A meta-analysis. Educational Research Review, 10, 29–44. https://doi.org/10.1016/j.edurev.2013.05.003

Gathercole, S. E., & Baddeley, A. D. (1993). Phonological working memory: A critical building block for reading development and vocabulary acquisition? European Journal of Psychology of Education, 8(3), 259. https://doi.org/10.1007/BF03174081

Geary, D. C., Bow‐Thomas, C. C., Liu, F., & Siegler, R. S. (1996). Development of Arithmetical Competencies in Chinese and American Children: Influence of Age, Language, and Schooling. Child Development, 67(5), 2022–2044. https://doi.org/10.1111/j.1467-8624.1996.tb01841.x

Göbel, S. M., Moeller, K., Pixner, S., Kaufmann, L., & Nuerk, H.-C. (2014). Language affects symbolic arithmetic in children: The case of number word inversion. Journal of Experimental Child Psychology, 119, 17–25. https://doi.org/10.1016/j.jecp.2013.10.001

Gollan, T. H., Montoya, R. I., Fennema-Notestine, C., & Morris, S. K. (2005). Bilingualism affects picture naming but not picture classification. Memory & Cognition, 33(7), 1220–1234. https://doi.org/10.3758/BF03193224

Grosjean, F. (2010). Bilingual: Life and reality. Harvard University Press.

Haspelmath, M., Dryer, M. S., Gil, D., & Comrie, B. (2005). The World Atlas of Language Structures. Oxford Univ. Press.

Ifrah, G., & Bellos, D. (2000). The universal history of numbers: From prehistory to the invention of the computer. Wiley.

Imbo, I., Vanden Bulcke, C., De Brauwer, J., & Fias, W. (2014). Sixty-four or four-and-sixty? The influence of language and working memory on children’s number transcoding. Frontiers in Psychology, 5. https://doi.org/10.3389/fpsyg.2014.00313

Ivanova, I., & Costa, A. (2008). Does bilingualism hamper lexical access in speech production? Acta Psychologica, 127(2), 277–288. https://doi.org/10.1016/j.actpsy.2007.06.003

Krinzinger, H., Gregoire, J., Desoete, A., Kaufmann, L., Nuerk, H.-C., & Willmes, K. (2011). Differential Language Effects on Numerical Skills in Second Grade. Journal of Cross-Cultural Psychology, 42(4), 614–629. https://doi.org/10.1177/0022022111406252

Lê, M.-L. T., & Noël, M.-P. (2020). Transparent number-naming system gives only limited advantage for preschooler’s numerical development: Comparisons of Vietnamese and French-speaking children. PLOS ONE, 15(12), e0243472. https://doi.org/10.1371/journal.pone.0243472

Lonnemann, J., & Yan, S. (2015). Does number word inversion affect arithmetic processes in adults? Trends in Neuroscience and Education, 4(1), 1–5. https://doi.org/10.1016/j.tine.2015.01.002

McCloskey, M. (1992). Cognitive mechanisms in numerical processing: Evidence from acquired dyscalculia. Cognition, 44(1–2), 107–157. https://doi.org/10.1016/0010-0277(92)90052-J

McCloskey, M., Caramazza, A., & Basili, A. (1985). Cognitive mechanisms in number processing and calculation: Evidence from dyscalculia. Brain and Cognition, 4(2), 171–196. https://doi.org/10.1016/0278-2626(85)90069-7

McClung, N. A., & Arya, D. J. (2018). Individual Differences in Fourth-Grade Math Achievement in Chinese and English. Frontiers in Education, 3, 29. https://doi.org/10.3389/feduc.2018.00029

Meeuwissen, M., Roelofs, A., & Levelt, W. J. M. (2003). Planning levels in naming and reading complex numerals. Memory & Cognition, 31(8), 1238–1248. https://doi.org/10.3758/BF03195807

Miller, K. F., Smith, C. M., Zhu, J., & Zhang, H. (1995). Preschool Origins of Cross-National Differences in Mathematical Competence: The Role of Number-Naming Systems. Psychological Science, 6(1), 56–60.

Miller, K. F., & Stigler, J. W. (1987). Counting in Chinese: Cultural variation in a basic cognitive skill. Cognitive Development, 2(3), 279–305. https://doi.org/10.1016/S0885-2014(87)90091-8

Miura, I. T., Kim, C. C., Chang, C.-M., & Okamoto, Y. (1988). Effects of Language Characteristics on Children’s Cognitive Representation of Number: Cross-National Comparisons. Child Development, 59(6), 1445. https://doi.org/10.2307/1130659

Moeller, K., Zuber, J., Olsen, N., Nuerk, H.-C., & Willmes, K. (2015). Intransparent German number words complicate transcoding – a translingual comparison with Japanese. Frontiers in Psychology, 6. https://doi.org/10.3389/fpsyg.2015.00740

Nuerk, H., Weger, U., & Willmes, K. (2005). Language effects in magnitude comparison: Small, but not irrelevant. Brain and Language, 92(3), 262–277. https://doi.org/10.1016/j.bandl.2004.06.107

Nuerk, H.-C., Weger, U., & Willmes, K. (2004). On the Perceptual Generality of the Unit-Decade Compatibility Effect. Experimental Psychology, 51(1), 72–79. https://doi.org/10.1027/1618-3169.51.1.72

Pixner, S., Zuber, J., Heřmanová, V., Kaufmann, L., Nuerk, H.-C., & Moeller, K. (2011). One language, two number-word systems and many problems: Numerical cognition in the Czech language. Research in Developmental Disabilities, 32(6), 2683–2689. https://doi.org/10.1016/j.ridd.2011.06.004

Poncin, A., Van Rinsveld, A., & Schiltz, C. (2019). Units-first or tens-first: Does language matter when processing visually presented two-digit numbers? Quarterly Journal of Experimental Psychology, 73(5), 726–738. https://doi.org/10.1177/1747021819892165

Power, R. J. D., & Dal Martello, M. F. (1990). The dictation of Italian numerals. Language and Cognitive Processes, 5(3), 237–254. https://doi.org/10.1080/01690969008402106

Rodic, M., Zhou, X., Tikhomirova, T., Wei, W., Malykh, S., Ismatulina, V., Sabirova, E., Davidova, Y., Tosto, M. G., Lemelin, J.-P., & Kovas, Y. (2015). Cross-cultural investigation into cognitive underpinnings of individual differences in early arithmetic. Developmental Science, 18(1), 165–174. https://doi.org/10.1111/desc.12204

Salillas, E., & Martínez, A. (2018). Linguistic Traces in Core Numerical Knowledge: An Approach From Bilingualism. In Language and Culture in Mathematical Cognition (pp. 173–196). Elsevier. https://doi.org/10.1016/B978-0-12-812574-8.00008-0

Salillas, E., & Wicha, N. Y. Y. (2012). Early Learning Shapes the Memory Networks for Arithmetic: Evidence From Brain Potentials in Bilinguals. Psychological Science, 23(7), 745–755. https://doi.org/10.1177/0956797612446347

Seron, X., & Fayol, M. (1994). Number transcoding in children: A functional analysis. British Journal of Developmental Psychology, 12(3), 281–300. https://doi.org/10.1111/j.2044-835X.1994.tb00635.x

Steiner, A. F., Banfi, C., Finke, S., Kemény, F., Clayton, F. J., Göbel, S. M., & Landerl, K. (2021). Twenty-four or four-and-twenty: Language modulates cross-modal matching for multidigit numbers in children and adults. Journal of Experimental Child Psychology, 202, 104970. https://doi.org/10.1016/j.jecp.2020.104970

Sullivan, M. D., Poarch, G. J., & Bialystok, E. (2018). Why is Lexical Retrieval Slower for Bilinguals? Evidence from Picture Naming. Bilingualism (Cambridge, England), 21(3), 479–488. https://doi.org/10.1017/S1366728917000694

Van Rinsveld, A., Brunner, M., Landerl, K., Schiltz, C., & Ugen, S. (2015). The relation between language and arithmetic in bilinguals: Insights from different stages of language acquisition. Frontiers in Psychology, 6. https://doi.org/10.3389/fpsyg.2015.00265

Van Rinsveld, A., Dricot, L., Guillaume, M., Rossion, B., & Schiltz, C. (2017). Mental arithmetic in the bilingual brain: Language matters. Neuropsychologia, 101, 17–29. https://doi.org/10.1016/j.neuropsychologia.2017.05.009

Van Rinsveld, A., & Schiltz, C. (2016). Sixty-twelve = Seventy-two? A cross-linguistic comparison of children’s number transcoding. British Journal of Developmental Psychology, 34(3), 461–468. https://doi.org/10.1111/bjdp.12151

Vander Beken, H., & Brysbaert, M. (2018). Studying texts in a second language: The importance of test type. Bilingualism: Language and Cognition, 21(5), 1062–1074. https://doi.org/10.1017/S1366728917000189

Volmer, E., Grabner, R. H., & Saalbach, H. (2018). Language switching costs in bilingual mathematics learning: Transfer effects and individual differences. Zeitschrift Für Erziehungswissenschaft, 21(1), 71–96. https://doi.org/10.1007/s11618-017-0795-6

Wicha, N. Y., Dickson, D. S., & Martinez-Lincoln, A. (2018). Arithmetic in the Bilingual Brain. In Language and Culture in Mathematical Cognition (pp. 145–172). Elsevier. https://doi.org/10.1016/B978-0-12-812574-8.00007-9

Xenidou-Dervou, I., Gilmore, C., van der Schoot, M., & van Lieshout, E. C. D. M. (2015). The developmental onset of symbolic approximation: Beyond nonsymbolic representations, the language of numbers matters. Frontiers in Psychology, 6. https://doi.org/10.3389/fpsyg.2015.00487

Zuber, J., Pixner, S., Moeller, K., & Nuerk, H.-C. (2009). On the language specificity of basic number processing: Transcoding in a language with inversion and its relation to working memory capacity. Journal of Experimental Child Psychology, 102(1), 60–77. https://doi.org/10.1016/j.jecp.2008.04.003

---

## [Decision Letter · Decision Letter 1]

4 Jan 2022

PONE-D-21-18244R1Number transcoding in bilinguals - a transversal developmental studyPLOS ONE

Dear Dr. Lachelin,

Thank you for submitting your revised manuscript to PLOS ONE. I have sent it back to the two expert reviewers who evaluated the first version. As you will see below, both reviewers are quite positive about the changes you made to the manuscript and think that it has considerably improved. I agree with them. Both reviewers, however, suggest a number of additional changes that may improve the manuscript further. I will not detail here the different points as the reviewers are very thorough in their evaluations. But I do encourage you to take into account these suggestions in another round of revision.

We look forward to receiving your revised manuscript.

Kind regards,

Jérôme Prado

Academic Editor

PLOS ONE

Reviewers' comments:

Reviewer #1: 1. The authors added a discussion of theoretical models, following my request. This is big improvement. In the discussion section, the theoretical relation with the model is good in my opinion (see only minor comments below). In contrast, the presentation of models in the introduction still feels a bit out-of-context. I didn’t expect a general summary of theoretical models of number transcoding, but a more concrete explanation of how these issues are related with the present study.

2. Lines 104-105: why are 70’s a particular case?

3. Line 107: I did not understand the Belgian Wallonia dialect, perhaps elaborate. The 80’s are same as in France-French, but what’s the difference? Do they say septante and nonante?

4. Lines 268-269, line 341: was z-scoring done separately for each age group?

5. Fig. 1 seems to me unnecessary. It could be easily explained in the text. For your consideration.

6. Lines 296-301: you wrote precisely which conditions will be compared, this is excellent, but please write also the specific predictions. Don’t leave it to the reader to guess.

7. The model description in lines 304-310 is confusing. You talk about what you did and what you intended to, going back and forth between the two. Perhaps just write what your model actually was. Then, if you want, you can tell about additional things you tried but failed (personally I feel this is neither needed nor common, but it’s up to you). This should be an extremely simple paragraph to read, but I found myself having to re-read it.

Similar issues in lines 346-351.

8. The statistics is not sufficiently ordered / detailed. I will not go through each analysis, because there are many of them. I only provide examples here, but please fix these issues in all analyses, not just the examples I give here:

- Define the LMM variables fully and explicitly. E.g. was age a categorical or numerical factor? If categorical, specify the levels. Were Subject and Item defined as random factors (they should have been)? Which variables were within/between subjects?

- Write statistical results either in a table or in the text, no need for both. For example lines 312-319 vs. Table 1. Lines 353-356 can be simply deleted.

- When talking about follow-up analyses, write what the statistical test was and its result. E.g. line 314, lines 326-327, and more (which statistical test does the z stand for? You use it a lot, perhaps you wrote somewhere in methods but I can’t remember all the details).

- When the variable had 2 possible values, or for 2 x 2 interactions, describe the direction of the results in the text (e.g. lines 312-319, effects of Number Size and Size-Language interaction, and the info is missing in several other places too).

- For each analysis, I’d recommend presenting the interesting effect first, and the unimportant/trivial/boring effects later. You sometimes start with the boring one, and this is… well… a bit boring.

- Linear mixed models: how was the p-value computed? There are number of ways to do this in LMM, explain which one was used.

- Perhaps also explain Table 1, 3rd column. Degrees of freedom are typically integer. I know that in some cases they may be floating point, but it may be a good idea to say something about why this happened or what they mean.

- Use standard terminology to describe the LMM. E.g., say random factor (with or without random slopes) instead of other terminology, e.g. in lines 375-376, which I admit I did not understand.

- When relevant (e.g. in follow-up analyses), specify whether the p values are one-tailed or two-tailed.

9. Line 343-344: interesting and important point. But please explain why this is important.

10. Line 302: I assume that when analyzing reaction times, you removed the incorrect-response trials? This should be the case, as incorrect trials cannot be expected to exhibit the same RTs. Same issue in the matching task.

11. Lines 397-403 – the statistics is not written clearly, and I suspect it’s also not optimal. You write that the hypothesis on language of acquisition was not supported, but I was not convinced this is correct. I suspect that if you analyze all age groups together, of course limiting to numbers < 60, you will get a significant effect.

Certainly, I wouldn’t present it as if the effect existed only in 5th grade and nothing at all for older ages (line 439). You have very nice data, no reason to undermine it.

This is also a more-general issue about follow-up analyses: you don’t have to run t-test (or whatever similar test it was) on small portions of the data. Why not just run LMM on the relevant subset of data?

12. The detailed analyses in the annex are interesting. I think it would be nice if mentioned them very briefly in the main text – perhaps just say the main conclusions and reference the annex. When I read the previous manuscript version I was worried about these issues, other readers may feel the same and you have good answers. Also, some of the things you wrote in the response to review, which were correct and important in my opinion, may be added to the annex (currently it’s stats and data, without much conclusions).

E.g., lines 507-509 may be a relevant place to mention this.

13. Line 492 claims that number matching is transcoding from verbal to visual. This is incorrect. You can’t know which transcoding direction the participants use in a number matching task.

14. Please write in supplementary material files which task they refer to.

15. Line 493-494: the finding that the transparency of power effect on z-scored RTs remains stable across age is interesting. However, you did not say which data supports it. In my opinion, the critical analysis that supports it is the absence of triple interaction in the z-scored data. So if you want to reach this conclusion, you should mention the triple interaction issue in the results section (maybe even move this z-scored analyses into the main text).

16. Line 497-515: I think you propose an interesting idea – that the difficulty in 70-99 may arise either from syntactic complexity or from morphological-retrieval complexity of these numbers. However, I was not convinced by your claim that the former interpretation necessarily relates to ADAPT and the latter necessarily relates to Dotan & Friedmann. I think that both models can accommodate both interpretations.

17. The idea of proactive interference (line 515) seems far-fetched. Retrieving 4 morphemes instead of 2 may be difficult for several different reasons, I am not sure I see a reason to point specifically at PI. In any case, Campbell’s and De Visscher’s studies on multiplication talk about a scenario quite different – not interference of single words because they are numerous, but of multi-word multiplication exercise because they are similar. The analogy seems to me more misleading than illuminating.

18. Line 543: presenting number reading as a form of free recall, or the matching task as a form of familiarity judgment, seems far-fetched and perhaps even misleading. I totally agree with you that the tasks have some resemblance, in the sense that the former two require unlimited retrieval, whereas the latter two require verification (lines 544-548). But you can say just that, no need to imply that the tasks are more identical than they really are.

19. Lines 576+: another limitation may be that it’s hard to disentangle language of math acquisition from an alternative interpretation that assumes intrinsic differences between German and French (“French is just harder”).

Reviewer #2: I read a revised version of the manuscript and first I applaud the authors for the endeavor to greatly rewrite the work including a complete change of the analyses. That is a large step that undoubtedly took quite some time but it is definitely a large improvement.

I noted that reviewer 1 is much more knowledgeable than me on the theoretical details. It looks like that improved a lot too but reviewer 1 is probably much better able than me to judge that (if also rereviewing the manuscript).

I do still have some remaining questions, mostly about the analyses.

p. 4-5 for the reader unfamiliar with those languages it would be helpful to give an example in words, not just in formula of 10*x+y (and also define x and y). You do this for French now but not for German (I’d also give the example both in the original language and its counterpart in English).

I also doubt if ‘transparency of power’ is the best term. It sounds nice/interesting but it’s only the names of the tens that are affected, not the hundreds or thousands (admittedly, it returns for tens of thousands, but still). So isn’t it just ‘transparency of tens/decades’?

Line 173 ‘These results support the above-mentioned TCM stating that precise numbers are encoded in a language dependent format [63].’ I was under the impression that the results mentioned above all related to studies that offered math in a verbal format to find differences in language of presentation. For instance, the text speaks of for instance ‘problems presented in the LM+’. But if this is the case, the conclusion above cannot be drawn. The use of a verbal context means we can’t know if the precise numbers (by the way, does ‘precise numbers’ refer to exact quantities or symbolic numbers?) are stored in a language-dependent or -independent format.

Line 177 ‘However, they agree with recent reports from the bilingualism literature that academic knowledge acquired in one language is not (or difficultly) transferable to the other language [55].’ That is a bold statement! That basically says I can’t use the knowledge I learn in English in my native language? Luckily this is not true. This study says that there is a switch cost between languages in math problem solving. A switch cost in mathematics is quite far away from this conclusion that in general academic knowledge learned in one language is not transferable to the other language!

Together it would be nice to see a better paragraph explaining the present research question to earlier knowledge. What is it that we don’t know and how does study contribute to filling that gap?

Line 212: The two tasks might thus reveal a somewhat different result pattern, as already observed in Van Rinsveld et al. [69]. Please elaborate on how.

Present study: I said this before, when describing the population also describe the mother tongue of the children and a bit more of the language background in Luxembourgh besides language in the classroom.

Specifically it is relevant to know and possibly not widely known by the audience that Luxembourgish strongly resembles German. So one may argue that for the 70 German- and Luxembourgish speaking children, LM1 and L1 are (virtually) identical in terms of number words, whereas for the other 27 this isn’t the case. That means there are two linguistically very diverse groups of participants. It would at least be relevant to know at what age the immigrant children had started learning German and French.

This information is also relevant because in the way the study is framed now, the youngest age group wouldn’t be able to do the task in French at all because they wouldn’t know it yet. But they do. But how well? And to what degree can all results be explained by just very poor proficiency in one group that hasn’t properly been taught French at all?

Line 265. There are 2 sets of 5 practice and 14 test stimuli but the experiment lasted 45 minutes. That sounds extremely long for this task. If this test was part of a larger test battery please mention the context.

Line 242 still makes the cut at 60 instead of 70. (there are no 14 numbers above and 14 below 60 in the set by the way. There are 4 from each decade). Also explain this decision to leave out the 60 numbers earlier than the results section (it’s not a result, it’s a methodological choice).

Line 285. Have you considered post-error slowing (the ‘oops’ effect)? It may be wise to remove RTs after an error too.

Could you give some descriptive statistics? Like M and SD of RT/accuracy per task, broken down by the relevant factors, so by age group, language and number size.

306 we aimed to model the individual variability, ideally considering individual differences (i.e. Subject) and age group (i.e. Age) related differences. Furthermore, we aimed to model per-item variability (i.e. Item), particularly related to language.

I’m a bit confused here on what exactly you did especially with age group. Is this about random intercepts or random slopes? Slopes for what? And you modeled age group as a fixed effect already so you can’t add a random effect for it too, so it makes sense that that didn’t work.

So what did the model syntax look like exactly?

It sounds like it was something like RT~Age*Language*Number Size + (1|Subject) + (1+language|Item)?

Or also (1+language|Subject)? If not, why no random slope for language per subject given that you say you run a maximal model?

Nd could you tell which method you used to obtain p-values? Conditional F-tests with Kenward-Roger or Satterthwaite corrected degrees of freedom probably?

Lines 314, 325 and also in later sections. Specify how you did your follow-up analyses when applicable (and what were the resulting statistics)? Explain what kind of analysis (the same?) on what kind of subset of the data. I see no more F tests but z-scores, if these are Wald tests from mixed effects models, those aren’t very reliable.

Line 347 and beyond, what do you mean that the model did not fit? Do you mean that it did not converge? Why specifically drop age here, it seems the most relevant factor? I also learned that dropping a random slope is the very very last thing one should try as it greatly reduces p-values (so it inflates significant findings). https://academic.oup.com/esr/article/35/2/258/5306121. Now non-convergence can be a beast but other solutions to solve it do exist. This is what I learned to try, in this order:

– Most recent package versions (particularly lme4)?

– Center instead of standardize (or vice versa) cont. predictors

– Increase number of iterations

– Restart model fitting from previous (unconverged) estimates

– Different optimizer(s); compare estimates from many different optimizers; (compare to) brms model (Bayesian model)

- Simplify your model, Barr et al.: remove: (a) random correlations (b) random intercepts (c) random slopes; last resort, try to avoid that!

By the way, alternatively I could also imagine leaving accuracy out altogether because of its ceiling effects.

Line 358 in order to explain the interaction language * number size you give the results for French but also give the contrasting results for German (I guess no sig difference? But I shouldn’t need to guess.)? You need them both to understand the interaction.

p. 17 I’m a bit surprised that the language difference is so weak given the highly significant effect in table 3 and also the large differences in the figures. Is this correct?

The analyses with only 4 syllable words made me wonder how to count syllables in French reliably given that so many of them are silent. If I look it up there seems to be consensus that silent e’s at the end of a word don’t count as syllables meaning that many words are one syllable shorter.

Concerning this language difference in the verbal to visual task: isn’t German also easier in the sense of at which point you can already eliminate multiple choice options on the go? If you hear ‘vierund…’ you can already exclude the +1/-1 and the +11/-11 answers, leaving only two possible options halfway through the word. For French, specifically for the large numbers, hearing ‘quatrevingt…’ may still leave 3 or 4 options open halfway through the word.

Minor points

Line 151. comparable results than for problems presented  comparable results to problems presented. Also in line 194.

Line 161/ who had either learned arithmetic in English or Spanish  who had learned arithmetic in either English or Spanish.

Line 227 switches to French (LM2) after the 7th grade  after 6th grade or in 7th grade.

Line 358 says ‘40s and ‘40s instead of ‘50s.

Quite some typos in the supplementary table with written out number words like ‘soixantetroi’, ‘siebenndneunzig’, etc.

---

## [Author Response · Author response to Decision Letter 1]

8 Feb 2022

Dear Editor, Prof. Dr Prado, 

dear Reviewer,

We would like to thank the editor and the two anonymous reviewers for their very valuable comments and their constructive suggestions on this second revision. 

We identified two major points which needed further details or adjustements: (1) the introduction of the cognitive number transcoding models, and (2) the description of the results. 

In this second revision of the manuscript we have addressed these concerns in depth and believe the manuscript’s quality has yet again significantly improved as a consequence. In the following, we describe how we implemented the different points raised by both reviewers into the revised manuscript. 

Below you can find our answers (A) to the comments (C) to each of the two reviewer’s points. Please note that adjustments in the manuscript are indicated in this colour code.

Reviewer #1:

C.1The authors added a discussion of theoretical models, following my request. This is big improvement. In the discussion section, the theoretical relation with the model is good in my opinion (see only minor comments below). In contrast, the presentation of models in the introduction still feels a bit out-of-context. I didn’t expect a general summary of theoretical models of number transcoding, but a more concrete explanation of how these issues are related with the present study.

A1. We thank the reviewer for this comment. According to the suggestion that the models felt out of context, we have added several sentences to link the different models with the topics of transcoding, multilingualism, and second language acquisition. We have also specified our hypothesis and linked it to the ADAPT model in a later paragraph. However, please note that the study was not designed to specifically test those models, hence most of the matches between our study and the models' predictions are made a posteriori, (l. 51 to 79). Note that we have removed the description of the different brain parts of the TCM to avoid a lengthy paragraph.

Several cognitive models have been proposed to describe transcoding. While some are taking also into account cognitive development, most of these models do not specifically account for transcoding differences between languages and multilingualism. These models are summarized below in two main categories: semantic and asemantic (Barrouillet et al., 2004).

In semantic models, transcoding requires an obligatory access of the number’s magnitude. For example McCloskey (McCloskey, 1992; McCloskey et al., 1985) proposed a transcoding model in which the entry number - regardless from the input’s format - accesses an abstract semantic representation. Power and Dal Martello (Power & Dal Martello, 1990) further proposed a model specifically for number dictation (i.e. verbal to Arabic format) which differs from the previous one in that semantic representations reflect the verbal word structure of the numbers, thus predicting differences between languages. Semantic models hence assume that transcoding difficulties depend on the quality and maturity of the semantic representations, therefore predicting worse performances in children, in particular for larger numbers.

Asemantic models propose that numbers can be transcoded without accessing magnitude. Deloche and Seron (Deloche & Seron, 1982) proposed such a model for number naming (i.e. Arabic to verbal) where the first step involves parsing the input into primitives which are then submitted to a set of rules for the output system. Later on, Barrouillet and colleagues (Barrouillet et al., 2004) proposed ADAPT (A Developmental Asemantic Procedural Model), explicitly including a developmental perspective on transcoding. In ADAPT, number words become lexicalized with training, hence directly retrieved from long-term memory and bypassing the procedural processes. Therefore transcoding in children would depend on language-dependent characteristics. Yet, the language-dependent characteristics should diminish with increasing lexicalization. Another more general model considering number representations is the Triple Code Model (TCM). The TCM proposes a functional and topographical framework of how and where in the brain approximate, visual symbolic and verbal codes are processed. Approximate processing of magnitudes (i.e. not exact quantities) are taken over by the horizontal segment of the intraparietal sulcus. Visual symbols, depending on visuospatial abilities, are processed by the parietal and fusiform gyrus (number form area). Finally, verbal codes, depending on linguistic processes, involve the left angular gyrus of the temporal-parietal gyrus (Dehaene et al., 2003). The TCM also implies asemantic language-dependent transcoding from or to a verbal code (Dehaene, 1992). Language-specific training in transcoding would therefore increase the strength of the association between each code and their respective brain areas. More recently, Dotan and Friedmann (Dotan & Friedmann, 2018) proposed a model for Arabic to verbal transcoding where numbers are first visually analysed for identity, order and decimal structure to build a language independent number word frame, which is passed to a second system which applies language-specific rules which are associated with their phonological and articulatory counter-parts. In sum, asemantic models predict that verbal transcoding depends on language characteristics. However, those models do not explicitly model multilingualism, nor how transcoding develops by acquiring number words in a second language.

See also lines (307-318) about the hypothesis, which we linked to the above-mentioned predictions, herein we also integrated C6 concerning more precise hypotheses:

For coherence between the models and the hypothesis, we also added the following sentence in the discussion about transparency of power (lines 513-516):

However, this interpretation can be excluded, because the same comparison in German did not reveal any differences. Assuming that 5th graders do not yet have mature magnitude representations this result also speaks against the predictions of semantic models.

C2 Lines 104-105: why are 70’s a particular case?

A 2. We thank the reviewer for this comment. The remarks on teens in 70’s French number words were added as an answer to a comment from reviewer 2 (see reviewer 2 first revision, C2.1 and A 2.1: “I’d say that the true base 20 starts at 80, as the 60’s follow the regular base 10 system and the 70’s are neither (as they are 60+10, 60+11 etc). And then there are the irregular number words for 11-16.”). However, after further reflections on the morpho-syntactic structure of French number words, it becomes clear to us that the considerations on teens also apply to ‘90s number words, which might have led to confusion. Yet, as described in the annexed analyses provided in the previous revision comparing each decade, we did not find differences between ’80s (without teens) and ‘70s or ‘90s with teens (lines 106-110)

However, `80s to `90s follow a base-20 system, e.g. 87 = 4*20 + 7, “quatre-vingt-sept”, literally “four-twenty-seven”. Moreover, the teens (11 to 16) are lexical primitives, like in English and in stark contrast to the more transparent Asian languages. Note that the transparency contrast is additionally increased for 71 to 76 and 91 to 96, which are composed with a base-20 and teens, e.g. 96 = 4*20 + 16, “quatre-vingt-seize”, “four-twenty-sixteen”.

C3 Line 107: I did not understand the Belgian Wallonia dialect, perhaps elaborate. The 80’s are same as in France-French, but what’s the difference? Do they say septante and nonante?

A.3. Yes in Belgian Wallonia, septante and nonante are commonly used (see Seron & Fayol, 1994). We adjusted the sentence to improve the clarity of that description of the french counting system and added emphasis with italic about the corresponding english translation (lines 110 to 113). (See also R2 C1).

Furthermore, the vigesimal system is subject to regional variances, for example in Belgian Wallonia `70s and ‘90s are not vigesimal (i.e. “septante” for “seventy” and “nonante” for “ninety”, see (Seron & Fayol, 1994)) and in some Swiss-French cantons the vigesimal system is entirely absent (i.e. including for 80, “huitante”, literally “eighty”).

C4 Lines 268-269, line 341: was z-scoring done separately for each age group?

A 4. We thank the reviewer for noticing this important point, yes it was done separately for each age group, we added the following sentence (line 278-279):

To control for age related variabilities analogous analyses were conducted on z-score transformed RT, separately for each group.

C5 Fig. 1 seems to me unnecessary. It could be easily explained in the text. For your consideration.

A 5. We added Fig.1 after noticing a problem on our side to describe the analyses design such that it is easily understood and we think the figure might add clarity about the design to the reader. We agree this could be a repetition of information, but the design rationale being fundamental to understanding the interpretation of the results, we decided to keep Fig. 1.

C6 Lines 296-301: you wrote precisely which conditions will be compared, this is excellent, but please write also the specific predictions. Don’t leave it to the reader to guess.

A.6. We agree that adding the predictions with the analysis designs would improve the reader’s understanding. We modified the paragraph adding more precise precise predictions (see lines 307-318): 

The analyses aimed at testing for all age groups the two main hypotheses: (1) the effect of transparency of power in French and (2) the effect of language of math acquisition. First, the effect of transparency of power in French, corresponding to the cost for vigesimal numbers in the base-20 system was tested by comparing small numbers (i.e. ’30s, ’40s, and ’50s) to large numbers (i.e. ’70s, ’80s, and ’90s) in French only (see A in Fig 1). For this hypothesis, we predicted that both tasks would lead to worse performances for large compared to small numbers, henceforth a cost. Developmentally, the transparency cost might either be re-absorbed with increasing training using LM2 number words or remain stable across age groups. Additional control for the potential confounder of a number size effect was implemented by applying the same comparison within German (A.1 in Fig 1). Second, the hypothesis concerning the language of math acquisition was tested by comparing only small numbers (i.e. ’30s, ’40s, and ’50s) in French and German (B in Fig 1). We predicted that both tasks would lead to a cost for French compared to German. Developmentally, this cost could remain stable throughout adolescence and adulthood, or gradually diminish with age, potentially even resorbing in adults.

C7 The model description in lines 304-310 is confusing. You talk about what you did and what you intended to, going back and forth between the two. Perhaps just write what your model actually was. Then, if you want, you can tell about additional things you tried but failed (personally I feel this is neither needed nor common, but it’s up to you). This should be an extremely simple paragraph to read, but I found myself having to re-read it. Similar issues in lines 346-351.

A.7. We thank the reviewer for this comment about clarity which is reflected also by the second reviewer comments (see R2 C12). To facilitate understanding we have completely written anew the model’s description, as follows (lines 321 -332):

The linear mixed model was defined with main effects and interactions between the fixed between-group factor Age (levels: 5th, 8th, 11th grade and adults) and two fixed within-group factors: Language (German, French), and Number Size (Large, Small), all levels being defined as categorical. As random factors, we modelled individual differences (i.e. Subject) and item-related variability (i.e. Item), the latter being considered separately by language. Individual differences were modelled using random intercepts per subject. Item-related variability was modelled using both random intercepts and slopes as a function of languages. In sum, the model takes the following in R syntax (A): 

 RT ~ 1 + Age*Language*Number Size + (1|Subject) + (1 + Language|Item).

P-values and degrees of freedom were obtained with the Satterthwaite approximation method by the comparison of the full model against the model without the effect (Singmann et al., 2020). Follow-ups were calculated by comparing estimated marginal means (EMM) obtained with the emmeans package (Lenth, 2021), and p-values were adjusted with the Bonferroni method.

We have changed also the model description of correct responses (lines 375-380):

Correct responses (CR) were analysed using a binomial approach and p-values estimated by the likelihood ratio test. Since applying the same model as for RT (see (A)) did not converge, we had to drop the random per-item factor and the fixed factor Age. In sum the model had the following syntax (B):

(B) CR ~ Language*Size + (1|Sujet)

The failure of convergence with the more complex model might be due to ceiling effects, as the task was very simple, particularly for older age groups, see Fig 4 and S1 Table 2.

C8 The statistics is not sufficiently ordered / detailed. I will not go through each analysis, because there are many of them. I only provide examples here, but please fix these issues in all analyses, not just the examples I give here:

C8b Define the LMM variables fully and explicitly. E.g. was age a categorical or numerical factor? If categorical, specify the levels. Were Subject and Item defined as random factors (they should have been)? Which variables were within/between subjects?

A8.a. We added the details of the levels, cfr the paragraph above (lines 321 -332). 

C8b Write statistical results either in a table or in the text, no need for both. For example lines 312-319 vs. Table 1. Lines 353-356 can be simply deleted.

A8.b. We thank the reviewer for this suggestion, in agreement with this suggestion, we removed the within text results to keep them exclusively in the tables. Consequently, we have adjusted some formulations. Note that, according to reviewer 2 suggestion (R2 C11), we have however added the descriptive statistics of the effects. For example (lines 334-342, but see also 382-390, 410-416 and 458-431).

The model for the reading aloud task RTs resulted in three significant main effects and three two-way interactions (see Table 1). The main effect of Age was decomposed with follow-up analyses showing that the 5th graders (1091(290) ms) were slower than the older groups, which had similar RT’s (8th 797(288) ms, 11th 780(247) ms and adults 828(320) ms; standard deviations in parenthesis). Furthermore, the main effect of Language indicated overall slower naming in French than in German, and the main effect of Number Size indicated slower naming for bigger than smaller numbers. Significant two-way interactions were also found between Age and Language, as well as Age and Number Size. Finally, critically for the hypothesis, the two-way interaction between Language and Number Size was significant, while the triple interaction was not significant, see Table 1 and Fig 2.

C8c When talking about follow-up analyses, write what the statistical test was and its result. E.g. line 314, lines 326-327, and more (which statistical test does the z stand for? You use it a lot, perhaps you wrote somewhere in methods but I can’t remember all the details).

A8.c. We have fixed this, the follow-up analyses were done by pairwise comparisons of the estimated marginal means of the models. The z stand for Walden z test which could be calculated since the degrees of freedom was defined as asymptotic. As underlined by the reviewer we have changed the parameters to calculate Satterthwaite corrected degrees of freedom (and added Bonferroni corrected p-values) to enable pairwise t-test comparisons with which the reader would be more familiar. Note that those changes did not modify the follow-up significancy patterns (lines 348-350, but also 300-302):

Follow-up analyses were performed on the model's estimated marginal means (EMMs) of the interaction's terms (see Singmann, 2021) with Satterthwaite estimation of degrees of freedom. Then EMM were compared with two-tailed pairwise tests with Bonferroni adjusted p-values.

See also changes at lines 422-423:

Follow-up analyses were applied comparing the estimated marginal means with paired comparisons on Satterthwaite corrected degrees of freedom and Bonferroni adjusted p-values

C8d. When the variable had 2 possible values, or for 2 x 2 interactions, describe the direction of the results in the text (e.g. lines 312-319, effects of Number Size and Size-Language interaction, and the info is missing in several other places too).

A8.d. We thank the reviewer for this comment. We have added the direction of the Number Size and the Language The Number-Size and Language interaction is described in the follow-up for the reading task. For the matching task, we do not describe the directions of the two-way interactions since there is a three-way interaction (and also given the analyses design in Fig. 1.), which in turn is described in the follow-up in the matching task. (see lines 337-342).

Furthermore, the main effect of Language indicated overall slower naming in French than in German, and the main effect of Number Size indicated slower naming for bigger than smaller numbers. Significant two-way interactions were also found between Age and Language and Age and Number Size. Finally, critically for the hypothesis, the two-way interaction between Language and Number Size was significant, while the three-way interaction was not significant, see Table 1 and Fig 2.

C8e. For each analysis, I’d recommend presenting the interesting effect first, and the unimportant/trivial/boring effects later. You sometimes start with the boring one, and this is… well… a bit boring.

A8.e. We thank the reviewer for the suggestion to change the order of presentation of the effects. The order might be a little boring to read, but on the other side, it is a standard order (main effects, two-way interactions, three-way, ect) which is now even more important to respect since the reader will need to rely on the table to find the respective statistical result (see C8.b.). With the current changes (i.e. only having the results in the tables) we hope the eventually attentive and hurried reader might jump directly jump to the follow-up or sum paragraphs, whose logic is explained visually in Fig.1. 

C8.f. Linear mixed models: how was the p-value computed? There are number of ways to do this in LMM, explain which one was used.

A.8.f. We thank the reviewer for this remark, we specified how the p-values were computed (computing an ANOVA with the complete model compared to the model without the effect in question). (lines 329 -330).

P-values and degrees of freedom were obtained with the Satterthwaite approximation method by the comparison of the full model against the model without the effect.

C8.g. Perhaps also explain Table 1, 3rd column. Degrees of freedom are typically integer. I know that in some cases they may be floating point, but it may be a good idea to say something about why this happened or what they mean

A.8.g. We thank the reviewer for the comment. The degrees of freedom are calculated with Satterthwaite approximation (which relies on parametric bootstrapping, see (Luke, 2017)). Please see the answer above for the corresponding line (A.8.f.).

C8.h. Use standard terminology to describe the LMM. E.g., say random factor (with or without random slopes) instead of other terminology, e.g. in lines 375-376, which I admit I did not understand.

A.8.h. We thank the reviewer for this comment. We have adapted the terminology accordingly (l. 325-326).

Individual differences were modelled using random intercepts per subject. Item-related variability was modelled using both random intercepts and slopes as a function of languages 

C8.i. When relevant (e.g. in follow-up analyses), specify whether the p values are one-tailed or two-tailed.

A.8.i. We thank the reviewer for this comment. All p-values are two-tailed. We added the information in the revision as following (l. 349-350):

Then EMM were compared with pairwise tests on the relevant interactions, two-tailed p-value are Bonferroni adjusted.

C9 Line 343-344: interesting and important point. But please explain why this is important.

A.9. We thank the reviewer for this comment, according to the suggestion, we have added the explanation, as follows (lines 369-372):

The interaction subsisted also in further analyses limiting the dataset to number words with 4 syllables (see S2 Table 3) to control for the possible world length effect confounder (see (see Ellis & Hennelly, 1980)). Critically none of the analyses resulted in three-way interactions between Age, Language and Number Size.

C10 Line 302: I assume that when analyzing reaction times, you removed the incorrect-response trials? This should be the case, as incorrect trials cannot be expected to exhibit the same RTs. Same issue in the matching task

A.10. We thank the reviewer for this comment, indeed the RT analyses are done on the correct responses exclusively. This is described in the data filtering part, at the beginning of the results (lines 294-296). 

Finally, for reaction time analyses all incorrect responses were removed, additionally removing 2.93 % of the reading task and 3 % of the matching task, corresponding to the total error rate.

C11 Lines 397-403 – the statistics is not written clearly, and I suspect it’s also not optimal. You write that the hypothesis on language of acquisition was not supported, but I was not convinced this is correct. I suspect that if you analyze all age groups together, of course limiting to numbers < 60, you will get a significant effect.

A 11. We thank the reviewer for this comment which overlaps with reviewer 2 (C.12). Indeed despite the fact that the difference between German and French seems visible on the verbal matching task’s RT figures, it was not significant in the follow-up analyses (we even tried with less conservative p-value adjustments, but we obtain the same pattern of significance, see R2 A.12). We tried some additional analyses excluding the 5th graders from the dataset, since they have the biggest variance and difference. For the follow-up contrasts regarding the language of acquisition, we obtained almost the same pattern: near significant difference in 8th graders (t(78.3) = 2.65, p = .06), significant for 11th (t(96.8) = 3.1, p = .01) but not for adults (t(122.2) = 1.88, p = .37). Nevertheless, the z-score transformed data (which we originally decided to compare also because of this doubt) does not lead to an interaction with age. 

C12 Certainly, I wouldn’t present it as if the effect existed only in 5th grade and nothing at all for older ages (line 439). You have very nice data, no reason to undermine it. This is also a more-general issue about follow-up analyses: you don’t have to run t-test (or whatever similar test it was) on small portions of the data. Why not just run LMM on the relevant subset of data?

A.12. We thank the reviewer for this comment. Indeed we could have used a simple effect analysis as suggested. Here we followed the procedure suggested by (Singmann & Kellen, 2019), doing the follow-up with pairwise comparisons on the estimated marginal means. We have modified the summary to further clarify our findings (l. 473-477):

In summary, for the verbal-visual matching task the (1) transparency of power hypothesis could be confirmed, as transcoding performances were overall lower for French ’70s, ’80s and ’90s numbers having a vigesimal structure, stably across age groups. Concerning the hypothesis on (2) language of math acquisition, the advantage for transcoding in German (LM1) was also stable across age groups, when considering standardized RTs (see Fig. 6).

C13 The detailed analyses in the annex are interesting. I think it would be nice if mentioned them very briefly in the main text – perhaps just say the main conclusions and reference the annex. When I read the previous manuscript version I was worried about these issues, other readers may feel the same and you have good answers. Also, some of the things you wrote in the response to review, which were correct and important in my opinion, may be added to the annex (currently it’s stats and data, without much conclusions).

E.g., lines 507-509 may be a relevant place to mention this.

A.13. We thank the reviewer for this suggestionsand we added the supplementary analysis in the supplementary materials as suggested and refer to it in the text (at lines 546-549 and lines 661-663). 

Also, in answer to an interesting comment in the review process, we conducted additional analyses on reaction times (see S4), by adding all decades as levels for the Number Size factor (hence 8 levels from ‘30s to ‘90s). Those additional analyses confirm that for both tasks in French, reaction times become slower from the ‘70s decade onwards.

C14 Line 492 claims that number matching is transcoding from verbal to visual. This is incorrect. You can’t know which transcoding direction the participants use in a number matching task.

A 14. We thank the reviewer for this comment, we have corrected the sentence to adapt to the suggestion (line 530-531):

Notably, the cost was found for both reading aloud and number matching tasks that relies on transcoding from visual to verbal processes and vice-versa.

C15 Please write in supplementary material files which task they refer to.

A 15. We thank the reviewer for this suggestion, we have added the headers in the supplementary materials.

C 16Line 493-494: the finding that the transparency of power effect on z-scored RTs remains stable across age is interesting. However, you did not say which data supports it. In my opinion, the critical analysis that supports it is the absence of triple interaction in the z-scored data. So if you want to reach this conclusion, you should mention the triple interaction issue in the results section (maybe even move this z-scored analyses into the main text).

A.16. We thank the reviewer for this suggestion, we have added the following sentences in the corresponding results section (lines 444-446):

The linear model applied on z-scores (see Fig 6) replicated the results with raw data described above, except, the three-way interaction with age, suggesting that the age differences observed with raw RTs, might be caused by differences between age-group variability (see S2 Table 3)

C17 Line 497-515: I think you propose an interesting idea – that the difficulty in 70-99 may arise either from syntactic complexity or from morphological-retrieval complexity of these numbers. However, I was not convinced by your claim that the former interpretation necessarily relates to ADAPT and the latter necessarily relates to Dotan & Friedmann. I think that both models can accommodate both interpretations.

A.17. We thank the reviewer for this comment. We might have not expressed ourself clearly enough, we did not thought to contrast the ADAPT to the Dotan & Friedmann model, but to discuss how our results can fit with them – in particular with ADAPT. We have reformulated the paragraph as follows (see lines 551-555):

Finally, the cost for `70s, `80s and `90s numbers also fits with Dotan and Freidman’s recent transcoding model which suggests that reading `70s and `90s French number words requires additional irregular rules. When reading 75 for example, a number word frame would be structured by a decade and a teen instead of ten, then the ten frame filling would be changed from “7” to “6” (the filled frame would be: [6:tens] [15:teens] to give “soixante-quinze”, literally “sixty-fifteen”) (Dotan & Friedmann, 2018). 

C18 The idea of proactive interference (line 515) seems far-fetched. Retrieving 4 morphemes instead of 2 may be difficult for several different reasons, I am not sure I see a reason to point specifically at PI. In any case, Campbell’s and De Visscher’s studies on multiplication talk about a scenario quite different – not interference of single words because they are numerous, but of multi-word multiplication exercise because they are similar. The analogy seems to me more misleading than illuminating.

A.18. We thank the reviewer for this comment, we aknowledge the position that the reviewer is not convinced by the proactive interference interpretation. Yet, we would like to mention this possibility in the discussion, as this reflection emerged after reading more general language-related literature, where words from the same lexical domain highly interfere with each other. If we consider number words as a well-defined and very frequent lexical domain, it could be that “morphologically composed numbers”, such as the vigesimal numbers, might cause interference. To clarify that this interpretation originates from the language literature we changed the following (lines 559-560)

 Eventually, these morphemes, which are made of other numbers might interfere among each other and slow down RTs, in a similar fashion as for multiplication (i.e. proactive interference (Campbell, 1995; De Visscher & Noël, 2014)).

C19 Line 543: presenting number reading as a form of free recall, or the matching task as a form of familiarity judgment, seems far-fetched and perhaps even misleading. I totally agree with you that the tasks have some resemblance, in the sense that the former two require unlimited retrieval, whereas the latter two require verification (lines 544-548). But you can say just that, no need to imply that the tasks are more identical than they really are.

A.19. We thank the reviewer for this comment and respectfully disagree with the interpretation. Again, following the perspective of lexical retrieval, we argue that the underlying long-term memory cognitive mechanisms differ. For example considering the ADAPT model, which considers LTM retrieval for each morphemes, then both tasks might have recurred to a different form of retrieval (free recall vs recognition).

C 20 Lines 576+: another limitation may be that it’s hard to disentangle language of math acquisition from an alternative interpretation that assumes intrinsic differences between German and French (“French is just harder”).

A.20. We thank the reviewer for this comment, indeed both languages are intrinsically different. Concerning the difficulty – specifically for number words – there are mainly studies pointing at German number words to be harder to learn, compared, for example, to English number words. This is often explained by the inversion effect (see the introduction part in German and work from the Nuerk team in Tübingen for example). Therefore the interpretation that French is harder seems difficult to make since both languages have intrinsic charachteristics that would make it harder to learn: less transparency of order (German) and less transparency of power (French).

If what is meant by the reviewer is that it is harder to learn French in general as a language than German, then we also think both languages have their difficulties – for example compared to English. On one side French might be harder due to irregular verbs and less phono-grammatic transparency. But on the other hand German has multiple cases (Dativ, Genitiv and Akusativ) and a neutral “das” case which is inexistent in French. Also words are in average longer in German than in French. Finally, we are not aware of studies indicating if French is harder than German to learn.

 

Reviewer #2: 

I read a revised version of the manuscript and first I applaud the authors for the endeavor to greatly rewrite the work including a complete change of the analyses. That is a large step that undoubtedly took quite some time but it is definitely a large improvement.

I noted that reviewer 1 is much more knowledgeable than me on the theoretical details. It looks like that improved a lot too but reviewer 1 is probably much better able than me to judge that (if also rereviewing the manuscript).

I do still have some remaining questions, mostly about the analyses.

C1 p. 4-5 for the reader unfamiliar with those languages it would be helpful to give an example in words, not just in formula of 10*x+y (and also define x and y). You do this for French now but not for German (I’d also give the example both in the original language and its counterpart in English).

A.1. We thank the reviewer for this comment, we agree it would be helpful for the reader. We have implemented the following changes (and an italic emphasis to find the corresponding unit/decade) (lines 93 -113). For the ulterior changes refer to (R1 C3).

Transparency of power refers to the existence of different degrees of morpho-syntactic language transparencies in verbal numbers. In many Asian languages (i.e. Mandarin Chinese, Vietnamese) the morpho-syntactic structure of the verbal number system is highly consistent with the Arabic number system, and in general with the base-10 (i.e. 10*x + y, where x indexes the base index and y the unit, see (Miura et al., 1988)). This linguistic characteristic, also termed transparency of power, facilitates learning to count (Lê & Noël, 2020; Miller et al., 1995; Miller & Stigler, 1987) and solving arithmetic problems (Geary et al., 1996; McClung & Arya, 2018; Rodic et al., 2015). Another morpho-syntactic difference concerns the inversion between tens and ones as for example in Dutch and in German (i.e. y + 10*x; in German 42 is “zwei-und-vierzig”, literally “two-and-forty”). This linguistic characteristic, called transparency of order (Bahnmueller et al., 2018), explains some transcoding error patterns and reaction times (Clayton et al., 2020; Imbo et al., 2014; Moeller et al., 2015; Pixner et al., 2011; Poncin et al., 2019; Steiner et al., 2021; Zuber et al., 2009), gives rise to specific pattern of compatibility effects (Bahnmueller et al., 2015; H. Nuerk et al., 2005; H.-C. Nuerk et al., 2004) and complicates the solution of certain arithmetic problems (Göbel et al., 2014; Lonnemann & Yan, 2015; Xenidou-Dervou et al., 2015).

Transparency of power can also vary with a change of base, for example a base-20 system (20*x + y), also called vigesimal. The use of the vigesimal system is found for example in French, Basque, or in Diola-Fogny (see (Haspelmath et al., 2005)). French, to be precise, has a mixed system since it uses base-10 and base-20 systems. Up to the `60s in French, like in English, the counting system follows the base-10 rule. However, `80s to `90s follow a base-20 system, e.g. 87 = 4*20 + 7, “quatre-vingt-sept”, literally “four-twenty-seven”.

Moreover, the teens (11 to 16) are lexical primitives, like in English and in stark contrast to the more transparent Asian languages. Note that the transparency contrast is additionally increased for 71 to 76 and 91 to 96, which are composed with a base-20 and teens, e.g. 96 = 4*20 + 16, “quatre-vingt-seize”, literally “four-twenty-sixteen”. Furthermore, the vigesimal system is subject to regional variances, for example in Belgian Wallonia `70s and ‘90s are not vigesimal (i.e. “septante” for “seventy” and “nonante” for “ninety”, see (Seron & Fayol, 1994)) and in some Swiss-French cantons the vigesimal system is entirely absent (i.e. even for 80, “huitante”, literally “eighty”).

C2 I also doubt if ‘transparency of power’ is the best term. It sounds nice/interesting but it’s only the names of the tens that are affected, not the hundreds or thousands (admittedly, it returns for tens of thousands, but still). So isn’t it just ‘transparency of tens/decades’?

A.2. We thank the reviewer for this comment, we decided to use this terminology in the sense that it has been proposed in (Bahnmueller et al., 2018; Dowker & Nuerk, 2017), which is to “derive the power of each number directly from the number word” which we believe would include the concept of “transparency of tens”. In that sense vigesimal number words in French would fit with this definitions since, “quatre-vingt” (80), literally “four-twenty”, does not provide a direct way to derive the corresponding eight decimal power. 

C3 Line 173 ‘These results support the above-mentioned TCM stating that precise numbers are encoded in a language dependent format [63].’ I was under the impression that the results mentioned above all related to studies that offered math in a verbal format to find differences in language of presentation. For instance, the text speaks of ‘problems presented in the LM+’. But if this is the case, the conclusion above cannot be drawn. The use of a verbal context means we can’t know if the precise numbers (by the way, does ‘precise numbers’ refer to exact quantities or symbolic numbers?) are stored in a language-dependent or -independent format.

A.3. We understand the reviewers comment concerning the verbal format of the cited studies which would elicit a verbal context. However one of the studies mentioned compares LM Basque and LM Spanish using arabic digits (Salillas & Carreiras, 2014). Nevertheless, we have toned down the formulation by changing the following sentence (line 178-179)

These results tend to support the above-mentioned TCM stating that precise numbers are encoded in a language-dependent format (Dehaene et al., 1999).

C4 Line 177 ‘However, they agree with recent reports from the bilingualism literature that academic knowledge acquired in one language is not (or difficultly) transferable to the other language [55].’ That is a bold statement! That basically says I can’t use the knowledge I learn in English in my native language? Luckily this is not true. This study says that there is a switch cost between languages in math problem solving. A switch cost in mathematics is quite far away from this conclusion that in general academic knowledge learned in one language is not transferable to the other language!

A 4. We thank the reviewer for this comment, we understand that the statement is misleading compared to what we wanted to express. We have adjusted the sentence as follows (line 181-183).

However, they agree with recent reports from the bilingualism literature that academic knowledge acquired in one language is retrieved more efficiently in the learning language compared to another language (Volmer et al., 2018).

C5 Together it would be nice to see a better paragraph explaining the present research question to earlier knowledge. What is it that we don’t know and how does study contribute to filling that gap? 

A 5. We thank the reviewer for this suggestion. We have added a paragraph explaining the research question in the context of previous research about mathematics in Luxembourgish samples (see C 8.). We hope that this additional context information will help to better understand the relation with our previous studies as described just below (lines 192-198).

C6 Line 212: The two tasks might thus reveal a somewhat different result pattern, as already observed in Van Rinsveld et al. [69]. Please elaborate on how.

A 6. We thank the reviewer for this comment. We have adjusted the sentence according to the suggestion as follows (lines 221-223):

The two tasks might thus reveal a somewhat different result pattern such as more marked linguistic influences in the reading aloud free recall task than on the verbal-visual recognition task, as already observed in Van Rinsveld et al. (Van Rinsveld et al., 2016).

C7 Present study: I said this before, when describing the population also describe the mother tongue of the children and a bit more of the language background in Luxembourgh besides language in the classroom.

Specifically it is relevant to know and possibly not widely known by the audience that Luxembourgish strongly resembles German. So one may argue that for the 70 German- and Luxembourgish speaking children, LM1 and L1 are (virtually) identical in terms of number words, whereas for the other 27 this isn’t the case. That means there are two linguistically very diverse groups of participants. It would at least be relevant to know at what age the immigrant children had started learning German and French.

This information is also relevant because in the way the study is framed now, the youngest age group wouldn’t be able to do the task in French at all because they wouldn’t know it yet. But they do. But how well? And to what degree can all results be explained by just very poor proficiency in one group that hasn’t properly been taught French at all?

A 7. We thank the reviewer for this comment, we have adapted the first paragraph of the present study in order to add more information about the linguistic charachteristics of the sample that followed the Luxembourgish school curricula. Regarding the Luxembourgish speakers, this might probably have an influence in terms of L1 to LM1 transfer. As we have answered in the previous review, the sub-sample was unfortunately to small and not equally represented across the age groups to be compared separatly (lines 192– 198).

In the Luxembourgish multilingual school system, pre-schools (3 to 5 y.o.) are in Luxembourgish, which is linguistically close to German. The teaching language in primary school (1st to 6th grade, 6 to 12 y.o.) is German, except for French lessons. From 7th grade to 10th grade, the majority of subjects are taught in German, except for mathematics and French lessons, which are taught in French. Then from 11th grade until the end of obligatory school, all topics are thought in French. This multilingual education system aims to render students highly proficient in both German and French. In sum, critically for number transcoding, the language for teaching and learning mathematics switches from German to French at 7th grade (Languages in Luxembourg Schools, 2021). Hence students’ first language of math acquisition (LM1) is German, while their second language of math learning (LM2) is French. Therefore allowing within-subject designs from a sample with the same educational background. 

C8 Line 265. There are 2 sets of 5 practice and 14 test stimuli but the experiment lasted 45 minutes. That sounds extremely long for this task. If this test was part of a larger test battery please mention the context.

A 8. We thank the reviewer for this comment. Indeed the experiment was part of a bigger study where the participants passed additional tests (lines 250-253):

Stimuli for each task were 28 different two-digit numbers ranging from 31 to 98. These were further subdivided into two sets of 14 distinct stimuli each, one for the German and the other for the French blocks. The experiment was part of a larger study, which lasted 45 minutes, at the end of which the participants were compensated with a gift voucher.

C9 Line 242 still makes the cut at 60 instead of 70. (there are no 14 numbers above and 14 below 60 in the set by the way. There are 4 from each decade). Also explain this decision to leave out the 60 numbers earlier than the results section (it’s not a result, it’s a methodological choice).

A 9. We thank the reviewer for this comment. We thank for noticing this error in the stimuli description which we have corrected and adjusted adding further clarification information, see lines 270-272 above.

C10 Line 285. Have you considered post-error slowing (the ‘oops’ effect)? It may be wise to remove RTs after an error too.

A10. We thank the reviewer for this comment. Note that incorrect responses are excluded from the reaction time analyses of both tasks, hence excluding possible “post-error” slowed down trials (lines 394-397).

Finally, for reaction time analyses all incorrect responses were removed, additionally removing 2.93 % of the reading task and 3 % of the matching task, corresponding to the total error rate. 

C11 Could you give some descriptive statistics? Like M and SD of RT/accuracy per task, broken down by the relevant factors, so by age group, language and number size.

A.11. We thank the reviewer for this comment. We have added the measured mean differences in reaction times where suggested. Note that the break down tables for each level’s by means and standard errors for reaction time and accuracy are in the supplementary materials. (Below, we have removred the statistical results within the text and kept in the table only, according with reviewer 1 suggestion). For example (line 334 – 337, but also lines 352, 353, 385, 388,434,436,459,467):

The model for the reading aloud task RTs resulted in three significant main effects and three two-way interactions (see Table 1). The main effect of Age was decomposed with follow-up analyses showing that the 5th graders (1091(290) ms) were slower than the older groups, which had similar RT’s (8th 797(288) ms, 11th 780(247) ms and adults 828(320) ms; standard deviations in parenthesis).

As well in lines 422-429. Note that we changed from Walden z to t-tests given that we used Satterthwaite approximations and that readers might be more familiar with (see R2 C8.c and A8c.):

Follow-up analyses were applied comparing the estimated marginal means with paired comparisons on Satterthwaite corrected degrees of freedom and Bonferroni adjusted p-values. Follow-up analyses confirm the hypothesis relating to transparence of power, given a cost in French for ’70s, ’80s and ’90s compared to ’30s, ’40s and ’50s numbers for all age groups: 5th graders with a 593 ms cost ( t(98.0)=9.48,p<.001), 8th graders 364 ms cost (t(61.0)=5.34,p<.001), 11th graders a 356 ms cost (t(70.7)=3.28,p<.01), and Adults with a 219 ms cost ( t(87.2)=3.16,p=.01). In contrast, the same comparison in German did not result in any significant differences (all t<1.53,p>.76,max difference= 34 ms ). In sum, the cost for vigesimal numbers observed in the reading aloud task is replicated with the verbal-visual matching task for all age groups, see Fig 5.

C 12 306 we aimed to model the individual variability, ideally considering individual differences (i.e. Subject) and age group (i.e. Age) related differences. Furthermore, we aimed to model per-item variability (i.e. Item), particularly related to language.

I’m a bit confused here on what exactly you did especially with age group. Is this about random intercepts or random slopes? Slopes for what? And you modeled age group as a fixed effect already so you can’t add a random effect for it too, so it makes sense that that didn’t work.

So what did the model syntax look like exactly?

It sounds like it was something like RT~Age*Language*Number Size + (1|Subject) + (1+language|Item)?

Or also (1+language|Subject)? If not, why no random slope for language per subject given that you say you run a maximal model?

A.12 We thank the reviewer for this comment which joins also some of the first reviewers concerns (see R1 C7). We will explain herein into details how we designed and implemented the model. The initial maximal model that we had defined was the following:

 RT ~ 1 + Age*Language*Size + (Age|Subject) + (1 + Language|Item)

The reasoning was to have (Age|Subject) as random part reasoning that subjects variability would mostly depend on age group (i.e. different speed of processing, as well as different variabilities). The (1 + Language|Item) random part was designed to catch the number words different lengths and/or phonological characteristics further depending on Language characteristis (given that word length has been brought up as a confounder in bilingual designs (Ellis & Hennelly, 1980)). However the maximal model did not converge (despite trying different optimizers, increase the iterations to 1,000,000: we tried to “fix the convergence” following some common advices as the reviewer lists in C15). Since the model kept not converging, we tried removing different random parts of the maximal model. With this procedure, we reached to the final model Mend (the one presented in the results):

 MM <- RT ~ 1 + Age*Language*Size + (1 + Age| Subject) + (1 + Language|Item)

 M1 <- RT ~ 1 + Age*Language*Size + (1 + Age|| Subject) + (1 + Language| Item) 

 M2 <- RT ~ 1 + Age*Language*Size + (1 + Age| Subject) + (1 + Language|| Item)

 M3 <- RT ~ 1 + Age*Language*Size + (1 + Age|| Subject) + (1 + Language|| Item)

 M4 <- RT ~ 1 + Age*Language*Size + (1|Subject) + (1 + Language|| Item)

 Mend <- RT ~ 1 + Age*Language*Size + (1| Subject) + (1 + Language| Item) 

(Besides M4 and Mend, none of the above model did converge). The Mend model was judged satisfactory among the converging final model because it has a per-item random part, and a per-subject random part (though “only” for intercepts). The model’s R2 fit of .57 for the naming and .62 for the matching model) was also judged satisfactory (calculated with the package “MuMIn”although we do not report it in the manuscript given the following issues with effect sizes in linear mixed models: https://afex.singmann.science/forums/topic/compute-effect-sizes-for-mixed-objects). Therefore we did not search for further model possibilities. However, concerning the manuscript, as suggested by reviewer 1 (R2 C7 & A7), we removed the different models that were tried as well as the maximal model description in the results so to render the reading of the reults possibly more fluid and kept only a short sentence about how we obtained the final model (lines 301-302)

When our initially designed full model failed to converge, we reduced the model complexity by gradually simplifying the random effect structure until convergence was reached.

We have changed the paragraph about the model description as follows (lines 321-332):

The linear mixed model was defined with main effects and interactions between the fixed between-group factor Age (levels: 5th, 8th, 11th grade and adults) and two fixed within-group factors: Language (German, French), and Number Size (Large, Small), all levels being defined as categorical. As random factors, we modelled individual differences (i.e. Subject) and item-related variability (i.e. Item), the latter being considered separately by language. Individual differences were modelled using random intercepts per subject. Item-related variability was modelled using both random intercepts and slopes as a function of languages. In sum, the model takes the following R syntax form (A):

 (A) RT ~ 1 + Age*Language*Number Size + (1|Subject) + (1 + Language|Item).

P-values and degrees of freedom were obtained with the Satterthwaite approximation method by comparing the full model against the model without the effect (Singmann et al., 2020). Follow-ups were calculated by comparing estimated marginal means (EMM) obtained with the emmeans package (Lenth, 2021), and p-values were adjusted with the Bonferroni method.

And adapted lines 403-405 in the verbal-visual matching task:

For coherence of interpretation and comparison with the previous verbal-visual matching task, the same model and parameters for p-values, degrees of freedom and follow-up’s were applied here on RT.

C 13. Nd could you tell which method you used to obtain p-values? Conditional F-tests with Kenward-Roger or Satterthwaite corrected degrees of freedom probably?

A 13. We thank the reviewer for this comment. Satterthwaite degree of freedom approximation was used here. (Note that the comparison with Kenward-Rogers method did not lead to big differences in terms of df for our dataset, typically third or fourth digit position, but uses significantly longer computation time than Satterthwaite approximation), see cited paragraph above (lines 329-332, see also the note of Table 1).

C14. Lines 314, 325 and also in later sections. Specify how you did your follow-up analyses when applicable (and what were the resulting statistics)? Explain what kind of analysis (the same?) on what kind of subset of the data. I see no more F tests but z-scores, if these are Wald tests from mixed effects models, those aren’t very reliable.

A.14. We thank the reviewer for noticing this. The follow-up’s were computed comparing estimated marginal means from the model (from the package emmeans), we have added this information below (lines 348-350): 

Follow-up analyses were performed on the model's estimated marginal means (EMMs) of the interaction's terms [see 77] with Satterthwaite estimation of degrees of freedom. Then EMM were compared with two-tailed pairwise tests with Bonferroni adjusted p-values

C15 Line 347 and beyond, what do you mean that the model did not fit? Do you mean that it did not converge? Why specifically drop age here, it seems the most relevant factor? I also learned that dropping a random slope is the very very last thing one should try as it greatly reduces p-values (so it inflates significant findings). https://academic.oup.com/esr/article/35/2/258/5306121. Now non-convergence can be a beast but other solutions to solve it do exist. This is what I learned to try, in this order:

– Most recent package versions (particularly lme4)?

– Center instead of standardize (or vice versa) cont. predictors

– Increase number of iterations

– Restart model fitting from previous (unconverged) estimates

– Different optimizer(s); compare estimates from many different optimizers; (compare to) brms model (Bayesian model)

- Simplify your model, Barr et al.: remove: (a) random correlations (b) random intercepts (c) random slopes; last resort, try to avoid that!

By the way, alternatively I could also imagine leaving accuracy out altogether because of its ceiling effects.

A 15. We thank the reviewer for this very detailed and informed recommendation. We tried the most of these recommendations as described below, but honestly, we are not even sure how to interpret the mixed model’s results (if they converged) given the ceiling effect. As reported in S1 Table 2 and 4, there are some cells with 100% accuracies and other with only errors in only one or two trials. The task is not hard enough to induce participants in enough errors, making this measure insensible. Nevertheless, we tried to fit the model again with (1) the updated to the latest lmer package (i.e. version 1.1-27), (2) we increased the number of iterations to 1e6 (3) tried different optimizers (with all_fit = TRUE of the mixed package). Finally, the model is already as simplified as possible, so that removing an additional term would then also remove the meaning of doing the model . We could not center nor standardize the predictors, since the dependent variable accuracy is binomial. 

C 16 Line 358 in order to explain the interaction language * number size you give the results for French but also give the contrasting results for German (I guess no sig difference? But I shouldn’t need to guess.)? You need them both to understand the interaction.

A.16 We thank the reviewer for this comment. We have added the results for the contrast in German as suggested (lines 385-388):

Follow-up analyses confirmed with CR the pattern observed with RT: in French, 4.8% more errors were made with ’70s, ’80s and ’90s than ’30s, ’40s and ’50s numbers (z=-3.37,p<.01), indicating a cost for vigesimal number transcoding. The same contrast in German did not lead to any significant difference (z=0.63,n.s.).

 p. 17 I’m a bit surprised that the language difference is so weak given the highly significant effect in table 3 and also the large differences in the figures. Is this correct?

The analyses with only 4 syllable words made me wonder how to count syllables in French reliably given that so many of them are silent. If I look it up there seems to be consensus that silent e’s at the end of a word don’t count as syllables meaning that many words are one syllable shorter.

A 17. We thank the reviewer for this comment. We were also surprised to observe that the language effect is not very stable across ages given the figures, we have added the estimated differences. We re-runned the follow-up analyses with three different p-value adjustments (holm, Bonferroni, and free as advised here: https://cran.r-project.org/web/packages/afex/vignettes/afex_mixed_example.html#follow-up-analyses) to test the significance stability. The significancy patterns are confirmed by all three types of adjustments, only the 11th grade group fluctuates around the alpha level, depending on the adjustments done, with the Bonferonni (the one in the manuscript) being the less significant and most conservative p-value. 

Age group: Contrast: Holm Bonferroni Free

5th Large(FR)-small(FR) <.0001 <.0001 <.0001

 Small(FR)-small(GE) <.0001 <.0001 <.0001

 Large(GE)-small(GE) 0.127 0.7617 0.4185

8th Large(FR)-small(FR) <.0001 <.0001 <.0001

 Small(FR)-small(GE) 0.091 0.182 0.1319

 Large(GE)-small(GE) 0.8192 1 0.9957

11th Large(FR)-small(FR) 0.0064 0.0097 0.0086

 Small(FR)-small(GE) 0.0369 0.0737 0.0589

 Large(GE)-small(GE) 0.497 1 0.9039

Adults Large(FR)-small(FR) 0.0086 0.0129 0.0113

 Small(FR)-small(GE) 0.4086 0.8172 0.4423

 Large(GE)-small(GE) 0.7571 1 0.9758

Note: FR: French, GE: German. Large: ’70s ‘80s and ‘90s, small:’30s, ‘40s and ‘50s. In bold the contrasts concerning the language difference.

Note that we had obtained a similar follow-up pattern when running the analyses with ANOVA (cfr first manuscript). Also note that this observation was the motivation behind running the same analyses on standardized z-scores (as it would control for the bigger variability observed in 5th grade, see standard error in the figure and SD in the table). Indeed the z-score turn out to be very similar across age groups (very “flat lines” in the figure), suggesting that the non-significant effects with RT might be inflated by the variability in RT responses by the younger age group. 

Second, concerning the silent “e” at the end of the number words, we replicated the analyses by counting them as silent or not (i.e. 4 syllables for soi/xan/te/quinze). By counting them as silent we replicated for the naming task, the Language x Number Size interaction. For the matching task we replicate the three-way interaction. Note that, theoretically, item variability is also included now into the linear mixed model’s per-items random slopes and intercepts.

C 18 Concerning this language difference in the verbal to visual task: isn’t German also easier in the sense of at which point you can already eliminate multiple choice options on the go? If you hear ‘vierund…’ you can already exclude the +1/-1 and the +11/-11 answers, leaving only two possible options halfway through the word. For French, specifically for the large numbers, hearing ‘quatrevingt…’ may still leave 3 or 4 options open halfway through the word.

A 18. We thank the reviewer for this interesting reflection about possible participant’s strategic behaviour. We would like to bring to attention that the auditory and visual stimuli were presented sequentially, not simultaneously hence excluding the possibility for this strategic behaviour to be online. In order for this strategic “pre-selection” to occur, it would need a prior mental representation to have been built up by the participant (likely in verbal working memory, since transcoding is sensible to verbal interference (Frank et al., 2012)). Therefore the verbal to visual transcoding (which is critical for the hypothesis) must have occurred before the hypothetical strategic “pre-selection”. Also, a “pre-selection” would be possible in French by excluding the possibilities with a different unit.

C19 Minor points

A.19. We thank the reviewer for noticing the following minor points, which we have fixed accordingly in this revision. 

 Line 151. comparable results than for problems presented  comparable results to problems presented. Also in line 194.

C.19.a. see l.156, 203

The results indicate a cost for arithmetical problems written in number words in Filipino (i.e. being the L1 but LM-) compared to English (i.e. being an L2 but LM+), which in turn showed comparable results to problems presented as Arabic digits.

Previous studies on this specific population have reported language effects in magnitude comparison tasks, showing comparable compatibility effects to monolingual German [69].

 Line 161/ who had either learned arithmetic in English or Spanish  who had learned arithmetic in either English or Spanish.

C.19.b. lines 166-167

Recording electroencephalogram during a true or false judgment of simple multiplications, Salillas and Wicha (Salillas & Wicha, 2012) studied fluent Spanish-English bilingual adults who had learned arithmetic in either English or Spanish, the LM+ respectively

 Line 227 switches to French (LM2) after the 7th grade  after 6th grade or in 7th grade.

C.19.c. line 237

Importantly the language support for the formal acquisition of mathematics through school curricula is first in German (LM1) from 1st to 6th grade and switches to French (LM2) in 7th grade.

 Line 358 says ‘40s and ‘40s instead of ‘50s.

C.19.d. line 386

 ’30s, ’40s and ’50s numbers

 Quite some typos in the supplementary table with written out number words like ‘soixantetroi’, ‘siebenndneunzig’, etc.

C.19.e. Thank you for noticing those typos! We have corrected them in the present supplementary material (S3 Table 1). i.e. soixantetrois or siebenundneunzig.

 

Bibliography:

Bahnmueller, J., Moeller, K., Mann, A., & Nuerk, H.-C. (2015). On the limits of language influences on numerical cognition – no inversion effects in three-digit number magnitude processing in adults. Frontiers in Psychology, 6. https://doi.org/10.3389/fpsyg.2015.01216

Bahnmueller, J., Nuerk, H.-C., & Moeller, K. (2018). A Taxonomy Proposal for Types of Interactions of Language and Place-Value Processing in Multi-Digit Numbers. Frontiers in Psychology, 9. https://doi.org/10.3389/fpsyg.2018.01024

Barrouillet, P., Camos, V., Perruchet, P., & Seron, X. (2004). ADAPT: A Developmental, Asemantic, and Procedural Model for Transcoding From Verbal to Arabic Numerals. Psychological Review, 111(2), 368–394. https://doi.org/10.1037/0033-295X.111.2.368

Campbell, J. (1995). Mechanisms of Simple Addition and Multiplication: A Modified Network-interference Theory and Simulation. Mathematical Cognition, 1, 121–164.

Clayton, F. J., Copper, C., Steiner, A. F., Banfi, C., Finke, S., Landerl, K., & Göbel, S. M. (2020). Two-digit number writing and arithmetic in Year 1 children: Does number word inversion matter? Cognitive Development, 56, 100967. https://doi.org/10.1016/j.cogdev.2020.100967

De Visscher, A., & Noël, M.-P. (2014). The detrimental effect of interference in multiplication facts storing: Typical development and individual differences. Journal of Experimental Psychology: General, 143(6), 2380–2400. https://doi.org/10.1037/xge0000029

Dehaene, S. (1992). Varieties of numerical abilities. Cognition, 44(1), 1–42. https://doi.org/10.1016/0010-0277(92)90049-N

Dehaene, S., Piazza, M., Pinel, P., & Cohen, L. (2003). Three parietal circuits for number processing. Cognitive Neuropsychology, 20(3–6), 487–506. https://doi.org/10.1080/02643290244000239

Dehaene, S., Spelke, E., Pinel, P., Stanescu, R., & Tsivkin, S. (1999). Sources of Mathematical Thinking: Behavioral and Brain-Imaging Evidence. Science, 284(5416), 970–974. https://doi.org/10.1126/science.284.5416.970

Deloche, G., & Seron, X. (1982). From three to 3: A differential analysis of skills in transcoding quantities between patients with Broca’s and Wernicke’s aphasia. Brain, 105(4), 719–733. https://doi.org/10.1093/brain/105.4.719

Dotan, D., & Friedmann, N. (2018). A cognitive model for multidigit number reading: Inferences from individuals with selective impairments. Cortex, 101, 249–281. https://doi.org/10.1016/j.cortex.2017.10.025

Dowker, A., & Nuerk, H.-C. (Eds.). (2017). Linguistic Influences on Mathematical Cognition. Frontiers Media SA. https://doi.org/10.3389/978-2-88945-200-2

Ellis, N. C., & Hennelly, R. A. (1980). A bilingual word-length effect: Implications for intelligence testing and the relative ease of mental calculation in Welsh and English. British Journal of Psychology, 71(1), 43–51. https://doi.org/10.1111/j.2044-8295.1980.tb02728.x

Frank, M. C., Fedorenko, E., Lai, P., Saxe, R., & Gibson, E. (2012). Verbal interference suppresses exact numerical representation. Cognitive Psychology, 64(1), 74–92. https://doi.org/10.1016/j.cogpsych.2011.10.004

Geary, D. C., Bow‐Thomas, C. C., Liu, F., & Siegler, R. S. (1996). Development of Arithmetical Competencies in Chinese and American Children: Influence of Age, Language, and Schooling. Child Development, 67(5), 2022–2044. https://doi.org/10.1111/j.1467-8624.1996.tb01841.x

Göbel, S. M., Moeller, K., Pixner, S., Kaufmann, L., & Nuerk, H.-C. (2014). Language affects symbolic arithmetic in children: The case of number word inversion. Journal of Experimental Child Psychology, 119, 17–25. https://doi.org/10.1016/j.jecp.2013.10.001

Haspelmath, M., Dryer, M. S., Gil, D., & Comrie, B. (2005). The World Atlas of Language Structures. Oxford Univ. Press.

Imbo, I., Vanden Bulcke, C., De Brauwer, J., & Fias, W. (2014). Sixty-four or four-and-sixty? The influence of language and working memory on children’s number transcoding. Frontiers in Psychology, 5. https://doi.org/10.3389/fpsyg.2014.00313

Languages in Luxembourg schools. (2021). http://men.public.lu/en/themes-transversaux/langues-ecole-luxembourgeoise.html

Lê, M.-L. T., & Noël, M.-P. (2020). Transparent number-naming system gives only limited advantage for preschooler’s numerical development: Comparisons of Vietnamese and French-speaking children. PLOS ONE, 15(12), e0243472. https://doi.org/10.1371/journal.pone.0243472

Lenth, R. V. (2021). emmeans: Estimated marginal means, aka least-squares means [Manual]. https://CRAN.R-project.org/package=emmeans

Lonnemann, J., & Yan, S. (2015). Does number word inversion affect arithmetic processes in adults? Trends in Neuroscience and Education, 4(1), 1–5. https://doi.org/10.1016/j.tine.2015.01.002

Luke, S. G. (2017). Evaluating significance in linear mixed-effects models in R. Behavior Research Methods, 49(4), 1494–1502. https://doi.org/10.3758/s13428-016-0809-y

McCloskey, M. (1992). Cognitive mechanisms in numerical processing: Evidence from acquired dyscalculia. Cognition, 44(1–2), 107–157. https://doi.org/10.1016/0010-0277(92)90052-J

McCloskey, M., Caramazza, A., & Basili, A. (1985). Cognitive mechanisms in number processing and calculation: Evidence from dyscalculia. Brain and Cognition, 4(2), 171–196. https://doi.org/10.1016/0278-2626(85)90069-7

McClung, N. A., & Arya, D. J. (2018). Individual Differences in Fourth-Grade Math Achievement in Chinese and English. Frontiers in Education, 3, 29. https://doi.org/10.3389/feduc.2018.00029

Miller, K. F., Smith, C. M., Zhu, J., & Zhang, H. (1995). Preschool Origins of Cross-National Differences in Mathematical Competence: The Role of Number-Naming Systems. Psychological Science, 6(1), 56–60.

Miller, K. F., & Stigler, J. W. (1987). Counting in Chinese: Cultural variation in a basic cognitive skill. Cognitive Development, 2(3), 279–305. https://doi.org/10.1016/S0885-2014(87)90091-8

Miura, I. T., Kim, C. C., Chang, C.-M., & Okamoto, Y. (1988). Effects of Language Characteristics on Children’s Cognitive Representation of Number: Cross-National Comparisons. Child Development, 59(6), 1445. https://doi.org/10.2307/1130659

Moeller, K., Zuber, J., Olsen, N., Nuerk, H.-C., & Willmes, K. (2015). Intransparent German number words complicate transcoding – a translingual comparison with Japanese. Frontiers in Psychology, 6. https://doi.org/10.3389/fpsyg.2015.00740

Nuerk, H., Weger, U., & Willmes, K. (2005). Language effects in magnitude comparison: Small, but not irrelevant. Brain and Language, 92(3), 262–277. https://doi.org/10.1016/j.bandl.2004.06.107

Nuerk, H.-C., Weger, U., & Willmes, K. (2004). On the Perceptual Generality of the Unit-Decade Compatibility Effect. Experimental Psychology, 51(1), 72–79. https://doi.org/10.1027/1618-3169.51.1.72

Pixner, S., Zuber, J., Heřmanová, V., Kaufmann, L., Nuerk, H.-C., & Moeller, K. (2011). One language, two number-word systems and many problems: Numerical cognition in the Czech language. Research in Developmental Disabilities, 32(6), 2683–2689. https://doi.org/10.1016/j.ridd.2011.06.004

Poncin, A., Van Rinsveld, A., & Schiltz, C. (2019). Units-first or tens-first: Does language matter when processing visually presented two-digit numbers? Quarterly Journal of Experimental Psychology, 73(5), 726–738. https://doi.org/10.1177/1747021819892165

Power, R. J. D., & Dal Martello, M. F. (1990). The dictation of Italian numerals. Language and Cognitive Processes, 5(3), 237–254. https://doi.org/10.1080/01690969008402106

Rodic, M., Zhou, X., Tikhomirova, T., Wei, W., Malykh, S., Ismatulina, V., Sabirova, E., Davidova, Y., Tosto, M. G., Lemelin, J.-P., & Kovas, Y. (2015). Cross-cultural investigation into cognitive underpinnings of individual differences in early arithmetic. Developmental Science, 18(1), 165–174. https://doi.org/10.1111/desc.12204

Salillas, E., & Carreiras, M. (2014). Core number representations are shaped by language. Cortex, 52, 1–11. https://doi.org/10.1016/j.cortex.2013.12.009

Salillas, E., & Wicha, N. Y. Y. (2012). Early Learning Shapes the Memory Networks for Arithmetic: Evidence From Brain Potentials in Bilinguals. Psychological Science, 23(7), 745–755. https://doi.org/10.1177/0956797612446347

Seron, X., & Fayol, M. (1994). Number transcoding in children: A functional analysis. British Journal of Developmental Psychology, 12(3), 281–300. https://doi.org/10.1111/j.2044-835X.1994.tb00635.x

Singmann, H. (2021). Mixed Model Reanalysis of RT data. https://cran.r-project.org/web/packages/afex/vignettes/afex_mixed_example.html

Singmann, H., Bolker, B., Westfall, J., Aust, F., & Ben-Shachar, M. S. (2020). afex: Analysis of factorial experiments. https://CRAN.R-project.org/package=afex

Singmann, H., & Kellen, D. (2019). An Introduction to Mixed Models for Experimental Psychology. In D. Spieler & E. Schumacher (Eds.), New Methods in Cognitive Psychology (1st ed., pp. 4–31). Routledge. https://doi.org/10.4324/9780429318405-2

Steiner, A. F., Banfi, C., Finke, S., Kemény, F., Clayton, F. J., Göbel, S. M., & Landerl, K. (2021). Twenty-four or four-and-twenty: Language modulates cross-modal matching for multidigit numbers in children and adults. Journal of Experimental Child Psychology, 202, 104970. https://doi.org/10.1016/j.jecp.2020.104970

Van Rinsveld, A., Schiltz, C., Landerl, K., Brunner, M., & Ugen, S. (2016). Speaking two languages with different number naming systems: What implications for magnitude judgments in bilinguals at different stages of language acquisition? Cognitive Processing, 17(3), 225–241. https://doi.org/10.1007/s10339-016-0762-9

Volmer, E., Grabner, R. H., & Saalbach, H. (2018). Language switching costs in bilingual mathematics learning: Transfer effects and individual differences. Zeitschrift Für Erziehungswissenschaft, 21(1), 71–96. https://doi.org/10.1007/s11618-017-0795-6

Xenidou-Dervou, I., Gilmore, C., van der Schoot, M., & van Lieshout, E. C. D. M. (2015). The developmental onset of symbolic approximation: Beyond nonsymbolic representations, the language of numbers matters. Frontiers in Psychology, 6. https://doi.org/10.3389/fpsyg.2015.00487

Zuber, J., Pixner, S., Moeller, K., & Nuerk, H.-C. (2009). On the language specificity of basic number processing: Transcoding in a language with inversion and its relation to working memory capacity. Journal of Experimental Child Psychology, 102(1), 60–77. https://doi.org/10.1016/j.jecp.2008.04.003

---

## [Decision Letter · Decision Letter 2]

31 Mar 2022

PONE-D-21-18244R2Number transcoding in bilinguals - a transversal developmental studyPLOS ONE

Dear Dr. Lachelin,

Thank you for submitting your revised manuscript to PLOS ONE. I have sent it back to the original reviewers. You can see their feedback below and attached (for reviewer #2's comments). As you can see, reviewer #1 is happy with your revisions and has no further comment. Reviewer #2, however, raises one important statistical point that you need to consider before the manuscript can be considered for publication. I would thus encourage you to address the reviewer's comment (as well as the reviewer's other minor points) in another round of revision. 

We look forward to receiving your revised manuscript.

Kind regards,

Jérôme Prado

Academic Editor

PLOS ONE

Reviewers' comments:

Reviewer #1: All my comments have been addressed.

I just found one typo -- line 378, “suject” should be “subject”.

Reviewer #2: (No Response)

---

## [Author Response · Author response to Decision Letter 2]

18 May 2022

Dear Editor, Prof. Dr Prado,

dear Reviewer,

We thank again the reviewer for their constructive comments on the second revision’s manuscript.

Following those comments, we have made two major changes in the result part: (1) we wrote the results

anew with the model suggested in C1 and (2) we removed trials corresponding to post-error slow-down

(i.e. trials following an incorrect response). Therefore, those changes required writing the descriptive data

anew (that is all tables, figures, and supplementary analyses), which can be found in the present revision.

However, the significance patterns described in the revision 1 and 2 remained largely unchanged. The

only change in significance was a now significant three-way interaction between age, number size, and

language in the number reading task. This three-way interaction, however, is not significant with the two

additional analyses on z-scores and subset data of 4-syllable long number words. Accordingly, we only

made a few minor changes in some corresponding formulations in the discussion and in one sentence of

the abstract.

Reviewer #1:

C1. All my comments have been addressed.

I just found one typo -- line 378, “suject” should be “subject”.

A2. We thank the reviewer for noticing this typo, it has been corrected in the current revision.

Reviewer #2:

C1. I read the new version of the manuscript and I don’t have any major remarks about the introduction, they

have been adequately addressed, just a few small points. However, unfortunately I see that the analysis

was still not carried out correctly and that could have large consequences, which requires a revision of

the analysis. I’ll explain in more detail below.

The initial maximal model that we had defined was the following:

RT ~ 1 + Age*Language*Size + (Age|Subject) + (1 + Language|Item)

I understand that this model and most of its varieties did not converge until you removed this random

slope, because what you did, modeling (Age|Subject), is impossible. You seem to be misunderstanding

what a random slope means. What you are really specifying here is that you are allowing the strength of

the effect of age to vary across subjects. But that does not make sense in your design. Age does not differ

within subjects, so you can’t have an individual effect of age for a single participant. That would require

longitudinal data.

So you could (actually, should) have a random slope of age|item (allowing age group differences to vary

across items) but not age|subject.

Likewise, you can, and actually should, include a random slope of size|subject whereas size|item would be

equally nonsensical as age|subject.Furthermore, you have multiple measures of language for both the subjects and the items, so including

language|item but not language|subject seems arbitrary. Especially given the diverse linguistic

backgrounds of the children, it is likely that language effects vary across subjects.

A maximal model contains all the random slopes that are possible (and no random slopes that are

impossible), so you haven’t actually run the maximal model.

If you have a random intercept only but no (or not all) random slope(s), then what you specify is that

participants show individual variation in their overall RT (assuming you used sum-to-zero contrasts,

please specify if you did, because if not the main effects mean something different), but language and size

affect their RTs identically for everyone. And similarly, items are allowed to differ in overall RT and the

language effect is allowed to vary across items, but age effects are specified to be exactly identical across

items. These assumptions aren’t very likely to hold, and simulation studies have shown that this tends to

yield p-values that are far too low. They can even be underestimated by a factor 10 (see the frightening

‘Type 1’ column of tables 5 and 6 from Barr, Levy, Scheepers and Tily, 2013, that shows that omitting

random slopes can inflate type 1 error rate to as much as .44 instead of .05). Barr (2013) (same author,

but not in any sense related to me, just to clarify) even showed that you also need to incorporate random

slopes for the highest possible order interaction term, and that having a maximal model except for the

interaction terms was surprisingly the worst-performing model.

Keeping all this in mind, this would yield the following maximal model:

RT ~ 1 + Age*Language*Size + (1+ Language*Size|Subject) + (1 + Age*Language|Item)

Barr, D. J. (2013). Random effects structure for testing interactions in linear mixed-effects models. Front

Psychol, 4, 328. https://doi.org/10.3389/fpsyg.2013.00328

Barr, D. J., Levy, R., Scheepers, C., & Tily, H. J. (2013). Random effects structure for confirmatory

hypothesis testing: Keep it maximal. Journal of Memory and Language, 68(3).

https://doi.org/10.1016/j.jml.2012.11.001

A1. We thank the reviewer for this important and very constructively explained comment about defining

the maximal model. We now see the mistake we made in defining the maximal model with random slopes

Age|Subjects and Size|Item. In the present revision, we implemented the suggested maximal model for

both tasks. However, this model resulted in a singular error for the reading task, likely due to high

correlations between random parts of Number Size per subject and Age per item. Therefore, we removed

some terms from the random parts (see lines 325 to 338), aiming to preserve the most comprehensive

model possible without singular fit.

The maximum model was defined taking into account individual differences by using random

slopes and intercepts per Subject for the interaction between Language and Number Size.

Moreover, item-related variability was modelled using random intercepts and slopes per Item for

the interaction between Language and Age. This led to the model with the following R syntax

form (A0):

(𝐴0) 𝑅𝑇 ~ 1 + 𝐴𝑔𝑒 ∗ 𝐿𝑎𝑛𝑔𝑢𝑎𝑔𝑒 ∗ 𝑁𝑢𝑚𝑏𝑒𝑟 𝑆𝑖𝑧𝑒 + (1 + 𝐿𝑎𝑛𝑔𝑢𝑎𝑔𝑒 ∗ 𝑁𝑢𝑚𝑏𝑒𝑟 𝑆𝑖𝑧𝑒|𝑆𝑢𝑏𝑗𝑒𝑐𝑡) + (1 +

𝐿𝑎𝑛𝑔𝑢𝑎𝑔𝑒 ∗ 𝐴𝑔𝑒 |𝐼𝑡𝑒𝑚).However, since the maximal model led to a singular fit (due to high correlations between the

random parts of Number Size per Subject and Age per Item ) we reduced the complexity of the

model by removing those problematic random terms [78]. Therefore, the final model takes the

following R syntax form (A):

(𝐴) 𝑅𝑇 ~ 1 + 𝐴𝑔𝑒 ∗ 𝐿𝑎𝑛𝑔𝑢𝑎𝑔𝑒 ∗ 𝑁𝑢𝑚𝑏𝑒𝑟 𝑆𝑖𝑧𝑒 + (1 + 𝐿𝑎𝑛𝑔𝑢𝑎𝑔𝑒|𝑆𝑢𝑏𝑗𝑒𝑐𝑡) + (1 + 𝐿𝑎𝑛𝑔𝑢𝑎𝑔𝑒|𝐼𝑡𝑒𝑚).

This final model led to the same pattern of significance than in our previous manuscript version, except

for a significant three-way interaction between Age, Language and Number Size. However, the three-way

interaction was neither significant with z-score nor with subset dataset of 4 syllable long number words.

We therefore preserved our overall interpretation of the results, while adapting the wording in the abstract

(lines 31 to 33) and in the discussion (lines 510 to 511 and l. 588-591) to reflect the final result pattern.

Furthermore, we adapted the S2 Tables 1 to 6 and follow-up statistics with the values of the current model

(additional changes are due to the removal of post-error slowing, see A3.). In addition, during the process

of re-calculating and re-writing the results we noticed and corrected some small errors and typos, which

we did not note.

A few minor points:

C2. [R2C2] I also doubt if ‘transparency of power’ is the best term. It sounds nice/interesting but it is only the

names of the tens that are affected, not the hundreds or thousands (admittedly, it returns for tens of

thousands, but still). So is not it just ‘transparency of tens/decades’?

[R2A2]: We thank the reviewer for this comment, we decided to use this terminology in the sense that it

has been proposed in (Bahnmueller et al., 2018; Dowker & Nuerk, 2017), which is to “derive the power

of each number directly from the number word” which we believe would include the concept of

“transparency of tens”. In that sense vigesimal number words in French would fit with this definitions

since, “quatre-vingt” (80), literally “four-twenty”, does not provide a direct way to derive the

corresponding eight decimal power.

C2. I understand that, my point was that referring to this phenomenon as ‘transparency of power’

is a bit general since only a few tens are affected (70/80/90), and all the higher powers are fully

transparent. (for example 800 is huit cent, and 8000 is huit mille, there is nothing vigesimal going

on there.) Perhaps ‘transparency of power in the tens’ would better capture it but I understand

that that doesn’t sound very catchy.

A2. We thank the reviewer for this additional comment, indeed the suggested “transparency of

power in the tens” would be a finer terminology corresponding closer to the language structure

and leads to an interesting discussion. To be super-precise it should be named “transparency of

power in the tens between seventy and ninety”, since not only hundredths and thousands, but

other decades as well are not affected (i.e. 40 is not “deux-vingt”, as it would be in Basque). In

(Haspelmath et al., 2005) French is not even considered a decimal system and Basque is

considered a hybrid vigesimal-decimal for the reason that 3 and 4 digit numbers follow the

decimal structure as pointed out by the reviewer. However, we argue that the linguistic-cognitive

interaction would be caused by a “change of base” (Ms. line.102). We compare 30/40/50 vs.

70/80/90 in French, where the change of base occurs, so we reason that what we observed is a“transparency of power” phenomenon and therefore prefer to conserve this more concise

terminology.

C2. [see R2C10] Line 285. Have you considered post-error slowing (the ‘oops’ effect)? It may be wise to

remove RTs after an error too.

R2A10. We thank the reviewer for this comment. Note that incorrect responses are excluded from the

reaction time analyses of both tasks, hence excluding possible “post-error” slowed down trials (lines 394-

397). Finally, for reaction time analyses all incorrect responses were removed, additionally removing 2.93

% of the reading task and 3 % of the matching task, corresponding to the total error rate.

C2. That is not post-error slowing. Post means after. So as I said, post-error slowing refers to the

reaction times for the trial right AFTER an error. They tend to be slower because the participant

often realizes they just made a mistake and they are distracted. That is the oops effect.

A2. We thank the reviewer for this clarification. Since the accuracy rate is quite high in our

study, it makes sense to remove those trials following an error, as they might have been

additionally slower. In the present revision, we removed all trials following an error in both tasks

(within each block and before we removed the ‘60s to respect the chronological order of

presentation in the experiment). This had a slight influence on the descriptive, and we draw the

figures anew to exclude post-error slow down trials there too. See lines 296-298:

As suggested by an anonymous reviewer, we additionally removed the trials following an

error that might be affected by post-error slowing in particular given the high accuracy

rate [75], thus additionally accounting for 1.95 % and 5.63 % of the trials respectively.

We additionally re-wrote the descriptive S1 Tables 1 to 4, since the values changed after

removing both incorrect and subsequent trials corresponding to post-error slow-down and

corrected some typos.

C3. Concerning the model syntax, to be clear, I didn’t necessarily want the whole syntax written down in R

language in the manuscript, I put it here as a clarifying question in the review as to find out what you did

exactly because the text in the previous version was ambiguous. But if you prefer you can also describe

the random slopes and intercepts verbally, as not every reader will be familiar with the specifics of R

syntax. It does show really transparently what you did though, so I’m not saying it’s a bad thing, if that’s

your choice it’s fine. Just to clear up a possible misunderstanding that you don’t only put it in because I

said so, because that was not my intention.

A3. We understood it was not the reviewer’s intention to make us report the R syntax. We choose to

report it, as it is the most direct way to convey the model, for those familiar with R. In particular, since

there are at the moment no standards on how to report linear mixed models.

---

## [Decision Letter · Decision Letter 3]

27 Jun 2022

PONE-D-21-18244R3Number transcoding in bilinguals - a transversal developmental studyPLOS ONE

Dear Dr. Lachelin,

Thank you for submitting your manuscript to PLOS ONE. I have sent it to a reviewer of the previous version. As you will see below, the reviewer is still not satisfied with the way you dealt with the statistical analysis. I will give you a final opportunity to respond to the reviewer, either by modifying your analyses in ways suggested by the reviewer or by explaining why you chose another analysis strategy. I will make a decision upon receiving your revised manuscript and response letter.

We look forward to receiving your revised manuscript.

Kind regards,

Jérôme Prado

Academic Editor

PLOS ONE

Reviewers' comments:

Reviewer's Responses to Questions

**Comments to the Author**

1. If the authors have adequately addressed your comments raised in a previous round of review and you feel that this manuscript is now acceptable for publication, you may indicate that here to bypass the “Comments to the Author” section, enter your conflict of interest statement in the “Confidential to Editor” section, and submit your "Accept" recommendation.

Reviewer #2: (No Response)

2. Is the manuscript technically sound, and do the data support the conclusions?

Reviewer #2: Yes

3. Has the statistical analysis been performed appropriately and rigorously? 

Reviewer #2: No

4. Have the authors made all data underlying the findings in their manuscript fully available?

Reviewer #2: Yes

5. Is the manuscript presented in an intelligible fashion and written in standard English?

Reviewer #2: Yes

6. Review Comments to the Author

Reviewer #2: Unfortunately, the one important point that I made the previous time was not solved. The previous time I recommended to redo the analyses with the inclusion of random slopes. The authors tried this but it looks like they immediately removed most of them upon a singular fit error, without trying other approaches. Now I know that singular fit errors can be very difficult. But removing random slopes is exactly the strategy that Barr et al. explain to avoid at all cost. This is in the reference I sent the previous time (Barr, Levy, Scheepers, & Tily, 2013, page 276) and I explained elaborately in the previous review and in the papers I recommended so I won’t repeat myself here. Alpha inflation is thus still a potential problem. Barr et al. give plenty of alternative options that should be tried first, on p.276. Follow these steps, or give an argumentation of why these steps were not taken.

Hopefully the results will still be more or less the same and then it is only a minor revision. But it could also be a major revision depending on what happens.

I’m also still missing the coding scheme that was used. Was it sum to zero coding (= effect coding) coding, or dummy coding (= treatment coding)? The contrast matters for the correct interpretation of the main effects.

7. PLOS authors have the option to publish the peer review history of their article (what does this mean?). If published, this will include your full peer review and any attached files.

Reviewer #2: No

---

## [Author Response · Author response to Decision Letter 3]

4 Aug 2022

Dear reviewer,

Thank you again for your points regarding the model selection. We will shortly address the minor point regarding the contrasts and extensively the major point regarding model selection. 

Regarding the contrasts, per default the contrasts are set to sum when analyzing data with afex’s mixed: we added this information (p. 15, l. 344):

All contrasts were set to sum.

Regarding model selection, the importance of keeping the model as maximal as possible appeared clearly to us following your very helpful explanations in review n°3 (as suggested by Barr et al., 2013). The suggested theoretical maximum random effect model was consequently (MM):

(MM) RT ~ 1 + Age * Language * Number Size + (1 + Language * Number Size | Subject) + (1 + Age * Language | Item)

This model nevertheless led to singular fit. As reported in our previous answer (A1.), singular fit is due to correlations of (or approaching) +/- 1 or 0 between the random terms. As recommended by (Barr et al., 2013) in this case, we proceeded by backward selection:

“we removed some terms from the random parts (see lines 325 to 338), aiming to preserve the most comprehensive model possible without singular fit.”

And in the corresponding manuscript (l. 335 to 338)

“[…] led to a singular fit (due to high correlations between the random parts of Number Size per Subject and Age per Item ) we reduced the complexity of the model by removing those problematic random terms”

To solve singularity issues, it is recommended to either remove the problematic components or use Bayesian methods (see isSingular ?). We decided to degrade the maximal model (MM), since the use of Bayesian methods is beyond the scope of the present study. The choice of which random terms to remove was based on the variance correlation matrix (VarCor), starting with the terms with correlations of +/-1. Hence, for example the random parameter Language * Number Size | Subject was decomposed and tested separately into Language | Subject, and Number Size | Subject. Many of those models lead to singular fits again, which is not surprising given the small number of participants and trials in the data. 

At this point, we want to highlight that Barr et al. (2013)’s theoretically driven model selection method is not the sole way to proceed with model selection. How to proceed is still debated among statisticians and there are as of today no golden standards regarding model selection, with different guidelines and procedures between authors. For example:

• Matuschek et al. (2017) argue that the simplification of the model must be defined a priori.

• Bates et al. (2018) argue for “parsimonious models” instead of maximal. They argue that the model parametrization must be supported by the data since overparametrized models might be uninterpretable. The reason is that adding random terms might not add significantly explained variance. 

o “However, the advice to “keep it maximal” often creates hopelessly over-specified random effects because the number of correlation parameters to estimate rises quickly with the dimension of the random-effects vectors. The information in the data may not be sufficient to support estimations of such complex models and may result in singular covariance matrices, even when the LMM is identifiable in principle. In this case, we need to replace the complex LMM specification by a more parsimonious one”

Now, we consulted three different experts in statistics (from the LUCET group at University Luxembourg, University Potsdam, and university Marseille-Aix-Provence) who all agreed about the absence of a golden standard in the model selection process and could not point to a major issue in our model selection. Nevertheless, for the sake of completeness, and to settle the question of the robustness of our results and model, we decided to run all possible models by decomposing the interaction A*B or the random terms into A + B + A:B (see Table 1). We proceeded by systematically removing one term, one correlation, and one intercept at a time. For each formula, we retrieved the convergence and singularity errors, BIC and AIC on the number reading task’s data. This procedure indicates that most of the models lead to singular fits, but we could also find a converging model with an additional term M2 compared to the model M1 included in our previous manuscript version. 

(M2) RT ~ 1 + Age * Language * Number Size + (1 + Language + Number Size| Subject) + (1 + Language | Item)

Despite that this model M2 also converged for z-scores and 4-syllable long number words, it led to singular fits in the corresponding analyses of the verbal-visual matching task’s RT. The following most complex models without singular fits are:

(M1) RT ~ 1 + Age * Language * Number Size + (1 + Language | Subject) + (1 + Language | Item)

(M3) RT ~ 1 + Age * Language * Number Size + (1 + Number Size| Subject) + (1 + Language | Item)

Both models fitted for the number naming task’s data did not significantly differ in an ANOVA comparison. However, M1 has smaller BIC, and was therefore selected as the final model. Therefore we keep M1, as it was implemented in the previous revision, since it is the maximal possible model that fits with both datasets (please note that M2 gives the same results pattern as M1 in the number reading task, see Table 2). 

More generally, the results are in line with the results of all previous analyses. Indeed the significance pattern matches across the numerous analyses we conducted at the different stages of the revision of our work: from the first submitted ANOVA with the additional analyses on z-score transformed RT, and on stimuli restricted to 4 syllables-long number words, to the corresponding linear mixed model without 60’s decades and including the supplementary analyses suggested by the reviewer considering the decades as additional factors and removal of post-error slipping trials (see Table 2).

We would like to thank again the reviewer for the very thorough and constructive comments and suggestions throughout the process.

References:

Barr, D. J., Levy, R., Scheepers, C., & Tily, H. J. (2013). Random effects structure for confirmatory hypothesis testing: Keep it maximal. Journal of Memory and Language, 68(3), 255–278. https://doi.org/10.1016/j.jml.2012.11.001

Bates, D., Kliegl, R., Vasishth, S., & Baayen, H. (2018). Parsimonious Mixed Models (arXiv:1506.04967). arXiv. http://arxiv.org/abs/1506.04967

Matuschek, H., Kliegl, R., Vasishth, S., Baayen, H., & Bates, D. (2017). Balancing Type I error and power in linear mixed models. Journal of Memory and Language, 94, 305–315. https://doi.org/10.1016/j.jml.2017.01.001

Table 1: All degradations of the maximal model that were tested with their corresponding convergence, singularity errors; AIC and BIC.

formula Not Converge Is Singular AIC BIC

 1RT ~ 1 + Age * Lan * Siz + (1 + Lan + Siz | ID) + (1 + Age + Lan + Age:Lan | Itm) 1 1 28589.8 28923

 RT ~ 0 + Age * Lan * Siz + (1 + Lan + Siz | ID) + (1 + Age + Lan + Age:Lan | Itm) 1 1 28589.8 28923

 RT ~ 1 + Age * Lan * Siz + (0 + Lan + Siz | ID) + (1 + Age + Lan + Age:Lan | Itm) 1 1 28589.8 28923

 RT ~ 1 + Age * Lan * Siz + (1 + Lan + Siz | ID) + (0 + Age + Lan + Age:Lan | Itm) 1 1 28589.8 28923

 RT ~ 1 + Age * Lan * Siz + (0 + Lan + Siz | ID) + (0 + Age + Lan + Age:Lan | Itm) 1 1 28589.8 28923

 RT ~ 0 + Age * Lan * Siz + (1 + Lan + Siz | ID) + (0 + Age + Lan + Age:Lan | Itm) 1 1 28589.8 28923

 RT ~ 0 + Age * Lan * Siz + (0 + Lan + Siz | ID) + (1 + Age + Lan + Age:Lan | Itm) 1 1 28589.8 28923

 RT ~ 0 + Age * Lan * Siz + (0 + Lan + Siz | ID) + (0 + Age + Lan + Age:Lan | Itm) 1 1 28589.8 28923

 RT ~ 1 + Age * Lan * Siz + (1 + Lan + Siz + Lan:Siz | ID) + (1 + Age + Lan | Itm) 1 1 28484.9 28722

 RT ~ 0 + Age * Lan * Siz + (1 + Lan + Siz + Lan:Siz | ID) + (1 + Age + Lan | Itm) 1 1 28484.9 28722

 RT ~ 1 + Age * Lan * Siz + (0 + Lan + Siz + Lan:Siz | ID) + (1 + Age + Lan | Itm) 1 1 28484.9 28722

 RT ~ 1 + Age * Lan * Siz + (1 + Lan + Siz + Lan:Siz | ID) + (0 + Age + Lan | Itm) 1 1 28484.9 28722

 RT ~ 1 + Age * Lan * Siz + (0 + Lan + Siz + Lan:Siz | ID) + (0 + Age + Lan | Itm) 1 1 28484.9 28722

 RT ~ 0 + Age * Lan * Siz + (1 + Lan + Siz + Lan:Siz | ID) + (0 + Age + Lan | Itm) 1 1 28484.9 28722

 RT ~ 0 + Age * Lan * Siz + (0 + Lan + Siz + Lan:Siz | ID) + (1 + Age + Lan | Itm) 1 1 28484.9 28722

 RT ~ 0 + Age * Lan * Siz + (0 + Lan + Siz + Lan:Siz | ID) + (0 + Age + Lan | Itm) 1 1 28484.9 28722

 RT ~ 1 + Age * Lan * Siz + (1 + Lan + Siz | ID) + (1 + Age + Lan | Itm) 1 1 28580.3 28795

 RT ~ 0 + Age * Lan * Siz + (1 + Lan + Siz | ID) + (1 + Age + Lan | Itm) 1 1 28580.3 28795

 RT ~ 1 + Age * Lan * Siz + (0 + Lan + Siz | ID) + (1 + Age + Lan | Itm) 1 1 28580.3 28795

 RT ~ 1 + Age * Lan * Siz + (1 + Lan + Siz | ID) + (0 + Age + Lan | Itm) 1 1 28580.2 28795

 RT ~ 1 + Age * Lan * Siz + (0 + Lan + Siz | ID) + (0 + Age + Lan | Itm) 1 1 28580.3 28795

 RT ~ 0 + Age * Lan * Siz + (1 + Lan + Siz | ID) + (0 + Age + Lan | Itm) 1 1 28580.2 28795

 RT ~ 0 + Age * Lan * Siz + (0 + Lan + Siz | ID) + (1 + Age + Lan | Itm) 1 1 28580.3 28795

 RT ~ 0 + Age * Lan * Siz + (0 + Lan + Siz | ID) + (0 + Age + Lan | Itm) 1 1 28580.3 28795

 RT ~ 1 + Age * Lan * Siz + (1 + Siz | ID) + (1 + Age + Lan | Itm) 1 1 29105.5 29303

 RT ~ 0 + Age * Lan * Siz + (1 + Siz | ID) + (1 + Age + Lan | Itm) 1 1 29105.5 29303

 RT ~ 1 + Age * Lan * Siz + (0 + Siz | ID) + (1 + Age + Lan | Itm) 1 1 29105.5 29303

 RT ~ 1 + Age * Lan * Siz + (1 + Siz | ID) + (0 + Age + Lan | Itm) 1 1 29105.5 29303

 RT ~ 1 + Age * Lan * Siz + (0 + Siz | ID) + (0 + Age + Lan | Itm) 1 1 29107.7 29305

 RT ~ 0 + Age * Lan * Siz + (1 + Siz | ID) + (0 + Age + Lan | Itm) 1 1 29105.5 29303

 RT ~ 0 + Age * Lan * Siz + (0 + Siz | ID) + (1 + Age + Lan | Itm) 1 1 29105.5 29303

 RT ~ 0 + Age * Lan * Siz + (0 + Siz | ID) + (0 + Age + Lan | Itm) 1 1 29107.7 29305

 RT ~ 1 + Age * Lan * Siz + (1 + Lan | ID) + (1 + Age + Lan | Itm) 1 1 28599.3 28797

 RT ~ 0 + Age * Lan * Siz + (1 + Lan | ID) + (1 + Age + Lan | Itm) 1 1 28599.3 28797

 RT ~ 1 + Age * Lan * Siz + (0 + Lan | ID) + (1 + Age + Lan | Itm) 1 1 28599.3 28797

 RT ~ 1 + Age * Lan * Siz + (1 + Lan | ID) + (0 + Age + Lan | Itm) 1 1 28599.3 28797

 RT ~ 1 + Age * Lan * Siz + (0 + Lan | ID) + (0 + Age + Lan | Itm) 1 1 28603.1 28800

 RT ~ 0 + Age * Lan * Siz + (1 + Lan | ID) + (0 + Age + Lan | Itm) 1 1 28599.3 28797

 RT ~ 0 + Age * Lan * Siz + (0 + Lan | ID) + (1 + Age + Lan | Itm) 1 1 28599.3 28797

 RT ~ 0 + Age * Lan * Siz + (0 + Lan | ID) + (0 + Age + Lan | Itm) 1 1 28603.1 28800

 2RT ~ 1 + Age * Lan * Siz + (1 + Lan + Siz | ID) + (1 + Lan | Itm) 0 0 28576.5 28723

 RT ~ 0 + Age * Lan * Siz + (1 + Lan + Siz | ID) + (1 + Lan | Itm) 0 0 28576.5 28723

 RT ~ 1 + Age * Lan * Siz + (0 + Lan + Siz | ID) + (1 + Lan | Itm) 0 0 28576.5 28723

 RT ~ 1 + Age * Lan * Siz + (1 + Lan + Siz | ID) + (0 + Lan | Itm) 0 0 28576.5 28723

 RT ~ 1 + Age * Lan * Siz + (0 + Lan + Siz | ID) + (0 + Lan | Itm) 0 0 28576.5 28723

 RT ~ 0 + Age * Lan * Siz + (1 + Lan + Siz | ID) + (0 + Lan | Itm) 1 0 28576.5 28723

 RT ~ 0 + Age * Lan * Siz + (0 + Lan + Siz | ID) + (1 + Lan | Itm) 0 0 28576.5 28723

 RT ~ 0 + Age * Lan * Siz + (0 + Lan + Siz | ID) + (0 + Lan | Itm) 0 0 28576.5 28723

 RT ~ 1 + Age * Lan * Siz + (1 + Lan + Siz | ID) + (1 + Age | Itm) 1 1 28600 28786

 RT ~ 0 + Age * Lan * Siz + (1 + Lan + Siz | ID) + (1 + Age | Itm) 1 1 28600 28786

 RT ~ 1 + Age * Lan * Siz + (0 + Lan + Siz | ID) + (1 + Age | Itm) 1 1 28600 28786

 RT ~ 1 + Age * Lan * Siz + (1 + Lan + Siz | ID) + (0 + Age | Itm) 1 1 28600 28786

 RT ~ 1 + Age * Lan * Siz + (0 + Lan + Siz | ID) + (0 + Age | Itm) 1 1 28607.4 28794

 RT ~ 0 + Age * Lan * Siz + (1 + Lan + Siz | ID) + (0 + Age | Itm) 1 1 28600 28786

 RT ~ 1 + Age * Lan * Siz + (0 + Lan + Siz | ID) + (1 + Age | Itm) 1 1 28600 28786

 RT ~ 0 + Age * Lan * Siz + (0 + Lan + Siz | ID) + (0 + Age | Itm) 1 1 28607.4 28794

 3RT ~ 1 + Age * Lan * Siz + (1 + Siz | ID) + (1 + Lan | Itm) 0 0 29103.8 29233

 RT ~ 0 + Age * Lan * Siz + (1 + Siz | ID) + (1 + Lan | Itm) 0 0 29103.8 29233

 RT ~ 1 + Age * Lan * Siz + (0 + Siz | ID) + (1 + Lan | Itm) 0 0 29103.8 29233

 RT ~ 1 + Age * Lan * Siz + (1 + Siz | ID) + (0 + Lan | Itm) 0 0 29103.8 29233

 RT ~ 1 + Age * Lan * Siz + (0 + Siz | ID) + (0 + Lan | Itm) 0 0 29103.8 29233

 RT ~ 0 + Age * Lan * Siz + (1 + Siz | ID) + (0 + Lan | Itm) 0 0 29103.8 29233

 RT ~ 0 + Age * Lan * Siz + (0 + Siz | ID) + (1 + Lan | Itm) 0 0 29103.8 29233

 RT ~ 0 + Age * Lan * Siz + (0 + Siz | ID) + (0 + Lan | Itm) 0 0 29103.8 29233

 RT ~ 1 + Age * Lan * Siz + (1 + Siz | ID) + (1 + Age | Itm) 1 1 29124.3 29293

 RT ~ 0 + Age * Lan * Siz + (1 + Siz | ID) + (1 + Age | Itm) 1 1 29124.3 29293

 RT ~ 1 + Age * Lan * Siz + (0 + Siz | ID) + (1 + Age | Itm) 1 1 29124.3 29293

 RT ~ 1 + Age * Lan * Siz + (1 + Siz | ID) + (0 + Age | Itm) 1 1 29124.3 29293

 RT ~ 1 + Age * Lan * Siz + (0 + Siz | ID) + (0 + Age | Itm) 1 1 29124.3 29293

 RT ~ 0 + Age * Lan * Siz + (1 + Siz | ID) + (0 + Age | Itm) 1 1 29124.3 29293

 RT ~ 0 + Age * Lan * Siz + (0 + Siz | ID) + (1 + Age | Itm) 1 1 29124.3 29293

 RT ~ 0 + Age * Lan * Siz + (0 + Siz | ID) + (0 + Age | Itm) 1 1 29136.6 29306

 RT ~ 1 + Age * Lan * Siz + (1 + Lan | ID) + (1 + Age | Itm) 1 1 28619 28788

 RT ~ 0 + Age * Lan * Siz + (1 + Lan | ID) + (1 + Age | Itm) 1 1 28619 28788

 RT ~ 1 + Age * Lan * Siz + (0 + Lan | ID) + (1 + Age | Itm) 1 1 28619 28788

 RT ~ 1 + Age * Lan * Siz + (1 + Lan | ID) + (0 + Age | Itm) 1 1 28619 28788

 RT ~ 1 + Age * Lan * Siz + (0 + Lan | ID) + (0 + Age | Itm) 1 1 28619 28788

 RT ~ 0 + Age * Lan * Siz + (1 + Lan | ID) + (0 + Age | Itm) 1 1 28619 28788

 RT ~ 0 + Age * Lan * Siz + (0 + Lan | ID) + (1 + Age | Itm) 1 1 28619 28788

 RT ~ 0 + Age * Lan * Siz + (0 + Lan | ID) + (0 + Age | Itm) 1 1 28619 28788

 4RT ~ 1 + Age * Lan * Siz + (1 + Lan | ID) + (1 + Lan | Itm) 0 0 28594.1 28724

 RT ~ 0 + Age * Lan * Siz + (1 + Lan | ID) + (1 + Lan | Itm) 0 0 28594.1 28724

 RT ~ 1 + Age * Lan * Siz + (0 + Lan | ID) + (1 + Lan | Itm) 0 0 28594.1 28724

 RT ~ 1 + Age * Lan * Siz + (1 + Lan | ID) + (0 + Lan | Itm) 0 0 28594.1 28724

 RT ~ 1 + Age * Lan * Siz + (0 + Lan | ID) + (0 + Lan | Itm) 0 0 28594.1 28724

 RT ~ 0 + Age * Lan * Siz + (1 + Lan | ID) + (0 + Lan | Itm) 0 0 28594.1 28724

 RT ~ 0 + Age * Lan * Siz + (0 + Lan | ID) + (1 + Lan | Itm) 0 0 28594.1 28724

 RT ~ 0 + Age * Lan * Siz + (0 + Lan | ID) + (0 + Lan | Itm) 0 0 28594.1 28724

 RT ~ 1 + Age * Lan * Siz + (1 + Lan + Siz || ID) + (1 + Age + Lan + Age:Lan | Itm) 1 1 28603 28942

 RT ~ 0 + Age * Lan * Siz + (1 + Lan + Siz || ID) + (1 + Age + Lan + Age:Lan | Itm) 1 1 28603 28942

 RT ~ 1 + Age * Lan * Siz + (0 + Lan + Siz || ID) + (1 + Age + Lan + Age:Lan | Itm) 1 1 28601 28934

 RT ~ 1 + Age * Lan * Siz + (1 + Lan + Siz || ID) + (0 + Age + Lan + Age:Lan | Itm) 1 1 28612 28950

 RT ~ 1 + Age * Lan * Siz + (0 + Lan + Siz || ID) + (0 + Age + Lan + Age:Lan | Itm) 1 1 28601 28934

 RT ~ 0 + Age * Lan * Siz + (1 + Lan + Siz || ID) + (0 + Age + Lan + Age:Lan | Itm) 1 1 28612 28950

 RT ~ 0 + Age * Lan * Siz + (0 + Lan + Siz || ID) + (1 + Age + Lan + Age:Lan | Itm) 1 1 28601 28934

 RT ~ 0 + Age * Lan * Siz + (0 + Lan + Siz || ID) + (0 + Age + Lan + Age:Lan | Itm) 1 1 28601 28934

 RT ~ 1 + Age * Lan * Siz + (1 + Lan + Siz + Lan:Siz || ID) + (1 + Age + Lan | Itm) 1 1 28499 28775

 RT ~ 0 + Age * Lan * Siz + (1 + Lan + Siz + Lan:Siz || ID) + (1 + Age + Lan | Itm) 1 1 28499 28775

 RT ~ 1 + Age * Lan * Siz + (0 + Lan + Siz + Lan:Siz || ID) + (1 + Age + Lan | Itm) 1 1 28497 28768

 RT ~ 1 + Age * Lan * Siz + (1 + Lan + Siz + Lan:Siz || ID) + (0 + Age + Lan | Itm) 1 1 28499 28775

 RT ~ 1 + Age * Lan * Siz + (0 + Lan + Siz + Lan:Siz || ID) + (0 + Age + Lan | Itm) 1 1 28497 28768

 RT ~ 0 + Age * Lan * Siz + (1 + Lan + Siz + Lan:Siz || ID) + (0 + Age + Lan | Itm) 1 1 28499 28775

 RT ~ 0 + Age * Lan * Siz + (0 + Lan + Siz + Lan:Siz || ID) + (1 + Age + Lan | Itm) 1 1 28497 28768

 RT ~ 0 + Age * Lan * Siz + (0 + Lan + Siz + Lan:Siz || ID) + (0 + Age + Lan | Itm) 1 1 28497 28768

 RT ~ 1 + Age * Lan * Siz + (1 + Lan + Siz || ID) + (1 + Age + Lan | Itm) 1 1 28594 28814

 RT ~ 0 + Age * Lan * Siz + (1 + Lan + Siz || ID) + (1 + Age + Lan | Itm) 1 1 28594 28814

 RT ~ 1 + Age * Lan * Siz + (0 + Lan + Siz || ID) + (1 + Age + Lan | Itm) 1 1 28592 28807

 RT ~ 1 + Age * Lan * Siz + (1 + Lan + Siz || ID) + (0 + Age + Lan | Itm) 1 1 28603 28823

 RT ~ 1 + Age * Lan * Siz + (0 + Lan + Siz || ID) + (0 + Age + Lan | Itm) 1 1 28592 28807

 RT ~ 0 + Age * Lan * Siz + (1 + Lan + Siz || ID) + (0 + Age + Lan | Itm) 1 1 28603 28823

 RT ~ 0 + Age * Lan * Siz + (0 + Lan + Siz || ID) + (1 + Age + Lan | Itm) 1 0 28592 28807

 RT ~ 0 + Age * Lan * Siz + (0 + Lan + Siz || ID) + (0 + Age + Lan | Itm) 1 1 28592 28807

 RT ~ 1 + Age * Lan * Siz + (1 + Siz || ID) + (1 + Age + Lan | Itm) 1 1 29108 29311

 RT ~ 0 + Age * Lan * Siz + (1 + Siz || ID) + (1 + Age + Lan | Itm) 1 1 29108 29311

 RT ~ 1 + Age * Lan * Siz + (0 + Siz || ID) + (1 + Age + Lan | Itm) 1 1 29106 29303

 RT ~ 1 + Age * Lan * Siz + (1 + Siz || ID) + (0 + Age + Lan | Itm) 1 1 29108 29311

 RT ~ 1 + Age * Lan * Siz + (0 + Siz || ID) + (0 + Age + Lan | Itm) 1 1 29108 29305

 RT ~ 0 + Age * Lan * Siz + (1 + Siz || ID) + (0 + Age + Lan | Itm) 1 1 29108 29311

 RT ~ 0 + Age * Lan * Siz + (0 + Siz || ID) + (1 + Age + Lan | Itm) 1 1 29106 29303

 RT ~ 0 + Age * Lan * Siz + (0 + Siz || ID) + (0 + Age + Lan | Itm) 1 1 29108 29305

 RT ~ 1 + Age * Lan * Siz + (1 + Lan || ID) + (1 + Age + Lan | Itm) 1 1 28601 28804

 RT ~ 0 + Age * Lan * Siz + (1 + Lan || ID) + (1 + Age + Lan | Itm) 1 1 28601 28804

 RT ~ 1 + Age * Lan * Siz + (0 + Lan || ID) + (1 + Age + Lan | Itm) 1 1 28599 28797

 RT ~ 1 + Age * Lan * Siz + (1 + Lan || ID) + (0 + Age + Lan | Itm) 1 1 28601 28804

 RT ~ 1 + Age * Lan * Siz + (0 + Lan || ID) + (0 + Age + Lan | Itm) 1 1 28603 28800

 RT ~ 0 + Age * Lan * Siz + (1 + Lan || ID) + (0 + Age + Lan | Itm) 1 1 28601 28804

 RT ~ 0 + Age * Lan * Siz + (0 + Lan || ID) + (1 + Age + Lan | Itm) 1 1 28599 28797

 RT ~ 0 + Age * Lan * Siz + (0 + Lan || ID) + (0 + Age + Lan | Itm) 1 1 28603 28800

 RT ~ 1 + Age * Lan * Siz + (1 + Lan + Siz || ID) + (1 + Lan | Itm) 1 0 28591 28743

 RT ~ 0 + Age * Lan * Siz + (1 + Lan + Siz || ID) + (1 + Lan | Itm) 1 0 28591 28743

 RT ~ 1 + Age * Lan * Siz + (0 + Lan + Siz || ID) + (1 + Lan | Itm) 0 0 28589 28735

 RT ~ 1 + Age * Lan * Siz + (1 + Lan + Siz || ID) + (0 + Lan | Itm) 1 0 28591 28743

 RT ~ 1 + Age * Lan * Siz + (0 + Lan + Siz || ID) + (0 + Lan | Itm) 1 0 28589 28735

 RT ~ 0 + Age * Lan * Siz + (1 + Lan + Siz || ID) + (0 + Lan | Itm) 1 0 28591 28743

 RT ~ 0 + Age * Lan * Siz + (0 + Lan + Siz || ID) + (1 + Lan | Itm) 1 0 28589 28735

 RT ~ 0 + Age * Lan * Siz + (0 + Lan + Siz || ID) + (0 + Lan | Itm) 1 0 28589 28735

 RT ~ 1 + Age * Lan * Siz + (1 + Lan + Siz || ID) + (1 + Age | Itm) 1 1 28615 28807

 RT ~ 0 + Age * Lan * Siz + (1 + Lan + Siz || ID) + (1 + Age | Itm) 1 1 28615 28807

 RT ~ 1 + Age * Lan * Siz + (0 + Lan + Siz || ID) + (1 + Age | Itm) 1 1 28613 28799

 RT ~ 1 + Age * Lan * Siz + (1 + Lan + Siz || ID) + (0 + Age | Itm) 1 1 28615 28807

 RT ~ 1 + Age * Lan * Siz + (0 + Lan + Siz || ID) + (0 + Age | Itm) 1 1 28620 28806

 RT ~ 0 + Age * Lan * Siz + (1 + Lan + Siz || ID) + (0 + Age | Itm) 1 1 28615 28807

 RT ~ 1 + Age * Lan * Siz + (0 + Lan + Siz || ID) + (1 + Age | Itm) 1 1 28613 28799

 RT ~ 0 + Age * Lan * Siz + (0 + Lan + Siz || ID) + (0 + Age | Itm) 1 1 28620 28806

 RT ~ 1 + Age * Lan * Siz + (1 + Siz || ID) + (1 + Lan | Itm) 1 0 29106 29241

 RT ~ 0 + Age * Lan * Siz + (1 + Siz || ID) + (1 + Lan | Itm) 0 0 29106 29241

 RT ~ 1 + Age * Lan * Siz + (0 + Siz || ID) + (1 + Lan | Itm) 0 0 29104 29233

 RT ~ 1 + Age * Lan * Siz + (1 + Siz || ID) + (0 + Lan | Itm) 0 0 29106 29241

 RT ~ 1 + Age * Lan * Siz + (0 + Siz || ID) + (0 + Lan | Itm) 0 0 29104 29233

 RT ~ 0 + Age * Lan * Siz + (1 + Siz || ID) + (0 + Lan | Itm) 0 0 29106 29241

 RT ~ 0 + Age * Lan * Siz + (0 + Siz || ID) + (1 + Lan | Itm) 0 0 29104 29233

 RT ~ 0 + Age * Lan * Siz + (0 + Siz || ID) + (0 + Lan | Itm) 0 0 29104 29233

 RT ~ 1 + Age * Lan * Siz + (1 + Siz || ID) + (1 + Age | Itm) 1 1 29126 29301

 RT ~ 0 + Age * Lan * Siz + (1 + Siz || ID) + (1 + Age | Itm) 1 1 29126 29301

 RT ~ 1 + Age * Lan * Siz + (0 + Siz || ID) + (1 + Age | Itm) 1 1 29124 29293

 RT ~ 1 + Age * Lan * Siz + (1 + Siz || ID) + (0 + Age | Itm) 1 1 29126 29301

 RT ~ 1 + Age * Lan * Siz + (0 + Siz || ID) + (0 + Age | Itm) 1 1 29124 29293

 RT ~ 0 + Age * Lan * Siz + (1 + Siz || ID) + (0 + Age | Itm) 1 1 29126 29301

 RT ~ 0 + Age * Lan * Siz + (0 + Siz || ID) + (1 + Age | Itm) 1 1 29124 29293

 RT ~ 0 + Age * Lan * Siz + (0 + Siz || ID) + (0 + Age | Itm) 1 1 29137 29306

 RT ~ 1 + Age * Lan * Siz + (1 + Lan || ID) + (1 + Age | Itm) 1 1 28627 28802

 RT ~ 0 + Age * Lan * Siz + (1 + Lan || ID) + (1 + Age | Itm) 1 1 28627 28802

 RT ~ 1 + Age * Lan * Siz + (0 + Lan || ID) + (1 + Age | Itm) 1 1 28619 28788

 RT ~ 1 + Age * Lan * Siz + (1 + Lan || ID) + (0 + Age | Itm) 1 1 28621 28796

 RT ~ 1 + Age * Lan * Siz + (0 + Lan || ID) + (0 + Age | Itm) 1 1 28619 28788

 RT ~ 0 + Age * Lan * Siz + (1 + Lan || ID) + (0 + Age | Itm) 1 1 28621 28796

 RT ~ 0 + Age * Lan * Siz + (0 + Lan || ID) + (1 + Age | Itm) 1 1 28619 28788

 RT ~ 0 + Age * Lan * Siz + (0 + Lan || ID) + (0 + Age | Itm) 1 1 28619 28788

 RT ~ 1 + Age * Lan * Siz + (1 + Lan || ID) + (1 + Lan | Itm) 1 1 28623 28758

 RT ~ 0 + Age * Lan * Siz + (1 + Lan || ID) + (1 + Lan | Itm) 1 1 28623 28758

 RT ~ 1 + Age * Lan * Siz + (0 + Lan || ID) + (1 + Lan | Itm) 0 0 28594 28724

 RT ~ 1 + Age * Lan * Siz + (1 + Lan || ID) + (0 + Lan | Itm) 0 0 28596 28731

 RT ~ 1 + Age * Lan * Siz + (0 + Lan || ID) + (0 + Lan | Itm) 0 0 28594 28724

 RT ~ 0 + Age * Lan * Siz + (1 + Lan || ID) + (0 + Lan | Itm) 1 0 28596 28731

 RT ~ 0 + Age * Lan * Siz + (0 + Lan || ID) + (1 + Lan | Itm) 0 0 28594 28724

 RT ~ 0 + Age * Lan * Siz + (0 + Lan || ID) + (0 + Lan | Itm) 0 0 28594 28724

RT ~ 1 + Age * Lan * Siz + (1 + Lan + Siz | ID) + (1 + Age + Lan + Age:Lan || Itm) 1 1 28618 29030

 RT ~ 0 + Age * Lan * Siz + (1 + Lan + Siz | ID) + (1 + Age + Lan + Age:Lan || Itm) 1 1 28618 29030

 RT ~ 1 + Age * Lan * Siz + (0 + Lan + Siz | ID) + (1 + Age + Lan + Age:Lan || Itm) 1 1 28618 29030

 RT ~ 1 + Age * Lan * Siz + (1 + Lan + Siz | ID) + (0 + Age + Lan + Age:Lan || Itm) 1 1 28616 29022

 RT ~ 1 + Age * Lan * Siz + (0 + Lan + Siz | ID) + (0 + Age + Lan + Age:Lan || Itm) 1 1 28616 29022

 RT ~ 0 + Age * Lan * Siz + (1 + Lan + Siz | ID) + (0 + Age + Lan + Age:Lan || Itm) 1 1 28616 29022

 RT ~ 0 + Age * Lan * Siz + (0 + Lan + Siz | ID) + (1 + Age + Lan + Age:Lan || Itm) 1 1 28618 29030

 RT ~ 0 + Age * Lan * Siz + (0 + Lan + Siz | ID) + (0 + Age + Lan + Age:Lan || Itm) 1 1 28616 29022

 RT ~ 1 + Age * Lan * Siz + (1 + Lan + Siz + Lan:Siz | ID) + (1 + Age + Lan || Itm) 1 1 28492 28723

 RT ~ 0 + Age * Lan * Siz + (1 + Lan + Siz + Lan:Siz | ID) + (1 + Age + Lan || Itm) 1 1 28492 28723

 RT ~ 1 + Age * Lan * Siz + (0 + Lan + Siz + Lan:Siz | ID) + (1 + Age + Lan || Itm) 1 1 28492 28723

 RT ~ 1 + Age * Lan * Siz + (1 + Lan + Siz + Lan:Siz | ID) + (0 + Age + Lan || Itm) 1 1 28490 28716

 RT ~ 1 + Age * Lan * Siz + (0 + Lan + Siz + Lan:Siz | ID) + (0 + Age + Lan || Itm) 1 1 28490 28716

 RT ~ 0 + Age * Lan * Siz + (1 + Lan + Siz + Lan:Siz | ID) + (0 + Age + Lan || Itm) 1 1 28490 28716

 RT ~ 0 + Age * Lan * Siz + (0 + Lan + Siz + Lan:Siz | ID) + (1 + Age + Lan || Itm) 1 1 28492 28723

 RT ~ 0 + Age * Lan * Siz + (0 + Lan + Siz + Lan:Siz | ID) + (0 + Age + Lan || Itm) 1 1 28490 28716

 RT ~ 1 + Age * Lan * Siz + (1 + Lan + Siz | ID) + (1 + Age + Lan || Itm) 1 1 28586 28794

 RT ~ 0 + Age * Lan * Siz + (1 + Lan + Siz | ID) + (1 + Age + Lan || Itm) 1 1 28586 28794

 RT ~ 1 + Age * Lan * Siz + (0 + Lan + Siz | ID) + (1 + Age + Lan || Itm) 1 1 28586 28794

 RT ~ 1 + Age * Lan * Siz + (1 + Lan + Siz | ID) + (0 + Age + Lan || Itm) 1 1 28589 28792

 RT ~ 1 + Age * Lan * Siz + (0 + Lan + Siz | ID) + (0 + Age + Lan || Itm) 1 1 28584 28787

 RT ~ 0 + Age * Lan * Siz + (1 + Lan + Siz | ID) + (0 + Age + Lan || Itm) 1 1 28589 28792

 RT ~ 0 + Age * Lan * Siz + (0 + Lan + Siz | ID) + (1 + Age + Lan || Itm) 1 1 28586 28794

 RT ~ 0 + Age * Lan * Siz + (0 + Lan + Siz | ID) + (0 + Age + Lan || Itm) 1 1 28584 28787

 RT ~ 1 + Age * Lan * Siz + (1 + Siz | ID) + (1 + Age + Lan || Itm) 1 1 29115 29307

 RT ~ 0 + Age * Lan * Siz + (1 + Siz | ID) + (1 + Age + Lan || Itm) 1 1 29115 29307

 RT ~ 1 + Age * Lan * Siz + (0 + Siz | ID) + (1 + Age + Lan || Itm) 1 1 29116 29308

 RT ~ 1 + Age * Lan * Siz + (1 + Siz | ID) + (0 + Age + Lan || Itm) 1 1 29113 29300

 RT ~ 1 + Age * Lan * Siz + (0 + Siz | ID) + (0 + Age + Lan || Itm) 1 1 29113 29300

 RT ~ 0 + Age * Lan * Siz + (1 + Siz | ID) + (0 + Age + Lan || Itm) 1 1 29113 29300

 RT ~ 0 + Age * Lan * Siz + (0 + Siz | ID) + (1 + Age + Lan || Itm) 1 1 29116 29308

 RT ~ 0 + Age * Lan * Siz + (0 + Siz | ID) + (0 + Age + Lan || Itm) 1 1 29113 29300

 RT ~ 1 + Age * Lan * Siz + (1 + Lan | ID) + (1 + Age + Lan || Itm) 1 1 28610 28802

 RT ~ 0 + Age * Lan * Siz + (1 + Lan | ID) + (1 + Age + Lan || Itm) 1 1 28610 28802

 RT ~ 1 + Age * Lan * Siz + (0 + Lan | ID) + (1 + Age + Lan || Itm) 1 1 28612 28804

 RT ~ 1 + Age * Lan * Siz + (1 + Lan | ID) + (0 + Age + Lan || Itm) 1 1 28603 28789

 RT ~ 1 + Age * Lan * Siz + (0 + Lan | ID) + (0 + Age + Lan || Itm) 1 1 28603 28789

 RT ~ 0 + Age * Lan * Siz + (1 + Lan | ID) + (0 + Age + Lan || Itm) 1 1 28603 28789

 RT ~ 0 + Age * Lan * Siz + (0 + Lan | ID) + (1 + Age + Lan || Itm) 1 1 28612 28804

 RT ~ 0 + Age * Lan * Siz + (0 + Lan | ID) + (0 + Age + Lan || Itm) 1 1 28603 28789

 RT ~ 1 + Age * Lan * Siz + (1 + Lan + Siz | ID) + (1 + Lan || Itm) 1 0 28578 28731

 RT ~ 0 + Age * Lan * Siz + (1 + Lan + Siz | ID) + (1 + Lan || Itm) 1 0 28578 28731

 RT ~ 1 + Age * Lan * Siz + (0 + Lan + Siz | ID) + (1 + Lan || Itm) 1 0 28578 28731

 RT ~ 1 + Age * Lan * Siz + (1 + Lan + Siz | ID) + (0 + Lan || Itm) 0 0 28576 28723

 RT ~ 1 + Age * Lan * Siz + (0 + Lan + Siz | ID) + (0 + Lan || Itm) 0 0 28576 28723

 RT ~ 0 + Age * Lan * Siz + (1 + Lan + Siz | ID) + (0 + Lan || Itm) 1 0 28576 28723

 RT ~ 0 + Age * Lan * Siz + (0 + Lan + Siz | ID) + (1 + Lan || Itm) 1 0 28578 28731

 RT ~ 0 + Age * Lan * Siz + (0 + Lan + Siz | ID) + (0 + Lan || Itm) 0 0 28576 28723

 RT ~ 1 + Age * Lan * Siz + (1 + Lan + Siz | ID) + (1 + Age || Itm) 1 1 28602 28794

 RT ~ 0 + Age * Lan * Siz + (1 + Lan + Siz | ID) + (1 + Age || Itm) 1 1 28602 28794

 RT ~ 1 + Age * Lan * Siz + (0 + Lan + Siz | ID) + (1 + Age || Itm) 1 1 28602 28794

 RT ~ 1 + Age * Lan * Siz + (1 + Lan + Siz | ID) + (0 + Age || Itm) 1 1 28600 28786

 RT ~ 1 + Age * Lan * Siz + (0 + Lan + Siz | ID) + (0 + Age || Itm) 1 1 28607 28794

 RT ~ 0 + Age * Lan * Siz + (1 + Lan + Siz | ID) + (0 + Age || Itm) 1 1 28600 28786

 RT ~ 1 + Age * Lan * Siz + (0 + Lan + Siz | ID) + (1 + Age || Itm) 1 1 28602 28794

 RT ~ 0 + Age * Lan * Siz + (0 + Lan + Siz | ID) + (0 + Age || Itm) 1 1 28607 28794

 RT ~ 1 + Age * Lan * Siz + (1 + Siz | ID) + (1 + Lan || Itm) 1 0 29106 29241

 RT ~ 0 + Age * Lan * Siz + (1 + Siz | ID) + (1 + Lan || Itm) 0 0 29106 29241

 RT ~ 1 + Age * Lan * Siz + (0 + Siz | ID) + (1 + Lan || Itm) 0 0 29106 29241

 RT ~ 1 + Age * Lan * Siz + (1 + Siz | ID) + (0 + Lan || Itm) 0 0 29104 29233

 RT ~ 1 + Age * Lan * Siz + (0 + Siz | ID) + (0 + Lan || Itm) 0 0 29104 29233

 RT ~ 0 + Age * Lan * Siz + (1 + Siz | ID) + (0 + Lan || Itm) 0 0 29104 29233

 RT ~ 0 + Age * Lan * Siz + (0 + Siz | ID) + (1 + Lan || Itm) 1 0 29106 29241

 RT ~ 0 + Age * Lan * Siz + (0 + Siz | ID) + (0 + Lan || Itm) 0 0 29104 29233

 RT ~ 1 + Age * Lan * Siz + (1 + Siz | ID) + (1 + Age || Itm) 1 1 29126 29301

 RT ~ 0 + Age * Lan * Siz + (1 + Siz | ID) + (1 + Age || Itm) 1 1 29126 29301

 RT ~ 1 + Age * Lan * Siz + (0 + Siz | ID) + (1 + Age || Itm) 1 1 29126 29301

 RT ~ 1 + Age * Lan * Siz + (1 + Siz | ID) + (0 + Age || Itm) 1 1 29124 29293

 RT ~ 1 + Age * Lan * Siz + (0 + Siz | ID) + (0 + Age || Itm) 1 1 29124 29293

 RT ~ 0 + Age * Lan * Siz + (1 + Siz | ID) + (0 + Age || Itm) 1 1 29124 29293

 RT ~ 0 + Age * Lan * Siz + (0 + Siz | ID) + (1 + Age || Itm) 1 1 29126 29301

 RT ~ 0 + Age * Lan * Siz + (0 + Siz | ID) + (0 + Age || Itm) 1 1 29137 29306

 RT ~ 1 + Age * Lan * Siz + (1 + Lan | ID) + (1 + Age || Itm) 1 1 28627 28802

 RT ~ 0 + Age * Lan * Siz + (1 + Lan | ID) + (1 + Age || Itm) 1 1 28627 28802

 RT ~ 1 + Age * Lan * Siz + (0 + Lan | ID) + (1 + Age || Itm) 1 1 28621 28796

 RT ~ 1 + Age * Lan * Siz + (1 + Lan | ID) + (0 + Age || Itm) 1 1 28619 28788

 RT ~ 1 + Age * Lan * Siz + (0 + Lan | ID) + (0 + Age || Itm) 1 1 28619 28788

 RT ~ 0 + Age * Lan * Siz + (1 + Lan | ID) + (0 + Age || Itm) 1 1 28619 28788

 RT ~ 0 + Age * Lan * Siz + (0 + Lan | ID) + (1 + Age || Itm) 1 1 28621 28796

 RT ~ 0 + Age * Lan * Siz + (0 + Lan | ID) + (0 + Age || Itm) 1 1 28619 28788

 RT ~ 1 + Age * Lan * Siz + (1 + Lan | ID) + (1 + Lan || Itm) 1 0 28596 28731

 RT ~ 0 + Age * Lan * Siz + (1 + Lan | ID) + (1 + Lan || Itm) 1 0 28596 28731

 RT ~ 1 + Age * Lan * Siz + (0 + Lan | ID) + (1 + Lan || Itm) 1 0 28596 28731

 RT ~ 1 + Age * Lan * Siz + (1 + Lan | ID) + (0 + Lan || Itm) 0 0 28594 28724

 RT ~ 1 + Age * Lan * Siz + (0 + Lan | ID) + (0 + Lan || Itm) 0 0 28594 28724

 RT ~ 0 + Age * Lan * Siz + (1 + Lan | ID) + (0 + Lan || Itm) 0 0 28594 28724

 RT ~ 0 + Age * Lan * Siz + (0 + Lan | ID) + (1 + Lan || Itm) 1 0 28596 28731

 RT ~ 0 + Age * Lan * Siz + (0 + Lan | ID) + (0 + Lan || Itm) 0 0 28594 28724

Notes: Lan = Language (German vs French), Size = Number Size (30’s, 40’s, 50’s vs 70’s 80’s, 90’s), Age = 5th, 8th, 11th and adults. ID = Subject, Itm = Item. Further degradation of the model were tested, but not shown here. The main models with all parameter, intercepts and random correlations (i.e. the “.0” models in the excel file), were tested using all_fit = TRUE and augmenting the iterations to 100000, leading to singularity and non convergence.

1 Maximum Model (MM)

2 Model however leads to singular fit with the auditory-visual matching task (M2)

3 ( M3 )

4 Model from the previous manuscript (M1)

 

Table 2: Comparison of the significance across the different analysis, sub-analysis and task datasets.

ANOVA:

 Lan * Size Age * Lan * Size 5th 8th 11th Adult 5th 8th 11th Adult

 Transparency of power LM2 cost

Number reading RT <0.001 .03 <.001 <.001 <.001 <.001 <.001 <.05 0.1 0.33

 zsRT <.0001 n.s. <.001 <.001 

 4Syll <0.001 n.s. <.002 <.002 

Visual-verbal matching RT <0.001 <0.001 <.001 <.001 <.01 <.01 <.001 n.s. n.s. n.s.

 zsRT <0.001 n.s. <.001 <.001 

 4Syll <0.001 <0.001 <.001 <.01 .09 .24 <.001 n.s. n.s. n.s.

Notes: Lan = Language (German vs French), Size = Number Size (30’s, 40’s, 50’s vs 70’s 80’s, 90’s), Age = 5th, 8th, 11th and adults. Transparency of power = hypothesis on the cost for 70’s 80’s, 90’s in French. LM2 cost = hypothesis for a cost in French. RT = reaction time. zRT = per-age group individual z-score transformation. 4Syll = data restricted to 4 syllable long number words.

M1: (RT ~ 1 + Age * Lan * Siz + (1 + Lan | ID) + (1 + Lan | Itm))

 Lan * Size Age * Lan * Size 5th 8th 11th Adult 5th 8th 11th Adult

 Transparency of power LM2 cost

Number reading RT <0.001 <.003 <.001 <.001 <.001 <.001 <.001 .05 .10 .32

 zRT <.0001 n.s. <.001 <.001 

 4Syll <0.001 n.s. <.002 <.002 

Visual-verbal matching RT <0.001 <0.001 <.001 <.001 <.01 <.01 <.001 n.s. n.s. n.s.

 zRT <0.001 n.s. <.001 <.001 

 4Syll <0.001 <0.001 <.001 <.01 .07 .15 <.001 n.s. n.s. n.s.

Notes: items with post-error slipping are removed.

Lan = Language (German vs French), Size = Number Size (30’s, 40’s, 50’s vs 70’s 80’s, 90’s), Age = 5th, 8th, 11th and adults. ID = Subject, Itm = Item. Transparency of power = hypothesis on the cost for 70’s 80’s, 90’s in French. LM2 cost = hypothesis for a cost in French. RT = reaction time. zRT = per-age group individual z-score transformation. 4Syll = data restricted to 4 syllable long number words.

M2: (RT ~ 1 + Age * Lan * Siz + (1 + Lan + Siz | ID) + (1 + Lan | Itm))

 Lan * Size Age * Lan * Size 5th 8th 11th Adult 5th 8th 11th Adult

 Transparency LM2 cost

Number reading RT <.001 0.006 <.001 <.001 <.001 <.001 <.001 <.05 .10 n.s.

 zRT <.001 .33 <.001 <.001 

 4Syll <.001 .36 <.001 <.001 

Visual-verbal matching Singular fit error

Notes: items following an error were removed.

Lan = Language (German vs French), Size = Number Size (30’s, 40’s, 50’s vs 70’s 80’s, 90’s), Age = 5th, 8th, 11th and adults. ID = Subject, Itm = Item. Transparency of power = hypothesis on the cost for 70’s 80’s, 90’s in French. LM2 cost = hypothesis for a cost in French. RT = reaction time. zRT = per-age group individual z-score transformation. 4Syll = data restricted to 4 syllable long number words.

---

## [Editor Report · Decision Letter 4]

9 Aug 2022

Number transcoding in bilinguals - a transversal developmental study

PONE-D-21-18244R4

Dear Dr. Lachelin,

Thank you for this last version of your manuscript. I carefully read the manuscript and your detailed response to reviewer #2. In my opinion, you adequately answered the statistical issue raised by the reviewer. I am therefore pleased to provisionally accept your manuscript for publication in PLOS ONE. Note that it will only be formally accepted for publication once it meets all outstanding technical requirements.

Kind regards,

Jérôme Prado

Academic Editor

PLOS ONE
---

## [Editor Report · Acceptance letter]

16 Aug 2022

PONE-D-21-18244R4 

Number transcoding in bilinguals - a transversal developmental study 

Dear Dr. Lachelin:

I'm pleased to inform you that your manuscript has been deemed suitable for publication in PLOS ONE. Congratulations! Your manuscript is now with our production department. 

Kind regards, 

on behalf of

Dr. Jérôme Prado 

Academic Editor

PLOS ONE